



# A fast, single-iteration ensemble Kalman smoother for sequential data assimilation

Colin Grudzien[1] and Marc Bocquet[2]

[1]Department of Mathematics and Statistics, University of Nevada, Reno, Reno, Nevada, USA
[2]CEREA, École des Ponts and EDF R&D, Île-de-France, France

**Correspondence:** Colin Grudzien (cgrudzien@unr.edu)

**Abstract.** Ensemble-variational methods form the basis of the state-of-the-art for nonlinear, scalable data assimilation, yet current designs may not be cost-effective for reducing prediction error in online, short-range forecast systems. We propose a novel, outer-loop optimization of the ensemble-variational formalism for applications in which forecast error dynamics are weakly nonlinear, such as synoptic meteorology. In order to rigorously derive our method and demonstrate its novelty, we review ensemble smoothers that appear throughout the literature in a unified Bayesian maximum-a-posteriori narrative, updating and simplifying some results. After mathematically deriving our technique, we systematically develop and inter-compare all studied schemes in the open-source Julia package DataAssimilationBenchmarks.jl, with pseudo-code provided for these methods. This high-performance numerical framework, supporting our mathematical results, produces extensive benchmarks that demonstrate the significant performance advantages of our proposed technique. In particular, our single-iteration ensemble Kalman smoother is shown both to improve prediction / posterior accuracy and to simultaneously reduce the leading order cost of iterative, sequential smoothers in a variety of relevant test cases for operational short-range forecasts. This long work is thus intended to present our novel single-iteration ensemble Kalman smoother, and to provide a theoretical and computational framework for the study of sequential, ensemble-variational Kalman filters and smoothers generally.

## 1 Introduction

### 1.1 Context

Ensemble-variational methods form the basis of the state-of-the-art for nonlinear, scalable data assimilation (DA) (Asch et al., 2016; Bannister, 2017). Estimators following an ensemble Kalman filter (EnKF) style analysis include the seminal maximum likelihood filter and four-dimensional ensemble var (Zupanski, 2005; Liu et al., 2008), the ensemble randomized maximum likelihood method (EnRML) (Gu and Oliver, 2007; Chen and Oliver, 2012; Raanes et al., 2019b), the iterative ensemble Kalman smoother (IEnKS) (Sakov et al., 2012; Bocquet and Sakov, 2013, 2014) and ensemble Kalman inversion (EKI) (Iglesias et al., 2013; Schillings and Stuart, 2018; Kovachki and Stuart, 2019). Unlike traditional 3D-VAR and 4D-VAR, which make use of the adjoint-based approximation for the gradient of the Bayesian maxiumum a posteriori (MAP) cost function, the above EnKF-based approaches utilize an ensemble of nonlinear forward model simulations to approximate the tangent-linear model. The MAP cost function gradient can be approximated by, e.g., finite-differences from the ensemble mean as in





the bundle variant of the IEnKS (Bocquet and Sakov, 2014). The ensemble-based approximation can thus eliminate the need to construct tangent-linear and adjoint code for the underlying nonlinear numerical model, which comes at a major cost in development time for operational DA systems.

These EnKF-based, ensemble-variational methods combine: the high-accuracy of the iterative solution to the Bayesian MAP formulation of the nonlinear DA problem (Sakov et al., 2012; Bocquet and Sakov, 2014); the relative simplicity of numerical model development and maintenance in ensemble-based DA (Kalnay et al., 2007); the ensemble-based analysis of time-dependent errors and possibly discontinuous, physical model parameters (Corazza et al., 2003); and a variational optimization of hyper-parameters for, e.g., inflation (Bocquet et al., 2015), localization (Lorenc, 2003) and surrogate models (Bocquet et al., 2020) to augment the estimation scheme. However, while the iterative ensemble-based schemes above are promising for moderately nonlinear and non-Gaussian DA, a barrier to their use in online, short-range forecast systems lies in the computational bottleneck of simulating the nonlinear, physics-based model in the ensemble sampling procedure. In order to produce forecast, filter and re-analyzed smoother statistics, these estimators may require multiple runs of the ensemble simulation over the data assimilation window (DAW) consisting of lagged past and current times.

When nonlinearity in the DA cycle is not dominated by the forecast error dynamics, as in synoptic meteorology, an iterative optimization over the dynamical forecast model may not produce a cost-effective reduction of forecast error. Particularly, when the linear-Gaussian approximation for the forecast error dynamics is adequate, nonlinearity in the DA cycle may instead by dominated by nonlinearity in the observation operator, nonlinearity in hyper-parameter optimization, and / or nonlinearity in temporally interpolating a re-analyzed, smoothed solution over the DAW. In this setting, our novel formulation of iterative, ensemble-variational smoothing has substantial advantages in balancing the computational cost / prediction accuracy trade off for these estimators.

## 1.2 Objectives and outline

This long paper achieves three connected primary objectives. Firstly, we review and refine a variety of already published smoother algorithms in a unified narrative of Bayesian MAP estimation. Pursuant to this, we simplify and update a number of results that appear dispersed throughout the literature. Secondly, we use this unified Bayesian framework to rigorously derive our novel iterative, sequential smoother, optimized for short-range forecast applications. Thirdly, we systematically develop all algorithms and our test cases in the open-source Julia package DataAssimilationBenchmarks.jl (Grudzien et al., 2021). This high-performance numerical framework, supporting our mathematical results, produces extensive simulation benchmarks, validating the performance advantages of our proposed technique. These simulations likewise establish fundamental performance metrics for all schemes, and certifies the accuracy and numerical efficiency of the Julia package DataAssimilationBenchmarks.jl. This manuscript is thus intended to introduce our novel iterative, sequential smoothing formalism, and to simultaneously provide a theoretical and computational framework for studying EnKF-based ensemble-variational filters and smoothers in perfect models.

Our proposed smoother technique is a relatively simple change of perspective with respect to existing Bayesian MAP estimators. We consider a hybridization of the classic ensemble Kalman smoother (EnKS) (Evensen and Van Leeuwen, 2000) with





the IEnKS to produce an iterative, fixed-lag, sequential smoother. We combine the filter solution and retrospective re-analysis

of the EnKS with a single iteration of the ensemble simulation over the lagged states, initialized with the re-analyzed, smoothed prior. The resulting scheme is a "single-iteration", ensemble Kalman smoother, denoted such as it produces its forecast, filter and re-analyzed smoother statistics with a single iteration of the ensemble simulation over the DAW. By doing so, we seek to minimize the leading order cost of ensemble-variational smoothing, i.e., the ensemble simulation in the nonlinear forecast model. However, we are free to iteratively optimize the filter cost function for any single observation without additional

iterations of the ensemble simulation.

We denote our general framework single-iteration smoothing, while the specific implementation presented here is denoted the single-iteration ensemble Kalman smoother (SIEnKS). For linear-Gaussian systems, with the perfect model hypothesis, the SIEnKS is a fully consistent Bayesian estimator, albeit one that uses redundant model simulations. When the forecast error dynamics are weakly nonlinear, yet other aspects of the DA cycle are moderately to strongly nonlinear, we demonstrate

the SIEnKS has prediction and posterior accuracy that is comparable to, and often better than, some fully iterative methods. However, the SIEnKS has a numerical cost that scales in iteratively solving the sequential filter cost functions for the DAW, i.e., the cost of the SIEnKS scales in matrix inversions in the ensemble dimension rather than in the cost of ensemble simulations, making our methodology suitable for operational short-range forecasting.

Over long DAWs, the performance of iterative, fixed-lag smoothers can degrade significantly due to the increasing nonlinear-

ity of temporally interpolating the posterior estimate over the window of lagged states. In a standard, single data assimilation (SDA) smoother, each observation is only assimilated once meaning that, in long lag windows, new observations are only distantly connected to the initial conditions of the ensemble simulation; in particular, this can introduce many local minima to the cost function, strongly affecting the performance of the optimization (Fillion et al., 2018, and references therein). To handle the increasing nonlinearity of the DA cycle over long DAWs, we derive a novel version of the method of multiple data

assimilation (MDA) (Emerick and Reynolds, 2013; Bocquet and Sakov, 2014). This new MDA technique takes advantage of the formulation of the single-iteration formalism to "partially" assimilate each observation within the DAW with a sequential, EnKS analysis. In particular, the filter analysis in the EnKS constrains the ensemble simulation to the observations while temporally interpolating the posterior estimate over the DAW – this constraint is shown to improve the filter and forecast accuracy at the end of long DAWs, as well as the stability of the joint posterior interpolation throughout. This key result is at the core

of how the SIEnKS is able to out-perform the predictive and posterior accuracy of a variety of sequential smoothing schemes while, at the same time, maintaining a lower leading-order numerical cost.

This work is organized as follows. Section 2 introduces our basic notations. Section 3 reviews the mathematical formalism for the ensemble transform Kalman filter (ETKF) based on the LETKF formalism of Hunt et al. (2007); Sakov and Oke (2008b); and Sakov and Bertino (2011). Subsequently, we discuss the extension of the ETKF to fixed-lag smoothing in terms

of: (i) the right-transform EnKS; (ii) the IEnKS; and (iii) the SIEnKS; each as different approximate solutions to the Bayesian MAP problem. Section 4 discusses several applications that distinguish the performance of these estimators. Section 5 provides an algorithmic cost analysis for these estimators and demonstrates forecast, filter and smoother benchmarks for the EnKS, the IEnKS and the SIEnKS in a variety of DA configurations. Section 6 summarizes these results and discusses future opportunities





for the single-iteration smoothing framework. Appendix A contains pseudo-code for the algorithms presented in this work,

which are implemented in the open-source Juila package DataAssimilationBenchmarks.jl (Grudzien et al., 2021). Note that, due to the difficulty of devising localization and / or hybridization for the IEnKS (Bocquet, 2016), we neglect a treatment of these techniques in this initial study of the SIEnKS, though this will be treated in a future work.

## 2   Notations

Matrices are denoted with upper-case bold and vectors with lower-case bold and italics. The standard Euclidean vector norm is

denoted $\| \boldsymbol{v} \| := \sqrt{\boldsymbol{v}^{\top}\boldsymbol{v}}$. For a symmetric, positive-definite matrix $\mathbf{A} \in \mathbb{R}^{N \times N}$, we denote the vector norm with respect to $\mathbf{A}$ as

$$\| \boldsymbol{v} \|_{\mathbf{A}} := \sqrt{\boldsymbol{v}^{\top}\mathbf{A}^{-1}\boldsymbol{v}}. \tag{1}$$

For a generic matrix $\mathbf{A} \in \mathbb{R}^{N \times M}$ with full column rank $M$, we denote the pseudo-inverse

$$\mathbf{A}^{\dagger} := \left(\mathbf{A}^{\top}\mathbf{A}\right)^{-1}\mathbf{A}^{\top}. \tag{2}$$

When $\mathbf{A}$ has full column rank as above, we define the vector "norm" with respect to $\mathbf{G} = \mathbf{A}\mathbf{A}^{\top}$ as

$$\| \boldsymbol{v} \|_{\mathbf{G}} := \sqrt{\left(\mathbf{A}^{\dagger}\boldsymbol{v}\right)^{\top}\left(\mathbf{A}^{\dagger}\boldsymbol{v}\right)}. \tag{3}$$

Note that in the case that $\mathbf{G}$ does not have full column rank, i.e., $N > M$, this is not a true norm on $\mathbb{R}^{N}$ as it is degenerate in the null space of $\mathbf{A}^{\dagger}$. This instead represents a lift of a non-degenerate norm in the column span of $\mathbf{A}$ to $\mathbf{R}^{N}$. In the case that $\boldsymbol{v}$ is in the column span of $\mathbf{A}$, we can equivalently write

$$\boldsymbol{v} = \mathbf{A}\boldsymbol{w}, \tag{4a}$$

$$\| \boldsymbol{v} \|_{\mathbf{G}} = \| \boldsymbol{w} \|, \tag{4b}$$

for a vector of weights $\boldsymbol{w} \in \mathbf{R}^{M}$.

Let $\boldsymbol{x}$ denote a random vector of physics-based model states. We assume that an initial, prior density on the model state $p(\boldsymbol{x}_0)$ is given, with a hidden Markov model of the form

$$\boldsymbol{x}_k = \mathcal{M}_k\left(\boldsymbol{x}_{k-1}\right), \tag{5a}$$

$$\boldsymbol{y}_k = \mathcal{H}_k\left(\boldsymbol{x}_k\right) + \boldsymbol{\epsilon}_k, \tag{5b}$$

determining the distribution of future states, with the dependence on the time $t_k$ denoted by the subscript $k$. For simplicity, we assume that $\Delta t := t_k - t_{k-1}$ is fixed for all $k$, though this is not a required restriction in any of the following arguments. The dimensions of the above system are denoted: (i) $N_x$ the model state dimension $\boldsymbol{x}_k \in \mathbb{R}^{N_x}$; (ii) $N_y$ the observation vector

dimension $\boldsymbol{y}_k \in \mathbb{R}^{N_y}$; and (iii) $N_e$ the ensemble-size, where an ensemble matrix is given as $\mathbf{E}_k \in \mathbb{R}^{N_x \times N_e}$. Model state and





observation variables are related via the (possibly) nonlinear observation operator $\mathcal{H}_k : \mathbb{R}^{N_x} \mapsto \mathbb{R}^{N_y}$. Observation noise $\epsilon_k$ is assumed to be an unbiased, white sequence such that

$$\mathbb{E}\left\{\epsilon_k \epsilon_l^\top\right\} = \delta_{k,l} \mathbf{R}_k, \tag{6}$$

where $\mathbb{E}$ is the expectation, $\mathbf{R}_k \in \mathbb{R}^{N_y \times N_y}$ is the observation error covariance matrix at time $t_k$ and $\delta_{k,l}$ denotes the Kronecker
delta function on the indices $k$ and $l$. The error covariance matrix $\mathbf{R}_k$ can be assumed invertible without losing generality. To avoid pathologies, we assume that the observation error covariance matrix is uniformly bounded in time.

The above configuration refers to a perfect model scenario in which the transition probability for $\mathrm{d}\boldsymbol{x} \subset \mathbb{R}^{N_x}$ is written

$$\mathcal{P}\left(\boldsymbol{x}_k \in \mathrm{d}\boldsymbol{x} | \boldsymbol{x}_{k-1}\right) = \delta_{\mathcal{M}_k(\boldsymbol{x}_{k-1})}(\mathrm{d}\boldsymbol{x}), \tag{7}$$

with $\delta_{\boldsymbol{v}}$ referring to the Dirac measure at $\boldsymbol{v} \in \mathbb{R}^{N_x}$. Similarly, we say that the transition "density" is proportional as

$$p(\boldsymbol{x}_k | \boldsymbol{x}_{k-1}) \propto \delta\left\{\boldsymbol{x}_k - \mathcal{M}_k\left(\boldsymbol{x}_{k-1}\right)\right\}, \tag{8}$$

where $\delta$ represents the Dirac distribution. The Dirac measure is singular with respect to Lebesgue measure, so this is simply a convenient abuse of notation that can be made rigorous with the generalized function theory of distributions (Taylor, 1996)[see section 3.4]. The perfect model assumption is utilized throughout this work to frame the studied assimilation schemes in a unified manner. Extending the single-iteration formalism to the case of model errors will be studied in a future work.

For a time series of model or observation states, with $l > k$, we define the notations

$$\boldsymbol{x}_{l:k} := \left\{\boldsymbol{x}_l, \boldsymbol{x}_{l-1}, \cdots, \boldsymbol{x}_k\right\}, \tag{9a}$$
$$\boldsymbol{y}_{l:k} := \left\{\boldsymbol{y}_l, \boldsymbol{y}_{l-1}, \cdots, \boldsymbol{y}_k\right\}. \tag{9b}$$

To distinguish between the various conditional probabilities under consideration, we make the following definitions. Let $l > k$, then the forecast density is denoted

$$p(\boldsymbol{x}_l | \boldsymbol{x}_{l-1:1}, \boldsymbol{y}_{l-1:1}), \tag{10}$$

the filter density is denoted

$$p(\boldsymbol{x}_l | \boldsymbol{y}_{l:1}) \tag{11}$$

and a smoother density for $\boldsymbol{x}_k$ given observations $\boldsymbol{y}_{l:1}$ is denoted

$$p(\boldsymbol{x}_k | \boldsymbol{y}_{l:1}). \tag{12}$$

In the above, the filter and smoother densities are marginals of the joint posterior density

$$p(\boldsymbol{x}_{l:1} | \boldsymbol{y}_{l:1}). \tag{13}$$



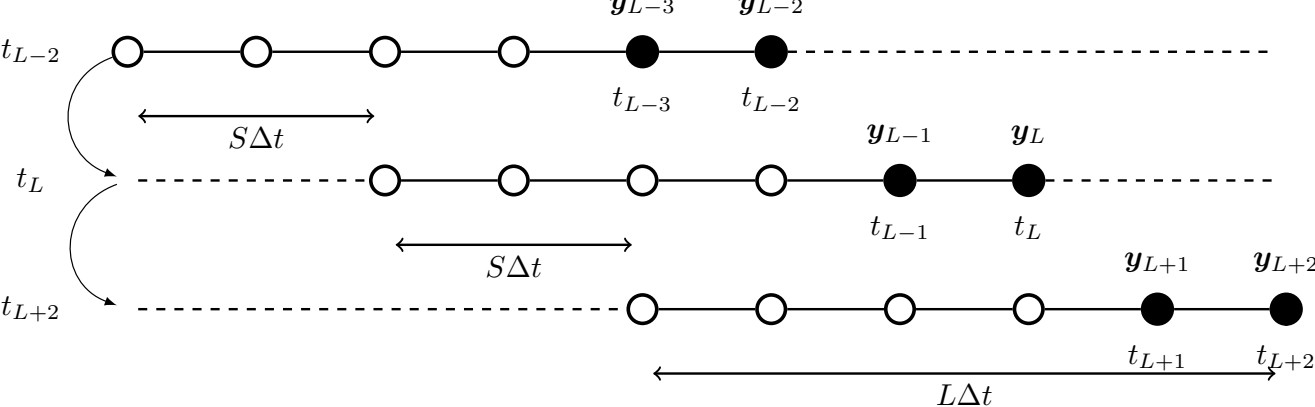

**Figure 1.** Three cycles of a shift $S = 2$, lag $L = 5$ smoother, cycle number is increasing top to bottom. Time indices on the left-hand margin indicate the current time for the associated cycle of the algorithm. New observations entering the current DAW are shaded black. The initial DAW ranges from $\{t_{L-6}, \cdots, t_{L-2}\}$. In the next cycle, this is shifted to $\{t_{L-4}, \cdots, t_L\}$, and thereafter is shifted to $\{t_{L-2}, \cdots, t_{L+2}\}$. States at the "zero" time indices: $t_{L-7}$ in the first cycle, $t_{L-5}$ in the second cycle, and $t_{L-3}$ in the third cycle, are estimated in addition to states in the DAW to connect the cycles in sequential DAWs.

The Markov hypothesis implies that the forecast density can furthermore be written as,

$$p(\boldsymbol{x}_k | \boldsymbol{x}_{k-1:1}, \boldsymbol{y}_{k-1:1}) = p(\boldsymbol{x}_k | \boldsymbol{x}_{k-1}). \tag{14}$$

For a fixed-lag smoother, we define a shift of length $S \geq 1$ analysis times and a lag of length $L \geq S$ analysis times, where

time $t_L$ denotes the present time. We use an algorithmically stationary DAW throughout the work, referring to the time indices $\{t_1, \cdots, t_L\}$. Smoother schemes estimate the joint posterior density $p(\boldsymbol{x}_{L:1} | \boldsymbol{y}_{L:1})$ or one of its marginals in a DA cycle; after each estimate is produced, the DAW is subsequently shifted in time by $S \times \Delta t$, and all states are re-indexed by $t_k := t_{k+S}$ to begin the next DA cycle. For a lag of $L$ and a shift of $S$, the observation vectors at times $\{t_{L-S+1}, \cdots, t_L\}$ correspond to observations newly entering the DAW at time $t_L$. When $S = L$, the DAWs are disconnected and adjacent in time, whereas for

$S < L$ there is an overlap between the estimated states in sequential DAWs. Figure 1 provides a schematic of how the DAW is shifted for a lag of $L = 5$ and shift $S = 2$. Following the convention in DA that there is no observation at time zero, in addition to the DAW, $\{t_1, \cdots, t_L\}$, states at time $t_0$ are estimated or utilized to connect estimates between adjacent / overlapping DAWs.

The ensemble matrix $\mathbf{E}_k^{\mathrm{i}} \in \mathbb{R}^{N_x \times N_e}$ is given a label i denoting the conditional density from which the ensemble is drawn. The ensemble $\mathbf{E}_k^{\mathrm{fore}}$ is assumed to have columns sampled independent and identically distributed (iid) according to the forecast

density, $\mathbf{E}_k^{\mathrm{filt}}$ is assumed to have columns iid according to the filter density and $\mathbf{E}_{k|L}^{\mathrm{smth}}$ is assumed to have columns iid according to a smoother density for the state at time $t_k$ given observations up to time $t_L$. Multiple data assimilation schemes will also utilize a balancing ensemble $\mathbf{E}_k^{\mathrm{bal}}$ and an MDA ensemble $\mathbf{E}_k^{\mathrm{mda}}$ to be defined in Section 4.3; time indices and labels may be suppressed when the meaning is still clear in context. The forecast model is given by $\mathbf{E}_{k+1}^{\mathrm{i}} = \mathcal{M}_{k+1}\left(\mathbf{E}_k^{\mathrm{j}}\right)$, where the type of ensemble input and output $\mathrm{i}, \mathrm{j} \in \{\mathrm{fore}, \mathrm{filt}, \mathrm{smth}, \mathrm{bal}, \mathrm{mda}\}$ (forecast / filter / smoother / balancing / MDA) is specified





according to the estimation scheme. We define the composition of the forecast model $\mathbf{E}_l^{\mathrm{i}} = \mathcal{M}_l \circ \cdots \circ \mathcal{M}_k = \mathcal{M}_{l:k}\left(\mathbf{E}_{k-1}^{\mathrm{j}}\right)$.
Let $\mathbf{1}$ denote the vector with all entries equal to one, such that the ensemble-based, empirical mean, the ensemble perturbation
matrix and the ensemble-based, empirical covariance are each defined as follows

$$\hat{\boldsymbol{x}}_k^{\mathrm{i}} := \mathbf{E}_k^{\mathrm{i}} \mathbf{1}/N_e, \tag{15a}$$

$$\mathbf{X}_k^{\mathrm{i}} := \mathbf{E}_k^{\mathrm{i}} - \hat{\boldsymbol{x}}_k^{\mathrm{i}} \mathbf{1}^\top$$

$$= \mathbf{E}_k^{\mathrm{i}}\left(\mathbf{I}_{N_e} - \mathbf{1}\mathbf{1}^\top/N_e\right), \tag{15b}$$

$$\mathbf{P}_k^{\mathrm{i}} := \mathbf{X}_k^{\mathrm{i}}\left(\mathbf{X}_k^{\mathrm{i}}\right)^\top/(N_e - 1). \tag{15c}$$

We define the background mean and covariance as

$$\overline{\boldsymbol{x}}_k^{\mathrm{i}} := \mathbb{E}\left\{\boldsymbol{x}_k^{\mathrm{i}}\right\}, \tag{16a}$$

$$\mathbf{B}_k^{\mathrm{i}} := \mathbb{E}\left\{\left[\boldsymbol{x}_k^{\mathrm{i}} - \overline{\boldsymbol{x}}_k^{\mathrm{i}}\right]\left[\boldsymbol{x}_k^{\mathrm{i}} - \overline{\boldsymbol{x}}_k^{\mathrm{i}}\right]^\top\right\}, \tag{16b}$$

to distinguish from the ensemble-based mean $\hat{\boldsymbol{x}}_i^{\mathrm{i}}$ and covariance $\mathbf{P}_k^{\mathrm{i}}$.

In the case where: (i) $\mathcal{M}_k := \mathbf{M}_k$ and $\mathcal{H}_k := \mathbf{H}_k$ are both linear transformations; (ii) the observation likelihood is given as

$$p(\boldsymbol{y}_k|\boldsymbol{x}_k) := N(\boldsymbol{y}_k - \mathbf{H}_k\boldsymbol{x}_k|\mathbf{0}, \mathbf{R}_k); \tag{17}$$

and (iii) the first prior is given as

$$\boldsymbol{x}_0 \sim N(\overline{\boldsymbol{x}}_0, \mathbf{B}_0); \tag{18}$$

we say that the DA configuration is of a perfect, linear-Gaussian model.

## 3  Deriving the SIEnKS

The single-iteration ensemble Kalman smoother is an outer-loop optimization of the filter, forecast and re-analysis steps of
other fixed-lag smoothers designed for prediction cycles with weakly nonlinear forecast error dynamics. This framework is not
restricted to any particular filter analysis; the ETKF analysis (Hunt et al., 2007) is utilized in the following for its operational
stability and efficiency, and in order to emphasize the commonality and differences between other well-known smoothing
schemes. Other types of filter analysis, such as the deterministic EnKF (DEnKF) of Sakov and Oke (2008a), are, furthermore,
compatible with the single-iteration ensemble Kalman smoother formalism presented in this work and may be considered in
future studies.

### 3.1  The ETKF

The filter problem can be expressed recursively in the Bayesian MAP formalism with an algorithmically stationary DAW as
follows. Suppose that there is a known filter density $p(\boldsymbol{x}_0|\boldsymbol{y}_0)$ from a previous DA cycle. Using the Markov hypothesis and the





independence of observation errors, we write the filter density up to proportionality via Bayes' law

$$p(\boldsymbol{x}_1|\boldsymbol{y}_{1:0}) \propto p(\boldsymbol{y}_1|\boldsymbol{x}_1,\boldsymbol{y}_0)p(\boldsymbol{x}_1,\boldsymbol{y}_0) \tag{19a}$$

$$\propto \underbrace{p(\boldsymbol{y}_1|\boldsymbol{x}_1)}_{(i)}\underbrace{\int p(\boldsymbol{x}_1|\boldsymbol{x}_0)p(\boldsymbol{x}_0|\boldsymbol{y}_0)\mathrm{d}\boldsymbol{x}_0}_{(ii)} \tag{19b}$$

as the product of the (i) likelihood of the observation given the forecast; and (ii) the forecast-prior. The forecast-prior (ii) is generated by the model propagation of the last filter density $p(\boldsymbol{x}_0|\boldsymbol{y}_0)$ with the transition kernel $p(\boldsymbol{x}_1|\boldsymbol{x}_0)$, marginalizing out $\boldsymbol{x}_0$. With a first prior density given, the above recursion inductively defines the forecast and filter densities, up to proportionality, at all times.

In the perfect, linear-Gaussian model, the forecast-prior and filter densities,

$$\int p(\boldsymbol{x}_1|\boldsymbol{x}_0)p(\boldsymbol{x}_0|\boldsymbol{y}_0)\mathrm{d}\boldsymbol{x}_0 \quad\text{and}\quad p(\boldsymbol{x}_1|\boldsymbol{y}_1), \tag{20}$$

are Gaussian. The Kalman filter equations recursively compute the mean $\overline{\boldsymbol{x}}_1^{\mathrm{fore}}/\overline{\boldsymbol{x}}_1^{\mathrm{filt}}$ and covariance $\mathbf{B}_1^{\mathrm{fore}}/\mathbf{B}_1^{\mathrm{filt}}$ of the random model state $\boldsymbol{x}_1$, parameterizing its distribution (Jazwinski, 1970). In this case, the filter problem can also be written in terms of the Bayesian MAP cost function

$$\mathcal{J}(\boldsymbol{x}_1) = \frac{1}{2}\parallel \boldsymbol{x}_1 - \overline{\boldsymbol{x}}_1^{\mathrm{fore}} \parallel_{\mathbf{B}_1^{\mathrm{fore}}}^2 + \frac{1}{2}\parallel \boldsymbol{y}_1 - \mathbf{H}_1\boldsymbol{x}_1 \parallel_{\mathbf{R}_1}^2 . \tag{21}$$

To render the above cost function into the right-transform analysis, define the matrix factor

$$\mathbf{B}_1^{\mathrm{fore}} := \boldsymbol{\Sigma}_1^{\mathrm{fore}}\left(\boldsymbol{\Sigma}_1^{\mathrm{fore}}\right)^\top, \tag{22}$$

where the choice of $\boldsymbol{\Sigma}_1^{\mathrm{fore}}$ can be arbitrary, but is typically given in terms of a singular value decomposition (SVD) (Sakov and Oke, 2008b). Instead of optimizing the cost function in Eq. (21) over the state vector $\boldsymbol{x}_1$, the optimization is equivalently written in terms of weights $\boldsymbol{w}$ where

$$\boldsymbol{x}_1 := \overline{\boldsymbol{x}}_1^{\mathrm{fore}} + \boldsymbol{\Sigma}_1^{\mathrm{fore}}\boldsymbol{w}; \tag{23}$$

thus re-writing Eq. (21) in terms of the weight vector $\boldsymbol{w}$, we obtain

$$\mathcal{J}(\boldsymbol{w}) = \frac{1}{2}\parallel \boldsymbol{w} \parallel^2 + \frac{1}{2}\parallel \boldsymbol{y}_1 - \mathbf{H}_1\overline{\boldsymbol{x}}_1^{\mathrm{fore}} - \mathbf{H}_1\boldsymbol{\Sigma}_1^{\mathrm{fore}}\boldsymbol{w} \parallel_{\mathbf{R}_1}^2 . \tag{24}$$

Further, for the sake of compactness, define

$$\overline{\boldsymbol{y}}_1 := \mathbf{H}_1\overline{\boldsymbol{x}}_1^{\mathrm{fore}}, \tag{25a}$$

$$\overline{\boldsymbol{\delta}}_1 := \mathbf{R}_1^{-\frac{1}{2}}\left(\boldsymbol{y}_1 - \overline{\boldsymbol{y}}_1\right), \tag{25b}$$

$$\boldsymbol{\Gamma}_1 := \mathbf{R}_1^{-\frac{1}{2}}\mathbf{H}_1\boldsymbol{\Sigma}_1^{\mathrm{fore}}. \tag{25c}$$





The vector $\overline{\boldsymbol{\delta}}_1$ is the innovation vector, weighted inverse proportionally to the observation uncertainty. The matrix $\boldsymbol{\Gamma}_1$, in one dimension with $\mathbf{H}_1 := 1$, is equal to the standard deviation of the model forecast relative to the standard deviation of the observation error.

The cost function Eq. (24) is hence further reduced to

$$\mathcal{J}(\boldsymbol{w}) = \frac{1}{2} \parallel \boldsymbol{w} \parallel^2 + \frac{1}{2} \parallel \overline{\boldsymbol{\delta}}_1 - \boldsymbol{\Gamma}_1 \boldsymbol{w} \parallel^2 . \tag{26}$$

This cost function is quadratic in $\boldsymbol{w}$ and can be globally minimized where $\nabla_{\boldsymbol{w}} \mathcal{J} = \mathbf{0}$. Notice,

$$\nabla_{\boldsymbol{w}} \mathcal{J} = \boldsymbol{w} - \boldsymbol{\Gamma}_1^{\top} \left( \overline{\boldsymbol{\delta}}_1 - \boldsymbol{\Gamma}_1 \boldsymbol{w} \right) ; \tag{27}$$

setting the gradient equal to zero for $\overline{\boldsymbol{w}}$ we find

$$\mathbf{0} = \overline{\boldsymbol{w}} - \boldsymbol{\Gamma}_1^{\top} \left( \overline{\boldsymbol{\delta}}_1 - \boldsymbol{\Gamma}_1 \overline{\boldsymbol{w}} \right) \tag{28a}$$

$$\Leftrightarrow \boldsymbol{\Gamma}_1^{\top} \overline{\boldsymbol{\delta}}_1 = \left( \mathbf{I}_{N_x} + \boldsymbol{\Gamma}_1 \boldsymbol{\Gamma}_1^{\top} \right) \overline{\boldsymbol{w}} \tag{28b}$$

$$\Leftrightarrow \quad \overline{\boldsymbol{w}} = \left( \mathbf{I}_{N_x} + \boldsymbol{\Gamma}_1 \boldsymbol{\Gamma}_1^{\top} \right)^{-1} \boldsymbol{\Gamma}_1^{\top} \overline{\boldsymbol{\delta}}_1 . \tag{28c}$$

From Eq. (27) notice that

$$\nabla_{\boldsymbol{w}} \mathcal{J}|_{\boldsymbol{w}=\mathbf{0}} = -\boldsymbol{\Gamma}_1^{\top} \overline{\boldsymbol{\delta}}_1 . \tag{29}$$

Similarly, taking the gradient of Eq. (27), we find that the Hessian, $\mathbf{H}_{\mathcal{J}} := \nabla_{\boldsymbol{w}}^2 \mathcal{J}$, is equal to

$$\mathbf{H}_{\mathcal{J}} = \left( \mathbf{I}_{N_x} + \boldsymbol{\Gamma}_1 \boldsymbol{\Gamma}_1^{\top} \right) . \tag{30}$$

Therefore, with $\boldsymbol{w} = \mathbf{0}$ corresponding to $\overline{\boldsymbol{x}}_1^{\mathrm{fore}}$ as the initialization of the assimilation algorithm, the MAP weights $\overline{\boldsymbol{w}}$ are determined by a single iteration of Newton's descent method (Nocedal and Wright, 2006) – for iterate $i$ this has the general form of

$$\boldsymbol{w}^{i+1} := \boldsymbol{w}^i - \mathbf{H}_{\mathcal{J}}^{-1} \nabla \mathcal{J}|_{\boldsymbol{w}=\boldsymbol{w}^i} . \tag{31}$$

The MAP weights define the maximum-a-posteriori model state,

$$\overline{\boldsymbol{x}}_1^{\mathrm{filt}} := \overline{\boldsymbol{x}}_1^{\mathrm{fore}} + \boldsymbol{\Sigma}_1^{\mathrm{fore}} \overline{\boldsymbol{w}} ; \tag{32}$$

under the perfect, linear-Gaussian model assumption, $\mathcal{J}$ can then be re-written in terms of the filter MAP estimate as

$$\mathcal{J}(\boldsymbol{x}_1) = \frac{1}{2} \parallel \boldsymbol{x}_1 - \overline{\boldsymbol{x}}_1^{\mathrm{filt}} \parallel_{\mathbf{B}_1^{\mathrm{filt}}}^2 \tag{33a}$$

$$\Leftrightarrow \mathcal{J}(\boldsymbol{w}) = \frac{1}{2} \parallel \overline{\boldsymbol{x}}_1^{\mathrm{fore}} - \boldsymbol{\Sigma}_1^{\mathrm{fore}} \boldsymbol{w} - \overline{\boldsymbol{x}}_1^{\mathrm{filt}} \parallel_{\mathbf{B}_1^{\mathrm{filt}}}^2 . \tag{33b}$$





Defining the matrix decomposition $\mathbf{B}_1^{\mathrm{filt}} = \boldsymbol{\Sigma}_1^{\mathrm{filt}} \left( \boldsymbol{\Sigma}_1^{\mathrm{filt}} \right)^{\top}$ and the change of variables

$$\boldsymbol{\Omega}_1 := \left( \boldsymbol{\Sigma}_1^{\mathrm{filt}} \right)^{-1} \boldsymbol{\Sigma}_1^{\mathrm{fore}}, \tag{34a}$$

$$\boldsymbol{\gamma}_1 := \left( \boldsymbol{\Sigma}_1^{\mathrm{filt}} \right)^{-1} \left( \overline{\boldsymbol{x}}_1^{\mathrm{fore}} - \overline{\boldsymbol{x}}_1^{\mathrm{filt}} \right), \tag{34b}$$

equation (33b) becomes

$$245 \quad \mathcal{J}(\boldsymbol{w}) = \frac{1}{2} \parallel \boldsymbol{\gamma}_1 - \boldsymbol{\Omega}_1 \boldsymbol{w} \parallel^2 . \tag{35}$$

Compute the Hessian $\mathbf{H}_{\mathcal{J}} = \nabla_{\boldsymbol{w}}^2 \mathcal{J}$ from each of Eqs. (26) and (35); by the equivalence we find

$$\left( \mathbf{I}_{N_x} + \boldsymbol{\Gamma}_1 \boldsymbol{\Gamma}_1^{\top} \right) = \boldsymbol{\Omega}_1^{\top} \boldsymbol{\Omega}_1 \tag{36a}$$

$$\Leftrightarrow \left( \mathbf{I}_{N_x} + \boldsymbol{\Gamma}_1 \boldsymbol{\Gamma}_1^{\top} \right) = \left( \boldsymbol{\Sigma}_1^{\mathrm{fore}} \right)^{\top} \left( \boldsymbol{\Sigma}_1^{\mathrm{filt}} \right)^{-\top} \left( \boldsymbol{\Sigma}_1^{\mathrm{filt}} \right)^{-1} \boldsymbol{\Sigma}_1^{\mathrm{fore}} \tag{36b}$$

$$\Leftrightarrow \quad \mathbf{B}_1^{\mathrm{filt}} = \boldsymbol{\Sigma}_1^{\mathrm{fore}} \left( \mathbf{I}_{N_x} + \boldsymbol{\Gamma}_1 \boldsymbol{\Gamma}_1^{\top} \right)^{-1} \left( \boldsymbol{\Sigma}_1^{\mathrm{fore}} \right)^{\top} . \tag{36c}$$

If we define the covariance transform

$$\mathbf{T} := \mathbf{H}_{\mathcal{J}}^{-\frac{1}{2}}, \tag{37}$$

this derivation above describes the square root Kalman filter recursion (Tippett et al., 2003), when written for the exact mean and covariance, recursively computed in the perfect, linear-Gaussian model. The covariance update

$$\mathbf{B}_1^{\mathrm{filt}} = \left( \boldsymbol{\Sigma}_1^{\mathrm{fore}} \mathbf{T} \right) \left( \boldsymbol{\Sigma}_1^{\mathrm{fore}} \mathbf{T} \right)^{\top} \tag{38}$$

is written entirely in terms of the matrix factor $\boldsymbol{\Sigma}_k^{\mathrm{i}}$ and the covariance transform $\mathbf{T}$, such that the background covariance need not be explicitly computed in order to produce recursive estimates. Likewise, the Kalman gain update to the mean state is reduced to Eq. (32), in terms of the weights and the matrix factor. This reduction is at the core of the efficiency of the ETKF, in which one typically makes a reduced-rank approximation to the background covariances $\mathbf{B}_1^{\mathrm{i}}$.

Using the ensemble-based, empirical estimates for the background, as in Eq. (15), a modification of the above argument
must be used to solve the cost function $\mathcal{J}$ in the ensemble span, without direct inversion of $\mathbf{P}_1^{\mathrm{fore}}$ when this is reduced rank. We replace the background covariance norm-square with one defined by the ensemble-based covariance,

$$\parallel \boldsymbol{v} \parallel_{\mathbf{P}_1^{\mathrm{i}}}^2 = (N_e - 1) \left[ \left( \mathbf{X}_1^{\mathrm{i}} \right)^{\dagger} \boldsymbol{v} \right]^{\top} \left[ \left( \mathbf{X}_1^{\mathrm{i}} \right)^{\dagger} \boldsymbol{v} \right] . \tag{39}$$

Define the ensemble-based estimates

$$\boldsymbol{x}_1 := \hat{\boldsymbol{x}}_1^{\mathrm{fore}} + \mathbf{X}_1^{\mathrm{fore}} \boldsymbol{w}, \tag{40a}$$

$$265 \quad \hat{\boldsymbol{y}}_1 := \mathbf{H}_1 \hat{\boldsymbol{x}}_1^{\mathrm{fore}}, \tag{40b}$$

$$\hat{\boldsymbol{\delta}}_1 := \mathbf{R}_1^{-\frac{1}{2}} \left( \boldsymbol{y}_1 - \hat{\boldsymbol{y}}_1 \right), \tag{40c}$$

$$\mathbf{S}_1 := \mathbf{R}_1^{-\frac{1}{2}} \mathbf{H}_1 \mathbf{X}_1^{\mathrm{fore}}, \tag{40d}$$





where $\boldsymbol{w}$ is now a weight vector in $\mathbb{R}^{N_e}$. The ensemble-based cost function is then written as

$$\widetilde{\mathcal{J}}(\boldsymbol{w}) = \frac{1}{2} \parallel \hat{\boldsymbol{x}}_1^{\text{fore}} - \mathbf{X}_1^{\text{fore}}\boldsymbol{w} - \hat{\boldsymbol{x}}_1^{\text{fore}} \parallel_{\mathbf{P}_1^{\text{fore}}}^2 + \frac{1}{2} \parallel \boldsymbol{y}_1 - \mathbf{H}_1\hat{\boldsymbol{x}}_1^{\text{fore}} - \mathbf{H}_1\mathbf{X}_1^{\text{fore}}\boldsymbol{w} \parallel_{\mathbf{R}_1}^2 \tag{41a}$$

$$= \frac{1}{2}(N_e - 1) \parallel \boldsymbol{w} \parallel^2 + \frac{1}{2} \parallel \hat{\boldsymbol{\delta}}_1 - \mathbf{S}_1\boldsymbol{w} \parallel^2 . \tag{41b}$$

Define $\hat{\boldsymbol{w}}$ to be the minimizer of the cost function in Eq. (41). Hunt et al. (2007) demonstrate that, up to a gauge transformation, $\hat{\boldsymbol{w}}$ yields the minimizer of the state-space cost function, Eq. (21), when the estimate is restricted to the ensemble span. Equation (41) is quadratic in $\boldsymbol{w}$ and can be solved similarly to Eq. (26) to render

$$\hat{\boldsymbol{w}} := \boldsymbol{0} - \widetilde{\mathbf{H}}_{\widetilde{\mathcal{J}}}^{-1}\nabla\widetilde{\mathcal{J}}|_{\boldsymbol{w}=\boldsymbol{0}}, \tag{42a}$$

$$\mathbf{T} := \widetilde{\mathbf{H}}_{\widetilde{\mathcal{J}}}^{-\frac{1}{2}}, \tag{42b}$$

$$\mathbf{P}_1^{\text{filt}} = \left(\mathbf{X}_1^{\text{fore}}\mathbf{T}\right)\left(\mathbf{X}_1^{\text{fore}}\mathbf{T}\right)^\top / (N_e - 1). \tag{42c}$$

The ensemble transform Kalman filter (**ETKF**) equations are then given by

$$\mathbf{E}_1^{\text{filt}} = \hat{\boldsymbol{x}}_1^{\text{fore}}\mathbf{1}^\top + \mathbf{X}_1^{\text{fore}}\left(\hat{\boldsymbol{w}}\mathbf{1}^\top + \sqrt{N_e - 1}\mathbf{T}\mathbf{U}\right) \tag{43}$$

where $\mathbf{U} \in \mathbb{R}^{N_e \times N_e}$ can be any mean-preserving, orthogonal transformation, i.e., $\mathbf{U}\mathbf{1} = \mathbf{1}$. The simple choice of $\mathbf{U} := \mathbf{I}_{N_e}$ is sufficient, but it has been demonstrated that choosing a random, mean-preserving orthogonal transformation at each analysis as above can improve the stability of the ETKF, reducing the collapse of the variances to a few modes in the empirical covariance estimate (Sakov and Oke, 2008b). We remark that Eq. (43) can be written equivalently as a single right ensemble transformation,

$$\mathbf{E}_1^{\text{filt}} = \mathbf{E}_1^{\text{fore}}\boldsymbol{\Psi}_1, \tag{44a}$$

$$\boldsymbol{\Psi}_1 := \mathbf{1}\mathbf{1}^\top / N_e + \left(\mathbf{I}_{N_e} - \mathbf{1}\mathbf{1}^\top / N_e\right)\left(\hat{\boldsymbol{w}}\mathbf{1}^\top + \sqrt{N_e - 1}\mathbf{T}\mathbf{U}\right). \tag{44b}$$

The compact update notation in Eq. (44) is used to simplify analysis.

If the observation operator $\mathcal{H}_1$ is actually nonlinear, then the ETKF typically uses the following approximation to the quadratic cost function,

$$\mathbf{Y}_1 := \mathcal{H}_1\left(\mathbf{E}_1^{\text{fore}}\right), \tag{45a}$$

$$\hat{\boldsymbol{y}}_1 := \mathbf{Y}_1\mathbf{1}/N_e, \tag{45b}$$

$$\mathbf{S}_1 := \mathbf{R}_1^{-\frac{1}{2}}\mathbf{Y}_1 - \hat{\boldsymbol{y}}_1\mathbf{1}^\top. \tag{45c}$$

Substituting the definitions in Eq. (45) for the definitions in Eq. (40) gives the standard nonlinear analysis in the ETKF. Note that this framework extends to a fully iterative analysis of nonlinear observation operators, discussed in Section 4.1. Multiplicative covariance inflation is often used in the ETKF to handle the systematic underestimation of the forecast and filter covariance due to the sample error implied by a finite-size ensemble and nonlinearity of the forecast model $\mathcal{M}_1$ (Raanes et al., 2019a).





The standard ETKF cycle is summarized in Algorithm 5. This algorithm is broken into the sub-routines in Algorithms 1 - 4 which are re-used throughout our analysis, which emphasize the commonality and the differences of the studied smoother schemes. The filter analysis described above can be extended in several different ways when producing a smoother analysis on a DAW including lagged, past states, depending in part whether it is formulated as a marginal or a joint smoother (Cosme et al., 2012). The way in which this analysis is extended, utilizing a retrospective re-analysis or a 4D-MAP cost function, differentiates the EnKS from the IEnKS, and highlights the ways in which the SIEnKS differs from these other schemes.

### 3.2 The fixed-lag EnKS

The (right-transform) fixed-lag EnKS extends the filter analysis in the ETKF over the smoothing DAW by sequentially re-analyzing past states with future observations. This analysis is performed retrospectively in the sense that the filter cycle of the ETKF is left unchanged, while an additional inner-loop of the DA cycle performs an update on the estimated lagged state ensembles within the DAW, stored in memory. Assume $S = 1 \leq L$, the EnKS estimates the joint posterior density $p(\boldsymbol{x}_{L:1}|\boldsymbol{y}_{L:1})$ recursively, given the joint posterior estimate over the last DAW $p(\boldsymbol{x}_{L-1:0}|\boldsymbol{y}_{L-1:0})$. We begin by considering the filter problem as in Eq. (19).

Given $p(\boldsymbol{x}_{L-1:0}, \boldsymbol{y}_{L-1:0})$, we write the filter density up to proportionality

$$p(\boldsymbol{x}_L|\boldsymbol{y}_{L:0}) \propto p(\boldsymbol{y}_L|\boldsymbol{x}_L, \boldsymbol{y}_{L-1:0})p(\boldsymbol{x}_L, \boldsymbol{y}_{L-1:0}) \tag{46a}$$

$$\propto \underbrace{p(\boldsymbol{y}_L|\boldsymbol{x}_L)}_{(i)} \underbrace{\int p(\boldsymbol{x}_L|\boldsymbol{x}_{L-1})p(\boldsymbol{x}_{L-1:0}|\boldsymbol{y}_{L-1:0})\mathrm{d}\boldsymbol{x}_{L-1:0}}_{(ii)}, \tag{46b}$$

as the product of (i) the likelihood of the observation $\boldsymbol{y}_L$ given $\boldsymbol{x}_L$; and (ii) the forecast for $\boldsymbol{x}_L$ using the transition kernel on the last joint posterior estimate, marginalizing out $\boldsymbol{x}_{L-1:0}$. Recalling that $p(\boldsymbol{x}_L|\boldsymbol{y}_{L:1}) \propto p(\boldsymbol{x}_L|\boldsymbol{y}_{L:0})$, this provides a means to sample the filter marginal of the desired joint posterior. The usual ETKF filter analysis is performed to sample the filter distribution at time $t_L$, yet, to complete the smoothing cycle, the scheme must sample the joint posterior density $p(\boldsymbol{x}_{L:1}, \boldsymbol{y}_{L:1})$.

Consider that the marginal smoother density is proportional to

$$p(\boldsymbol{x}_{L-1}|\boldsymbol{y}_{L:0}) \propto p(\boldsymbol{y}_L|\boldsymbol{x}_{L-1}, \boldsymbol{y}_{L-1:0})p(\boldsymbol{x}_{L-1}, \boldsymbol{y}_{L-1:0}) \tag{47a}$$

$$\propto \underbrace{p(\boldsymbol{y}_L|\boldsymbol{x}_{L-1})}_{(i)} \underbrace{p(\boldsymbol{x}_{L-1}|\boldsymbol{y}_{L-1:0})}_{(ii)}, \tag{47b}$$

where: (i) is the likelihood of the observation $\boldsymbol{y}_L$ given the past state $\boldsymbol{x}_{L-1}$; (ii) is the marginal density for $\boldsymbol{x}_{L-1}$ from the last joint posterior.

Assume now the perfect, linear-Gaussian model – the corresponding Bayesian MAP cost function is given as

$$\mathcal{J}(\boldsymbol{x}_{L-1}) = \frac{1}{2} \parallel \boldsymbol{x}_{L-1} - \overline{\boldsymbol{x}}_{L-1|L-1}^{\mathrm{smth}} \parallel_{\mathbf{B}_{L-1|L-1}^{\mathrm{smth}}}^2 + \frac{1}{2} \parallel \boldsymbol{y}_L - \mathbf{H}_L\mathbf{M}_L\boldsymbol{x}_{L-1} \parallel_{\mathbf{R}_L}^2 \tag{48}$$





where $\overline{\boldsymbol{x}}^{\text{smth}}_{L-1|L-1}$ and $\mathbf{B}^{\text{smth}}_{L-1|L-1}$ are the mean and covariance of the marginal smoother density $p(\boldsymbol{x}_{L-1}|\boldsymbol{y}_{L-1:0})$. Take the matrix decomposition

$$\mathbf{B}^{\text{smth}}_{L-1|L-1} = \boldsymbol{\Sigma}^{\text{smth}}_{L-1|L-1} \left( \boldsymbol{\Sigma}^{\text{smth}}_{L-1|L-1} \right)^{\top}, \tag{49}$$

and write $\boldsymbol{x}_{L-1} = \overline{\boldsymbol{x}}^{\text{smth}}_{L-1|L-1} + \boldsymbol{\Sigma}^{\text{smth}}_{L-1|L-1}\boldsymbol{w}$, rendering the cost function as

$$\mathcal{J}(\boldsymbol{w}) = \frac{1}{2} \| \boldsymbol{w} \|^2 + \frac{1}{2} \| \boldsymbol{y}_L - \mathbf{H}_L\mathbf{M}_L(\overline{\boldsymbol{x}}^{\text{smth}}_{L-1|L-1} + \boldsymbol{\Sigma}^{\text{smth}}_{L-1|L-1}\boldsymbol{w}) \|^2_{\mathbf{R}_L} \tag{50a}$$

$$= \frac{1}{2} \| \boldsymbol{w} \|^2 + \frac{1}{2} \| \boldsymbol{y}_L - \mathbf{H}_L\overline{\boldsymbol{x}}^{\text{fore}}_L - \mathbf{H}_L\boldsymbol{\Sigma}^{\text{fore}}_L\boldsymbol{w} \|^2_{\mathbf{R}_L} \tag{50b}$$

$$= \frac{1}{2} \| \boldsymbol{w} \|^2 + \frac{1}{2} \| \overline{\boldsymbol{\delta}}_L - \boldsymbol{\Gamma}_L\boldsymbol{w} \|^2 . \tag{50c}$$

Let $\overline{\boldsymbol{w}}$ now denote the minimizer of Eq. (50). It is important to recognize that for

$$\boldsymbol{x}_L := \mathbf{M}_L \left( \overline{\boldsymbol{x}}^{\text{smth}}_{L-1|L-1} + \boldsymbol{\Sigma}^{\text{smth}}_{L-1|L-1}\boldsymbol{w} \right) \tag{51}$$

$$= \overline{\boldsymbol{x}}^{\text{fore}}_L + \boldsymbol{\Sigma}^{\text{fore}}_L\boldsymbol{w}, \tag{52}$$

such that the optimal weight vector for the smoothing problem $\overline{\boldsymbol{w}}$ is also the optimal weight vector for the filter problem.

The ensemble-based approximation,

$$\boldsymbol{x}_{L-1} = \hat{\boldsymbol{x}}^{\text{smth}}_{L-1|L-1} + \mathbf{X}^{\text{smth}}_{L-1|L-1}\boldsymbol{w}, \tag{53a}$$

$$\widetilde{\mathcal{J}}(\boldsymbol{w}) = \frac{1}{2}(N_e - 1) \| \boldsymbol{w} \|^2 + \frac{1}{2} \| \hat{\boldsymbol{\delta}}_L - \mathbf{S}_L\boldsymbol{w} \|^2, \tag{53b}$$

to the exact smoother cost function in Eq. (50) yields the retrospective analysis of the EnKS as

$$\hat{\boldsymbol{w}} := \mathbf{0} - \widetilde{\mathbf{H}}^{-1}_{\widetilde{\mathcal{J}}}\nabla\widetilde{\mathcal{J}}|_{\boldsymbol{w}=\mathbf{0}}, \tag{54a}$$

$$\mathbf{T} := \widetilde{\mathbf{H}}^{-\frac{1}{2}}_{\widetilde{\mathcal{J}}}, \tag{54b}$$

$$\mathbf{E}^{\text{smth}}_{L-1|L} = \hat{\boldsymbol{x}}^{\text{smth}}_{L-1|L-1}\mathbf{1}^{\top} + \mathbf{X}^{\text{smth}}_{L-1|L-1} \left( \hat{\boldsymbol{w}}\mathbf{1}^{\top} + \sqrt{N_e - 1}\mathbf{T}\mathbf{U} \right),$$

$$\equiv \mathbf{E}^{\text{smth}}_{L-1|L}\boldsymbol{\Psi}_L. \tag{54c}$$

The above equations can be generalized for arbitrary indices $k|L$ over the DAW, providing the complete description of the inner-loop between each filter cycle of the EnKS. After each new observation is assimilated with the ETKF analysis step, a smoother inner-loop makes a backward pass over the DAW applying the transform and the weights of the ETKF filter update to each past ensemble state stored in memory. This analysis is easily generalized to the case where there is a shift of the DAW with $S > 1$, though the EnKS does not process observations asynchronously by default. This means that the ETKF filter steps, and the subsequent retrospective re-analysis, must be performed in sequence over the observations, ordered in time, rather than making a global analysis over $\boldsymbol{y}_{L-S+1:L}$. A standard form of the EnKS is summarized in Algorithm 6, utilizing the sub-routines in Algorithms 1 - 4.





A schematic of the EnKS cycle for a lag of $L = 4$ and a shift of $S = 1$ is pictured in Fig. 2. Time moves forward from left to right in the horizontal axis with a step size of $\Delta t$. At each analysis time, the ensemble forecast from the last filter density is combined with the observation to produce the ensemble transform update. This transform is then utilized to produce the posterior estimate for all lagged ensemble states, conditioned on the new observation. The information in the posterior estimate thus flows in reverse time to the lagged states stored in memory, but the information flow is unidirectional in this scheme. It is understood then that re-initializing the improved posterior estimate for the lagged states in the dynamical model does not improve the filter estimate in the perfect, linear-Gaussian configuration. Indeed, define the product of the ensemble transforms

$$\boldsymbol{\Psi}_{k:l} := \boldsymbol{\Psi}_k \cdots \boldsymbol{\Psi}_l. \tag{55}$$

Then, for arbitrary $1 \leq k \leq l \leq L$,

$$\mathbf{M}_{l:k} \mathbf{E}_{k-1|k-1}^{\mathrm{smth}} \boldsymbol{\Psi}_{k:l} = \mathbf{M}_{l:k} \mathbf{E}_{k-1|l}^{\mathrm{smth}} \tag{56a}$$
$$= \mathbf{E}_{l|k-1}^{\mathrm{fore}} \boldsymbol{\Psi}_{k:l} \tag{56b}$$
$$= \mathbf{E}_{l|l}^{\mathrm{smth}}. \tag{56c}$$

This demonstrates that conditioning on the information from the observation is covariant with the dynamics. Raanes (2016) demonstrates the equivalence of the EnKS and the Rauch-Tung-Striebel (RTS) smoother where this property of perfect, linear-Gaussian models is well understood. In the RTS formulation of the retrospective re-analysis, the conditional estimate reduces to the map of the current filter estimate under the reverse time model $\mathbf{M}_k^{-1}$ (Jazwinski, 1970, see example 7.8, chapter 7). Note, however, that both of the EnKS and ensemble RTS smoothers produce their retrospective re-analyses via a recursive ensemble transform, without the need to make backward model simulations.

The covariance of conditioning on observations and the model dynamics does not hold, however, either in the case of non-linear dynamics or model error. Re-initializing the DA cycle in a perfect, nonlinear model with the conditional ensemble estimate $\mathbf{E}_{0|L}^{\mathrm{smth}}$ can dramatically improve the accuracy of the subsequent forecast and filter statistics. Particularly, this exploits the miss-match in perfect, nonlinear dynamics between $\mathcal{M}_{L:1}\left(\mathbf{E}_{0|L}^{\mathrm{smth}}\right) \neq \mathbf{E}_L^{\mathrm{filt}}$. Chaotic dynamics generates additional information about the initial value problem in the sense that initial conditions nearby to each other are distinguished by their subsequent evolution and divergence due to dynamical instability. Re-initializing the model forecast with the smoothed prior estimate brings new information into the forecast for states in the next DAW. This improvement in the accuracy of the ensemble statistics has been exploited to a great extent by utilizing the 4D-MAP ensemble cost function (Hunt et al., 2004). Particularly, the filter MAP cost function can be extended over multiple observations simultaneously, and in terms of lagged states directly. This alternative approach to extending the filter MAP analysis to the smoother MAP analysis is discussed in the following.

### 3.3 The Gauss-Newton, fixed-lag IEnKS

The following is an up-to-date reformulation of the Gauss-Newton IEnKS of Bocquet and Sakov (2013, 2014), and its derivations. Instead of considering the marginal smoother problem, now consider the joint posterior density directly and for a general



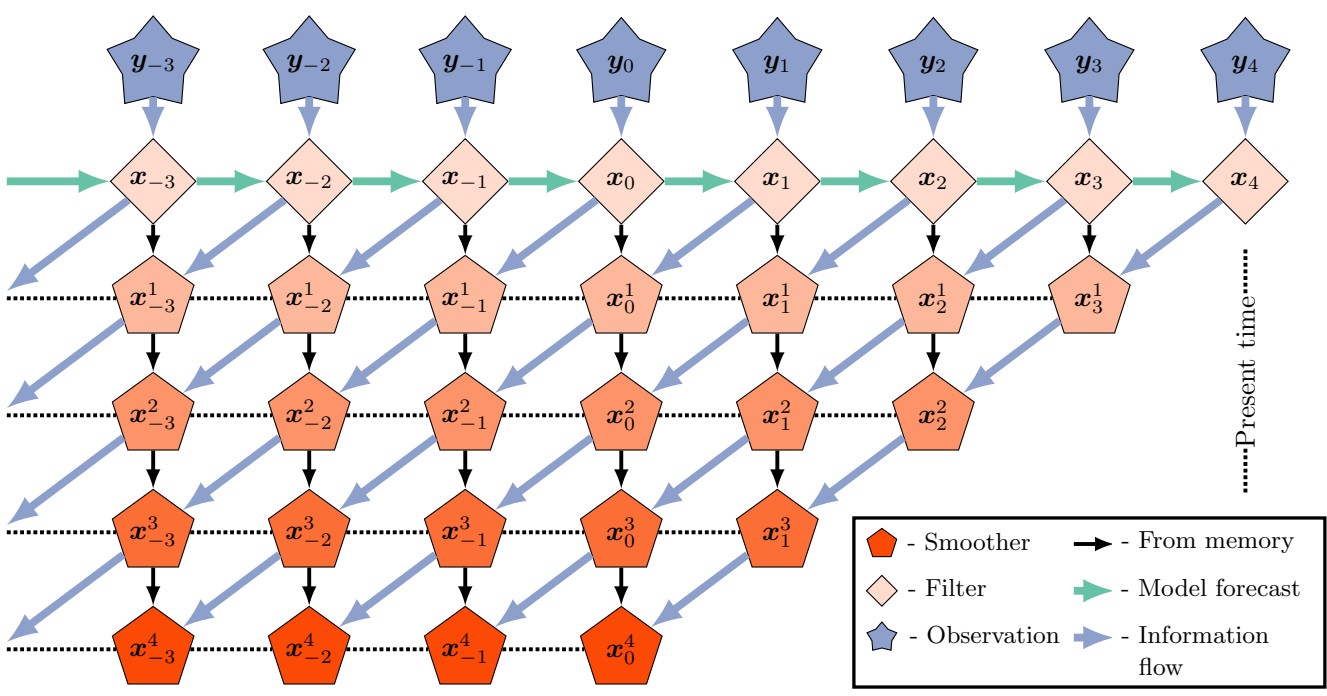

**Figure 2.** Lag$= 4$, shift$= 1$ EnKS. Observations are assimilated sequentially via the filter cost function and a retrospective re-analysis is applied to all ensemble states within the lag window stored in memory. Adapted from Asch et al. (2016).

shift $S$. For a general shift, the last posterior density is written as $p(\boldsymbol{x}_{L-S:1-S}|\boldsymbol{y}_{L-S:1-S})$. Using the independence of observation errors and the Markov assumption recursively,

$$p(\boldsymbol{x}_{L:1}|\boldsymbol{y}_{L:1-S}) \propto \int \mathrm{d}\boldsymbol{x}_0 \left[ \prod_{k=L-S+1}^{L} p(\boldsymbol{y}_k|\boldsymbol{x}_k) p(\boldsymbol{x}_k|\boldsymbol{x}_{k-1}) \right] \left[ \prod_{k=1}^{L-S} p(\boldsymbol{x}_k|\boldsymbol{x}_{k-1}) \right] p(\boldsymbol{x}_0|\boldsymbol{y}_{L-S:1-S}). \tag{57}$$

Additionally, using the perfect model assumption,

$\quad p(\boldsymbol{x}_k|\boldsymbol{x}_{k-1}) = \delta\{\boldsymbol{x}_k - \mathcal{M}_k(\boldsymbol{x}_{k-1})\}$ (58)

for every $k$. Therefore,

$$p(\boldsymbol{x}_{L:1}|\boldsymbol{y}_{L:1-S}) \propto \int \mathrm{d}\boldsymbol{x}_0 \underbrace{p(\boldsymbol{x}_0|\boldsymbol{y}_{L-S:1-S})}_{(i)} \underbrace{\left[ \prod_{k=L-S+1}^{L} p(\boldsymbol{y}_k|\boldsymbol{x}_k) \right]}_{(ii)} \underbrace{\left[ \prod_{k=1}^{L} \delta\{\boldsymbol{x}_k - \mathcal{M}_k(\boldsymbol{x}_{k-1})\} \right]}_{(iii)} \tag{59}$$

where term (i) in Eq. (59) represents the marginal smoother density for $\boldsymbol{x}_{0|L-S}$ over the last DAW; term (ii) represents the joint likelihood of the observations given the model state; and term (iii) represents the free forecast of the smoother estimate

for $\boldsymbol{x}_{0|L-S}$. Noting that $p(\boldsymbol{x}_{L:1}|\boldsymbol{y}_{L:1}) \propto p(\boldsymbol{x}_{L:1}|\boldsymbol{y}_{L:1-S})$, this provides a recursive form to sample the joint posterior density.





Under the perfect, linear-Gaussian model assumption, the above derivation leads to the following exact 4D-MAP cost function

$$\mathcal{J}(\boldsymbol{x}_0) := \frac{1}{2} \parallel \boldsymbol{x}_0 - \overline{\boldsymbol{x}}_{0|L-S}^{\text{smth}} \parallel_{\mathbf{B}_{0|L-S}^{\text{smth}}}^2 + \frac{1}{2} \sum_{k=L-S+1}^{L} \parallel \boldsymbol{y}_k - \mathbf{H}_k \mathbf{M}_{k:1} \boldsymbol{x}_0 \parallel_{\mathbf{R}_k}^2 . \tag{60}$$

The ensemble-based approximation, using notations as in Eq. (40), yields

$$\boldsymbol{x}_0 := \hat{\boldsymbol{x}}_{0|L-S}^{\text{smth}} + \mathbf{X}_{0|L-S}^{\text{smth}} \boldsymbol{w}, \tag{61a}$$

$$\widetilde{\mathcal{J}}(\boldsymbol{w}) := \frac{1}{2}(N_e - 1) \parallel \boldsymbol{w} \parallel^2 + \frac{1}{2} \sum_{k=L-S+1}^{L} \parallel \hat{\boldsymbol{\delta}}_k - \mathbf{S}_k \boldsymbol{w} \parallel^2 . \tag{61b}$$

Notice that Eq. (61b) is quadratic in $\boldsymbol{w}$; therefore, for the perfect, linear-Gaussian model, one can perform a global analysis at once over all new observations in the DAW.

The gradient and the Hessian of the ensemble-based 4D-MAP cost function are given as

$$\nabla \widetilde{\mathcal{J}} := (N_e - 1)\boldsymbol{w} - \sum_{k=L-S+1}^{L} \mathbf{S}_k^\top \left( \hat{\boldsymbol{\delta}}_k - \mathbf{S}_k \boldsymbol{w} \right), \tag{62a}$$

$$\widetilde{\mathbf{H}}_{\widetilde{\mathcal{J}}} := (N_e - 1)\mathbf{I}_{N_e} + \sum_{k=L-S+1}^{L} \mathbf{S}_k^\top \mathbf{S}_k, \tag{62b}$$

so that evaluating at $\boldsymbol{w} = \boldsymbol{0}$, the minimizer $\hat{\boldsymbol{w}}$ is again given by a single iteration of Newton's descent

$$\hat{\boldsymbol{w}} := \boldsymbol{0} - \widetilde{\mathbf{H}}_{\widetilde{\mathcal{J}}} \nabla \widetilde{\mathcal{J}}|_{\boldsymbol{w}=\boldsymbol{0}}. \tag{63}$$

Define the covariance transform again as $\mathbf{T} := \widetilde{\mathbf{H}}_{\widetilde{\mathcal{J}}}^{-\frac{1}{2}}$. We denote the right ensemble transform corresponding to the 4D-MAP analysis $\boldsymbol{\Psi}_{L-S+1:L}^{\text{4D}}$ to distinguish from the product of the sequential filter transforms $\boldsymbol{\Psi}_{L-S+1:L}$. The global analyses are defined such that

$$\boldsymbol{\Psi}_{L-S+1:L}^{\text{4D}} := \left\{ \mathbf{1}\mathbf{1}^\top/N_e + \left( \mathbf{I}_{N_e} - \mathbf{1}\mathbf{1}^\top/N_e \right) \left( \hat{\boldsymbol{w}}\mathbf{1}^\top + \sqrt{N_e - 1}\mathbf{T}\mathbf{U} \right) \right\}, \tag{64a}$$

$$\mathbf{E}_{0|L}^{\text{smth}} = \mathbf{E}_{0|L-S}^{\text{smth}} \boldsymbol{\Psi}_{L-S+1:L}^{\text{4D}}, \tag{64b}$$

where $\mathbf{U}$ is any mean-preserving, orthogonal matrix.

In the perfect, linear-Gaussian model, this formulation of the IEnKS is actually equivalent to the 4D-EnKF of Hunt et al. (2004); Fertig et al. (2007); and Harlim and Hunt (2007). The above scheme produces a global analysis of all observations within the DAW, even asynchronously from the standard filter cycle (Sakov et al., 2010). Particularly, one can generate a free ensemble forecast with initial conditions drawn iid as $p(\boldsymbol{x}_0|\boldsymbol{y}_{L-S:1-S})$; subsequently, all data available within the DAW is used to estimate the update to the initial ensemble. The perfect model assumption means that the updated initial ensemble $\mathbf{E}_{0|L}^{\text{smth}}$ can then be used to temporally interpolate the joint posterior estimate over the entire DAW from the marginal sample, i.e., for any $0 < k \le L$

$$\mathbf{M}_{k:1}\mathbf{E}_{0|L-S}^{\text{smth}}\boldsymbol{\Psi}_{L-S+1:L}^{\text{4D}} \equiv \mathbf{E}_{k|L}^{\text{smth}}. \tag{65}$$



When $\mathcal{M}_k$ and $\mathcal{H}_k$ are nonlinear, the IEnKS formulation is extended with additional iterations of Newton's descent as in Eq. (31) in order to iteratively optimize the update weights. Specifically, the gradient is given by

$$420 \quad \nabla\widetilde{\mathcal{J}} := (N_e - 1)\boldsymbol{w} - \sum_{k=L-S+1}^{L} \widetilde{\mathbf{Y}}_k^\top \mathbf{R}_k^{-1} \left[ \boldsymbol{y}_k - \mathcal{H}_k \circ \mathcal{M}_{k:1} \left( \hat{\boldsymbol{x}}_{0|L-S}^{\text{smth}} + \mathbf{X}_{0|L-S}^{\text{smth}} \boldsymbol{w} \right) \right], \qquad (66)$$

where $\widetilde{\mathbf{Y}}_k$ represents a directional derivative of the observation and state models with respect to the ensemble perturbations at the ensemble mean,

$$\widetilde{\mathbf{Y}}_k := \nabla|_{\hat{\boldsymbol{x}}_{0|L-S}^{\text{smth}}} \left[ \mathcal{H}_k \circ \mathcal{M}_{k:1} \right] \mathbf{X}_{0|L-S}^{\text{smth}}; \qquad (67)$$

this describes the sensitivities of the cost function with respect to the ensemble perturbations, mapped to the observation space. When the dynamics are weakly nonlinear, the ensemble perturbations of the EnKS and IEnKS are known to closely align with the span of the backward Lyapunov vectors of the nonlinear model along the true state trajectory (Bocquet and Carrassi, 2017); under these conditions, Eq. (67) can be interpreted as a directional derivative with respect to the forecast error growth along the dynamical instabilities of the nonlinear model, see Carrassi et al. (2021) and references therein.

In order to avoid an explicit computation of the tangent-linear model and the adjoint as in 4D-VAR, Sakov et al. (2012) and Bocquet and Sakov (2012) proposed two formulations to approximate the tangent-linear propagation of the ensemble perturbations. The "bundle" scheme makes an explicit approximation of finite differences in the observation space where, for an arbitrary ensemble, they define the approximate linearization

$$\mathbf{Y}_k := \frac{1}{\epsilon} \mathcal{H}_k \circ \mathcal{M}_{k:1} \left( \boldsymbol{x}_0 \mathbf{1}^\top + \epsilon \mathbf{X}_0 \right) \left( \mathbf{I}_{N_e} - \mathbf{1}\mathbf{1}^\top / N_e \right), \qquad (68)$$

for a small constant $\epsilon$. Alternatively the "transform" version provides a different approximation of the variational analysis, using the covariance transform $\mathbf{T}$ and its inverse as a pre- / post-conditioning of the perturbations used in the sensitivies approximation. The transform variant of the IEnKS is in some cases more numerically efficient than the bundle version, requiring fewer ensemble simulations, and it is explicitly related to the ETKF / EnKS / 4D-ETKF formalism presented thus far. For these reasons, the transform approximation is used as a basis of comparison with the other schemes in this work.

For the IEnKS transform variant, the following ensemble-based approximations are re-defined in each Newton iteration

$$440 \quad \mathbf{Y}_k := \mathcal{H}_k \left( \mathbf{E}_k \right), \qquad (69a)$$

$$\hat{\boldsymbol{y}}_k := \mathbf{Y}_k \mathbf{1}/N_e, \qquad (69b)$$

$$\mathbf{S}_k := \mathbf{R}_k^{-\frac{1}{2}} \left( \mathbf{Y}_k - \hat{\boldsymbol{y}}_k \mathbf{1}^\top \right) \mathbf{T}^{-1}, \qquad (69c)$$

$$\hat{\boldsymbol{\delta}} := \mathbf{R}_k^{-\frac{1}{2}} \left( \boldsymbol{y}_k - \hat{\boldsymbol{y}}_k \right), \qquad (69d)$$

where the first covariance transform is defined as $\mathbf{T} := \mathbf{I}_{N_e}$, the gradient and Hessian are computed as in Eq. (62) from the above and where the covariance transform is re-defined in terms of the Hessian, $\mathbf{T} := \widetilde{\mathbf{H}}_{\widetilde{\mathcal{J}}}^{-\frac{1}{2}}$, at the end of each iteration. With these definitions, the first iteration of the IEnKS transform variant corresponds to the solution of the nonlinear 4D-ETKF, but



subsequent iterates are initialized by pre-conditioning the initial ensemble perturbations via the update $\mathbf{T}$ and post-conditioning the sensitivities by the inverse transform $\mathbf{T}^{-1}$.

A revised and simplified form of the Gauss-Newton IEnKS, transform variant is presented for the first time in Algorithm 7. Note, while Algorithm 7 does not explicitly reference the sub-routine in Algorithm 1, many of the same steps are used in the inner-loop of the IEnKS when computing the sensitivities. It is important to notice that, for $S > 1$, the IEnKS only requires a single computation of the square root inverse of the Hessian of the 4D-MAP cost function, per iteration of the optimization, to process all observations in the DAW. On the other hand, the EnKS processes these observations sequentially, requiring $S$ total square root inverse calculations of the Hessian, corresponding to each of the sequential filter cost functions.

The IEnKS is computationally constrained by the fact that each iteration of the descent requires $L$ total ensemble simulations in the dynamical model $\mathcal{M}_k$. One can minimize this expense by using a single iteration of the IEnKS equations, which is what is denoted the "linearized" IEnKS (Lin-IEnKS) (Bocquet and Sakov, 2014). When the overall DA cycle is nonlinear, but only weakly nonlinear, this single iteration of the IEnKS algorithm can produce a dramatic improvement in the forecast accuracy versus the forecast / filter cycle of the EnKS. However, the overall nonlinearity of the DA cycle may be strongly influenced by other factors than the model forecast $\mathcal{M}_k$ itself. As as simple example, we may consider the case in which $\mathcal{H}_k$ is nonlinear yet $\mathcal{M}_k \equiv \mathbf{M}_k$ for all $k$. In this setting, it may be more numerically efficient to iterate upon the filter cost function rather than the full 4D-MAP cost function, which uses simulations of the dynamical model. Combining: (i) the filter step and retrospective re-analysis of the EnKS; and (ii) the single iteration of the ensemble simulation over the DAW as in the 4D-ETKF / Lin-IEnKS; we obtain an estimation scheme that sequentially solves nonlinear filter cost functions in the current DAW, while making an improved forecast in the next by transmitting the retrospective analyses through the dynamics via the updated initial ensemble. This single-iteration ensemble transform Kalman smoother (SIEnKS) is formalized in the following section.

## 3.4 The fixed-lag SIEnKS

### 3.4.1 Algorithm

Recall that, from Eq. (56), conditioning the ensemble with the right transform $\mathbf{\Psi}_k$ is covariant with the dynamcis. In a perfect, linear-Gaussian model, we can therefore estimate the joint posterior over the DAW via the model propagation of the marginal for $\boldsymbol{x}_{0|L}^{\mathrm{smth}}$ as in the IEnKS, but using the EnKS retrospective re-analysis to generate the initial condition. For arbitrary $1 \le S \le L$, define each of the right transforms $\{\mathbf{\Psi}_k\}_{k=L-S+1}^{L}$ as in the sequential filter analysis of the ETKF with Eq. (44). Rather than storing the ensemble matrix in memory for each time $t_k$ in the DAW, we instead store $\mathbf{E}_{0|L-S}^{\mathrm{smth}}$ and $\mathbf{E}_{L-S|L-S}^{\mathrm{smth}}$ to begin a DA cycle. Observations within the DAW are sequentially assimilated via the filter cycle initialized with $\mathbf{E}_{L-S|L-S}^{\mathrm{smth}}$, and a marginal smoother analysis is performed on $\mathbf{E}_{0|L-S}^{\mathrm{smth}}$ sequentially with these transforms. The joint posterior is interpolated over the DAW for any $1 \le k \le L$ via the model dynamics as

$$\mathbf{E}_{0|L}^{\mathrm{smth}} = \mathbf{E}_{0|L-S}^{\mathrm{smth}} \mathbf{\Psi}_{L-S+1:L}, \tag{70a}$$
$$\mathbf{E}_{k|L}^{\mathrm{smth}} := \mathcal{M}_{k:1}\left(\mathbf{E}_{0|L}^{\mathrm{smth}}\right). \tag{70b}$$



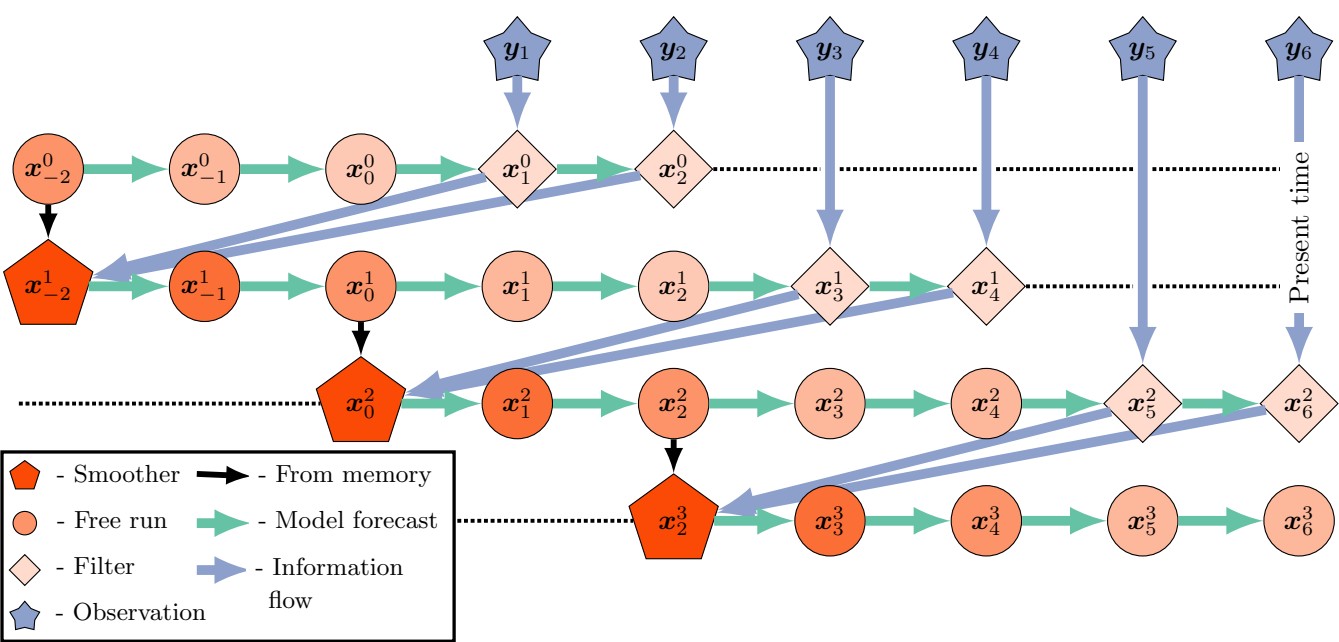

**Figure 3.** Lag= 4, shift= 2 SIEnKS diagram. An initial condition from the last smoothing cycle initializes a forecast simulation over the current DAW of $L = 4$ states. New observations entering the DAW are assimilated sequentially via the filter cost function. After each filter analysis, a retrospective re-analysis is applied to the initial ensemble, and this re-analyzed initial condition is evolved via the model $S$ analysis times forward to begin the next cycle.

Notice that for $S = 1$,

$$\boldsymbol{\Psi}_{L-S+1:L} \equiv \boldsymbol{\Psi}_{L-S+1:L}^{\mathrm{4D}} \tag{71}$$

so that in the perfect, linear-Gaussian model with $S = 1$ the SIEnKS and the Lin-IEnKS coincide. The SIEnKS and the Lin-IEnKS have very different treatments of nonlinearity in the DA cycle, but even in the perfect, linear-Gaussian model, a shift $S > 1$ distinguishes these algorithms. A schematic of the SIEnKS cycle for a lag of $L = 4$ and a shift of $S = 2$ is pictured in Fig. 3. This demonstrates how the sequential analysis of the filter cost function and sequential, retrospective re-analysis for each incoming observation differs from the global analysis of the (Lin-)IEnKS. A generic form of the SIEnKS is summarized in Algorithm 8, utilizing the sub-routines in Algorithms 1 - 4. Note that the version presented in Algorithm 8 is used to emphasize the commonality with the EnKS filter cycle. However, an equivalent outer-loop can be implemented to initialize each cycle with $\mathbf{E}_{0|L-S}^{\mathrm{smth}}$ alone, similar to the IEnKS. Such an outer-loop is utilized when we derive the SIEnKS MDA scheme in Algorithm 12 from the IEnKS MDA scheme in Algorithm 13.





### 3.4.2 Comparison with other schemes

Other well-known DA schemes combining a retrospective re-analysis and re-initialization of the ensemble forecast include the Running In Place (RIP) smoother of Kalnay and Yang (2010) and the One Step Ahead (OSA) smoother of Desbouvries et al. (2011). The RIP smoother iterates over the ensemble simulation and filter cost function, both, in order to apply a retrospective re-analysis to the first prior with a lag and shift of $L = S = 1$. The RIP smoother is similarly proposed as an outer-loop optimization of the EnKS in order to spin up the LETKF from a "cold start" of a forecast model and DA cycle (Yang et al., 2013). The OSA smoother is also proposed as an outer-loop optimization of the DA cycle, and integrates an EnKF framework, including for a two-stage, iterative optimization of dynamical forecast model parameters within the DA cycle (Gharamti et al., 2015; Ait-El-Fquih et al., 2016; Raboudi et al., 2018). The OSA smoother uses a single iteration and a lag and shift of $L = S = 1$, making a filter analysis of the incoming observation and a retrospective re-analysis of the prior. However, the OSA smoother differs from the SIEnKS in using an additional filter analysis while interpolating the joint posterior over the DAW, accounting for model error in the simulation of $\mathcal{M}_1 \left( \mathbf{E}_{0|1}^{\mathrm{smth}} \right)$. Without model error, the second filter analysis in the OSA smoother simulation is eliminated from the estimation scheme. Therefore, with an ETKF style filter analysis, a single iteration of the ensemble over the DAW, a perfect model assumption and a lag of $L = S = 1$, the SIEnKS, RIP and OSA smoothers all coincide.

The rationale of the SIEnKS is to focus computation on an iterative, ensemble-variational optimization of the nonlinear filter cost function in DA cycles for which the forecast error dynamics are weakly nonlinear. The SIEnKS, in particular, generalizes some of the ideas in these other DA schemes for weakly nonlinear forecast error dynamics, including for the application of: (i) arbitrary lags $L \geq 1$ and shifts $S \leq L$; (ii) the iterative optimization of the nonlinear filter cost function, due to hyper-parameters of the estimation scheme; (iii) multiple data assimilation; and (iv) asynchronous observations in the DA cycle. In order to illustrate the novelty of the SIEnKS, and to motivate its computational cost / prediction accuracy trade off advantages, we discuss each of these proposed applications in the following.

## 4 Applications of single-iteration smoothing

### 4.1 Nonlinear observation operators

Just as the IEnKS extends the linear 4D-MAP cost function, the filter cost function Eq. (41) can be extended with Newton iterates in the presence of a nonlinear observation operator. The maximum likelihood ensemble filter (MLEF) of Zupanski (2005) and Zupanski et al. (2008) is an estimator designed to process nonlinear observation operators and can be derived in the common ETKF formalism. Particularly, the algorithm can be granted a bundle and transform variant like the IEnKS (Asch et al., 2016, see section 6.7.2.1) designed to approximate the directional derivative of the nonlinear observation operator with respect to the forecast ensemble perturbations at the forecast mean,

$$\widetilde{\mathbf{Y}}_k := \nabla|_{\hat{\boldsymbol{x}}_k^{\mathrm{fore}}} \left[ \mathcal{H}_k \right] \mathbf{X}_k^{\mathrm{fore}}, \tag{72}$$





used in the nonlinear filter cost function gradient

$$\nabla\widetilde{\mathcal{J}} := (N_e - 1)\,\boldsymbol{w} - \widetilde{\mathbf{Y}}_k^\top \mathbf{R}_k^{-1} \left[ \boldsymbol{y}_k - \mathcal{H}_k \left( \hat{\boldsymbol{x}}_k^{\text{fore}} + \mathbf{X}_k^{\text{fore}} \boldsymbol{w} \right) \right]. \tag{73}$$

When the forecast error dynamics are weakly nonlinear, the MLEF-style nonlinear filter cost function optimization provides a direct extension of the SIEnKS framework. Consider the transform as defined in the sub-routine in Algorithm 9; this type of filter analysis is interchangeable with the usual ensemble transform in Algorithm 1. In this way, the EnKS and the SIEnKS can each process nonlinear observation operators with an iterative optimization in the filter cost function alone and subsequently apply their retrospective analyses as usual. We refer to the EnKS analysis with MELF transform as the maximum likelihood ensemble smoother (MLES), though we refer to the SIEnKS as usual whether it uses a single iteration or multiple iterations of the solution to the filter cost function. Note that only the transform step needs to be interchanged in Algorithms 6 and 8, so that we do not provide additional pseudo-code.

Consider that for the MLES and the SIEnKS, the number of Hessian square root inverse calculations expands in the number of iterations used in Algorithm 9 to compute the transform for each of the $S$ observations in the DAW. For each iteration of the IEnKS, this again requires only a single square root inverse calculation of the 4D-MAP cost function Hessian. However, even if the forecast error dynamics are weakly nonlinear, optimizing versus the nonlinear observation operator requires $L$ ensemble simulations per each iteration used to optimize the cost function.

## 4.2 Adaptive inflation and the finite-size formalism

Due to the bias of Kalman-like estimators in nonlinear dynamics, covariance inflation, as in Algorithm 4, is widely used to regularize these schemes. In particular, this can ameliorate the systematic under-estimation of the prediction / posterior uncertainty due to sample error and bias. Empirically tuning the multiplicative inflation coefficient $\lambda \geq 1$ can be effective in stationary dynamics. However, empirically tuning this parameter can be costly, potentially requiring many model simulations, and the tuned value may not be optimal across time scales in which the dynamical system becomes non-stationary. A variety techniques are used in practice for adaptive covariance estimation, inflation or augmentation, accounting for these deficiencies of Kalman-like estimators (Tandeo et al., 2020, and references therein).

One alternative to empirically tuning $\lambda$ is to derive an adaptive multiplicative covariance inflation factor via a hierarchical Bayesian model by including a prior on the background mean and covariance $p\left(\overline{\boldsymbol{x}}_1^{\text{fore}}, \mathbf{B}_1^{\text{fore}}\right)$, as in the finite-size formalism of Bocquet (2011), Bocquet and Sakov (2012) and Bocquet et al. (2015). This formalism seeks to marginalize out over the first two moments of the background, yielding a Gaussian mixture model for the forecast-prior as

$$p\left(\boldsymbol{x}_1 | \mathbf{E}_1^{\text{fore}}\right) = \int \mathrm{d}\overline{\boldsymbol{x}}_1^{\text{fore}} \mathrm{d}\mathbf{B}_1^{\text{fore}} p\left(\boldsymbol{x}_1 | \mathbf{E}_1^{\text{fore}}, \overline{\boldsymbol{x}}_1^{\text{fore}}, \mathbf{B}_1^{\text{fore}}\right) p\left(\overline{\boldsymbol{x}}_1^{\text{fore}}, \mathbf{B}_1^{\text{fore}} | \mathbf{E}_1^{\text{fore}}\right). \tag{74}$$

Using Jeffreys' hyperprior for $\overline{\boldsymbol{x}}_1^{\text{fore}}$ and $\mathbf{B}_1^{\text{fore}}$, the ensemble-based filter MAP cost function can be derived as proportional to

$$\widetilde{\mathcal{J}}(\boldsymbol{w}) := \frac{1}{2}\parallel \boldsymbol{y} - \mathcal{H}\left(\hat{\boldsymbol{x}}_1^{\text{fore}} + \mathbf{X}_1^{\text{fore}}\boldsymbol{w}\right) \parallel_{\mathbf{R}_1}^2 + \frac{N_e}{2}\log\left(\epsilon_{N_e} + \parallel \boldsymbol{w} \parallel^2\right) \tag{75}$$

where $\epsilon_{N_e} := 1 + \frac{1}{N_e}$. Notice that Eq. (75) is non-quadratic in $\boldsymbol{w}$ regardless of whether $\mathcal{H}_1$ is linear or nonlinear, such that one can iteratively optimize the solution to the nonlinear filter cost function with a Gauss-Newton approximation of the descent.





When accounting for the nonlinearity in the ensemble evolution and the sample error due to small ensemble sizes in perfect models, optimizing the extended cost function in Eq. (75) can be an effective means to regularize the bias of the EnKF. In the presence of significant model error, one may need to extend the finite-size formalism to the variant developed by Raanes et al. (2019a).

Algorithm 10 presents a revised version of the EnKF-N transform calculation of Bocquet et al. (2015); the version presented here is explicitly based on the IEnKS transform approximation of the gradient of the observation operator. The hyper-prior for the background mean and covariance is similarly introduced to the IEnKS and optimized over an extended 4D-MAP cost function. Note that, in the case when $\mathcal{H}_k \equiv \mathbf{H}_k$ is linear, a dual, scalar optimization can be performed for the filter cost function with less numerical expense. However, there is no similar reduction to the extended 4D-MAP cost function and, in order to emphasize the outer-loop differences while iteratively optimizing the inflation hyper-parameter, we focus on the transform variant analogous to the IEnKS optimization.

Extending the adaptive covariance inflation in the finite-size formalism to either the EnKS or the SIEnKS is simple, requiring that the ensemble transform calculation is interchanged with Algorithm 10 and that the tuned multiplicative inflation step is eliminated. The IEnKS-N transform variant, including adaptive inflation as above, is described in Algorithm 11. Notice that iteratively optimizing the inflation hyper-parameter comes at the additional expense of square root inverse Hessian calculations for the EnKS and the SIEnKS, while the IEnKS also requires $L$ additional ensemble simulations for each iteration.

### 4.3 Multiple data assimilation

When the lag $L > 1$ is long, temporally interpolating the posterior estimate in the DAW via the nonlinear model solution as in Eq. (70) becomes increasingly nonlinear. In chaotic dynamics, the small simulation errors introduced this way eventually degrade the posterior estimate, and this interpolation becomes unstable for $L$ sufficiently large. Furthermore, for the 4D-MAP cost function, observations only distantly connected with the initial condition at the beginning of the DAW render the cost function increasingly nonlinear, with more local minima that may strongly affect the performance of the optimization.

Multiple data assimilation is a commonly used technique based on statistical tempering (Neal, 1996), designed to relax the nonlinearity of performing the MAP estimate, by artificially inflating the variances of the observation errors with weights and assimilating these observations multiple times. Multiple data assimilation is made consistent with the Bayesian posterior in perfect, linear-Gaussian models by appropriately choosing weights so that, over all times an observation vector is assimilated, all of its associated weights sum to one (Emerick and Reynolds, 2013). Given Gaussian likelihood functions, this implies that

the sum of the precision matrices over the multiple assimilation steps equals $\mathbf{R}^{-1}$, as with the usual Kalman filter update.

Multiple data assimilation is integrated into the EnRML for static DAWs in reservoir modelling (Evensen, 2018, and references therein). With the fixed-lag, sequential EnKS, there is no reason to perform MDA as the assimilation occurs in a single pass over each observation with the filter step as in the ETKF. Sequential MDA, with DAWs shifting in time, was first derived with the IEnKS by Bocquet and Sakov (2014); in order to sample the appropriate density, the IEnKS MDA estimation is broken

over two stages. Firstly, in the "balancing" stage, the IEnKS "fully assimilates" all "partially assimilated observations", targeting the joint posterior statistics. Secondly, the window of the partially assimilated observations is shifted in time with the MDA



stage. The SIEnKS is similarly broken over these two stages, using the same weights as the IEnKS above. However, there is an important difference in the way MDA is formulated for the SIEnKS versus the IEnKS. For the SIEnKS, each observation in the DAW is sequentially assimilated with the EnKS filter cost function instead of the global analysis in the IEnKS. The sequential filter analysis, in particular, constrains the posterior interpolation to the observations in the balancing stage, whereas the posterior interpolation is performed with a free forecast from the marginal posterior in the IEnKS. Our novel SIEnKS MDA scheme is derived as follows.

Recall our algorithmically stationary DAW, $\{t_1, \cdots, t_L\}$, and suppose at the moment that there is a shift of $S = 1$ and an arbitrary lag $L$. We take the notation that the covariance matrices for the likelihood functions are inflated as

$$p\left(\boldsymbol{y}^\beta | \boldsymbol{x}\right) := N\left(\boldsymbol{y} - \mathcal{H}(\boldsymbol{x}) | \mathbf{0}, \beta^{-1} \mathbf{R}\right) \tag{76}$$

where the observation weights are assumed $0 < \beta \leq 1$. We index the weight for observation $\boldsymbol{y}_k$ at the present-time $t_L$ as $\beta_{k|L}$. For consistency with the perfect, linear-Gaussian model, we require that

$$\sum_{i=1}^{L} \beta_{i|L} = 1. \tag{77}$$

This implies that as we assimilate an observation vector $L$-total times, shifting the algorithmically stationary DAW, the sum of the weights used to assimilate the observation equals one.

We denote

$$\alpha_{k|L} := \sum_{i=k}^{L} \beta_{i|L} \tag{78}$$

as the fraction of the observation $\boldsymbol{y}_k$ that has been assimilated after the analysis step at the time $t_L$. Note that, under the Gaussian likelihood assumption, and assuming the independence of the fractional observations, this implies

$$\prod_{i=k}^{L} p\left(\boldsymbol{y}^{\beta_{i|L}} | \boldsymbol{x}\right) = p\left(\boldsymbol{y}^{\alpha_{k|L}} | \boldsymbol{x}\right). \tag{79}$$

Let $\boldsymbol{\beta}_{l:k|L}$ and $\boldsymbol{\alpha}_{l:k|L}$ denote the length-$(l - k + 1)$ vectors

$$\boldsymbol{\beta}_{l:k|L} = \begin{pmatrix} \beta_{l|L} & \cdots & \beta_{k|L} \end{pmatrix}, \tag{80a}$$

$$\boldsymbol{\alpha}_{l:k|L} = \begin{pmatrix} \alpha_{l|L} & \cdots & \alpha_{k|L} \end{pmatrix}. \tag{80b}$$

We then define the sequences

$$\boldsymbol{y}_{l:k}^{\boldsymbol{\beta}_{l:k|L}} := \left\{ \boldsymbol{y}_l^{\beta_{l|L}}, \boldsymbol{y}_{l-1}^{\beta_{l-1|L}}, \cdots, \boldsymbol{y}_k^{\beta_{k|L}} \right\}, \tag{81a}$$

$$\boldsymbol{y}_{l:k}^{\boldsymbol{\alpha}_{l:k|L}} := \left\{ \boldsymbol{y}_l^{\alpha_{l:L}}, \boldsymbol{y}_{l-1}^{\alpha_{l-1|L}}, \cdots, \boldsymbol{y}_k^{\alpha_{k|l}} \right\}, \tag{81b}$$

as the observations $\boldsymbol{y}_{l:k}$ in the current DAW $\{t_1, \cdots, t_L\}$, with: Eq. (81a), the corresponding MDA weights for this DAW; and, Eq. (81b), the total portion of each observation assimilated in the MDA conditional density for this DAW after the analysis step. Similar definitions apply with the indices $l : k|L - 1$, but relative to the previous DAW.



For the current DAW the balancing stage is designed to sample the joint posterior density

$$p\left(\boldsymbol{x}_{L:1}|\boldsymbol{y}_{L:1}\right) \tag{82}$$

where the current cycle is initialized with a sample of the MDA conditional density

$$p\left(\boldsymbol{x}_0|\boldsymbol{y}_{L-1:0}^{\boldsymbol{\alpha}_{L-1:0|L-1}}\right). \tag{83}$$

That is, from the previous cycle we have a marginal estimate for $\boldsymbol{x}_0$ given the sequence of observations $\boldsymbol{y}_{L-1:0}$, where the portion of observation $\boldsymbol{y}_k$ that has been assimilated already is given by $\alpha_{k|L-1}$. Notice that $\alpha_{0|L-1}=1$ so that $\boldsymbol{y}_0$ has already been fully assimilated. To fully assimilate $\boldsymbol{y}_1$, we note that $1-\alpha_{1|L-1}=\beta_{1|L}$, and therefore

$$p\left(\boldsymbol{x}_{1:0}|\boldsymbol{y}_{L-1:2}^{\boldsymbol{\alpha}_{L-1:2|L-1}},\boldsymbol{y}_{1:0}\right) \propto p\left(\boldsymbol{y}_1^{\beta_{1|L}}|\boldsymbol{x}_1\right)p(\boldsymbol{x}_1|\boldsymbol{x}_0)\,p\left(\boldsymbol{x}_0|\boldsymbol{y}_{L-1:0}^{\boldsymbol{\alpha}_{L-1:0|L-1}}\right). \tag{84}$$

The above corresponds to a single simulation / analysis step in an EnKS cycle where the observation $\boldsymbol{y}_1^{\beta_{1|L}}$ is assimilated and a retrospective re-analysis is applied to the ensemble at $t_0$.

More generally, to fully assimilate observation $\boldsymbol{y}_k$, we assimilate the remaining portion left un-assimilated from the last DAW, given as $1-\alpha_{k|L-1}$. We define an inductive step describing the density for $\boldsymbol{x}_{k:0}$ which has fully assimilated $\boldsymbol{y}_{k:0}$, though is yet to assimilate the remaining portions of observations $\boldsymbol{y}_{L-1:k+1}$, as

$$p\left(\boldsymbol{x}_{k:0}|\boldsymbol{y}_{L-1:k+1}^{\boldsymbol{\alpha}_{L-1:k+1|L-1}},\boldsymbol{y}_{k:0}\right) \propto p\left(\boldsymbol{y}_k^{1-\alpha_{k|L-1}}|\boldsymbol{x}_k\right)p(\boldsymbol{x}_k|\boldsymbol{x}_{k-1})p\left(\boldsymbol{x}_{k-1:0}|\boldsymbol{y}_{L-1:k}^{\boldsymbol{\alpha}_{L-1:k|L-1}},\boldsymbol{y}_{k-1:0}\right). \tag{85}$$

For $k=2,\cdots,L-2$, this describes a subsequent simulation / analysis step of an EnKS cycle but where the observation $\boldsymbol{y}_k^{1-\alpha_{k|L-1}}$ is assimilated and a retrospective analysis is applied to the ensemble at times $t_0,\cdots,t_{k-1}$. A subsequent EnKS analysis gives

$$p(\boldsymbol{x}_{L-1:0}|\boldsymbol{y}_{L-1:0}) \propto p\left(\boldsymbol{y}_{L-1}^{1-\alpha_{L-1|L-1}}|\boldsymbol{x}_{L-1}\right)p(\boldsymbol{x}_{L-1}|\boldsymbol{x}_{L-2})p\left(\boldsymbol{x}_{L-2:0}|\boldsymbol{y}_{L-1}^{\alpha_{L-1|L-1}},\boldsymbol{y}_{L-2:0}\right), \tag{86}$$

i.e., this samples the joint posterior for the last DAW. A final EnKS analysis is used to assimilate $\boldsymbol{y}_L$, for which no portion was already assimilated in the previous DAW,

$$p(\boldsymbol{x}_{L:1}|\boldsymbol{y}_{L:1}) \propto p(\boldsymbol{y}_L|\boldsymbol{x}_L)p(\boldsymbol{x}_L|\boldsymbol{x}_{L-1})p(\boldsymbol{x}_{L-1:0}|\boldsymbol{y}_{L-1:0}). \tag{87}$$

We thus define an initial ensemble

$$\mathbf{E}_0^{\mathrm{bal}} \sim p\left(\boldsymbol{x}_0|\boldsymbol{y}_{L-1:0}^{\boldsymbol{\alpha}_{L-1:0|L-1}}\right). \tag{88}$$

In the balancing stage, the observation error covariance weights are defined by

$$\eta_{k|L} := 1-\alpha_{k|L-1}, \tag{89}$$





where $\eta_{L|L} = 1$. In the case where $\beta_{k|L} = \frac{1}{L}$ for all $k$, we obtain the balancing weights as $\eta_{k|L} = \frac{k}{L}$ for all $k = 1, \cdots, L$. An EnKS cycle initialized as in Eq. (88), using the balancing weights in Eq. (89), sequentially and recursively samples

$$\mathbf{E}_{k:0}^{\mathrm{bal}} \sim p\left(\boldsymbol{x}_{k:0}|\boldsymbol{y}_{L-1:k+1}^{\boldsymbol{\alpha}_{L-1:k+1|L-1}}, \boldsymbol{y}_{k:0}\right) \tag{90}$$

from the inductive relationship in Eq. (85), where the final analysis gives $\mathbf{E}_{L:0}^{\mathrm{bal}} \equiv \mathbf{E}_{L:0|L}^{\mathrm{smth}}$ from Eq. (87).

To subsequently shift the DAW and initialize the next cycle, we target the density $p\left(\boldsymbol{x}_1|\boldsymbol{y}_{L:1}^{\boldsymbol{\alpha}_{L:1|L}}\right)$. Given $p\left(\boldsymbol{x}_0|\boldsymbol{y}_{L-1:0}^{\boldsymbol{\alpha}_{L-1:0|L-1}}\right)$, the target density is sampled by assimilating each observation $\boldsymbol{y}_k^{\beta_{k|L}}$ so that the portion of each observation assimilated becomes $\boldsymbol{y}_{L:1}^{\boldsymbol{\alpha}_{L:1|L}}$. Notice that for $k = 1, \cdots, L-2$,

$$p\left(\boldsymbol{x}_{k:0}|\boldsymbol{y}_{L-1:k+1}^{\boldsymbol{\alpha}_{L-1:k+1|L-1}}, \boldsymbol{y}_{k:0}^{\boldsymbol{\alpha}_{k:0|L}}\right) \propto p\left(\boldsymbol{y}_k^{\beta_{k|L}}|\boldsymbol{x}_k\right) p(\boldsymbol{x}_k|\boldsymbol{x}_{k-1}) p\left(\boldsymbol{x}_{k-1:0}|\boldsymbol{y}_{L-1:k}^{\boldsymbol{\alpha}_{L-1:k|L-1}}, \boldsymbol{y}_{k-1:0}^{\boldsymbol{\alpha}_{k-1:0|L}}\right). \tag{91}$$

The above recursion corresponds to an EnKS step in which the observation $\boldsymbol{y}_k^{\beta_{k|L}}$ is assimilated and a retrospective analysis is applied to ensembles at times $t_0, \cdots, t_{k-1}$. Subsequent EnKS analyses using the MDA weights then give

$$p\left(\boldsymbol{x}_{L-1:0}|\boldsymbol{y}_{L-1:0}^{\boldsymbol{\alpha}_{L-1:0|L}}\right) \propto p\left(\boldsymbol{y}_k^{\beta_{L-1|L}}|\boldsymbol{x}_{L-1}\right) p(\boldsymbol{x}_{L-1}|\boldsymbol{x}_{L-2}) p\left(\boldsymbol{x}_{L-2:0}|\boldsymbol{y}_{L-1}^{\boldsymbol{\alpha}_{L-1|L-1}}, \boldsymbol{y}_{L-2:0}^{\boldsymbol{\alpha}_{L-2:0|L}}\right), \tag{92}$$

$$p\left(\boldsymbol{x}_{L:0}|\boldsymbol{y}_{L:0}^{\boldsymbol{\alpha}_{L:0|L}}\right) \propto p\left(\boldsymbol{y}_L^{\beta_{L|L}}|\boldsymbol{x}_L\right) p(\boldsymbol{x}_L|\boldsymbol{x}_{L-1}) p\left(\boldsymbol{x}_{L-1:0}|\boldsymbol{y}_{L-1:0}^{\boldsymbol{\alpha}_{L-1:0|L}}\right). \tag{93}$$

We therefore perform a second EnKS cycle using the MDA observation error covariance weights $\beta_{k|L}$ to sample the target density. Given that $\eta_{1|L} = \beta_{1|L}$, the first analysis of the balancing stage in Eq. (84) is identical to the first analysis in the MDA stage, corresponding to $k = 1$ in Eq. (91). Therefore, this first EnKS analysis step can be re-used between the two stages.

Define an initial ensemble for the MDA stage, re-using the first analysis in the balancing stage, as

$$\mathbf{E}_1^{\mathrm{mda}} \equiv \mathbf{E}_1^{\mathrm{bal}} \sim p\left(\boldsymbol{x}_1|\boldsymbol{y}_{L-1:2}^{\boldsymbol{\alpha}_{L-1:2|L-1}}, \boldsymbol{y}_{1:0}\right). \tag{94}$$

An EnKS cycle initialized as in Eq. (94), using the MDA weights $\beta_k$, sequentially and recursively samples

$$\mathbf{E}_{k:1}^{\mathrm{mda}} \sim p\left(\boldsymbol{x}_{k:1}|\boldsymbol{y}_{L-1:k+1}^{\boldsymbol{\alpha}_{L-1:k+1|L-1}}, \boldsymbol{y}_{k:0}^{\boldsymbol{\alpha}_{k:0|L}}\right) \tag{95}$$

from the relationship in Eq. (91). The final analysis samples the density $p\left(\boldsymbol{x}_{L:1}|\boldsymbol{y}_{L:0}^{\boldsymbol{\alpha}_{L:0|L}}\right) \propto p\left(\boldsymbol{x}_{L:1}|\boldsymbol{y}_{L:1}^{\boldsymbol{\alpha}_{L:1|L}}\right)$, as in Eq. (93), which is used to initialize the next cycle. To make the scheme more efficient, we note that we need only sample the marginal $p\left(\boldsymbol{x}_1|\boldsymbol{y}_{L:1}^{\boldsymbol{\alpha}_{L:1|L}}\right)$ to re-initialze the next cycle of the algorithm; this means that the inner loop of the EnKS in the second stage needs only store and sequentially condition the ensemble $\mathbf{E}_1^{\mathrm{mda}}$ with the retrospective filter analyses in this stage. Combining the two stages together into a single cycle that produces forecast, filter and smoother statistics over the DAW $\{t_1, \cdots, t_L\}$, as well as the ensemble initialization for the next cycle, requires $2L$ ensemble simulations. Due to the convoluted nature of the indexing over multiple DAWs above, a schematic of the two stages of the SIEnKS MDA cycle is presented in Fig. 4.

The MDA algorithm is generalized to shift windows of $S > 1$ with the number of ensemble forecasts remaining invariant at $2L$ when using "blocks" of uniform MDA weights in the DAW. Assume that $L = SQ$ for some positive integer $Q$, so that we partition $\boldsymbol{y}_{L:1}$ into $Q$ total blocks of observations each of length $S$. In this case, the perfect, linear-Gaussian model consistency



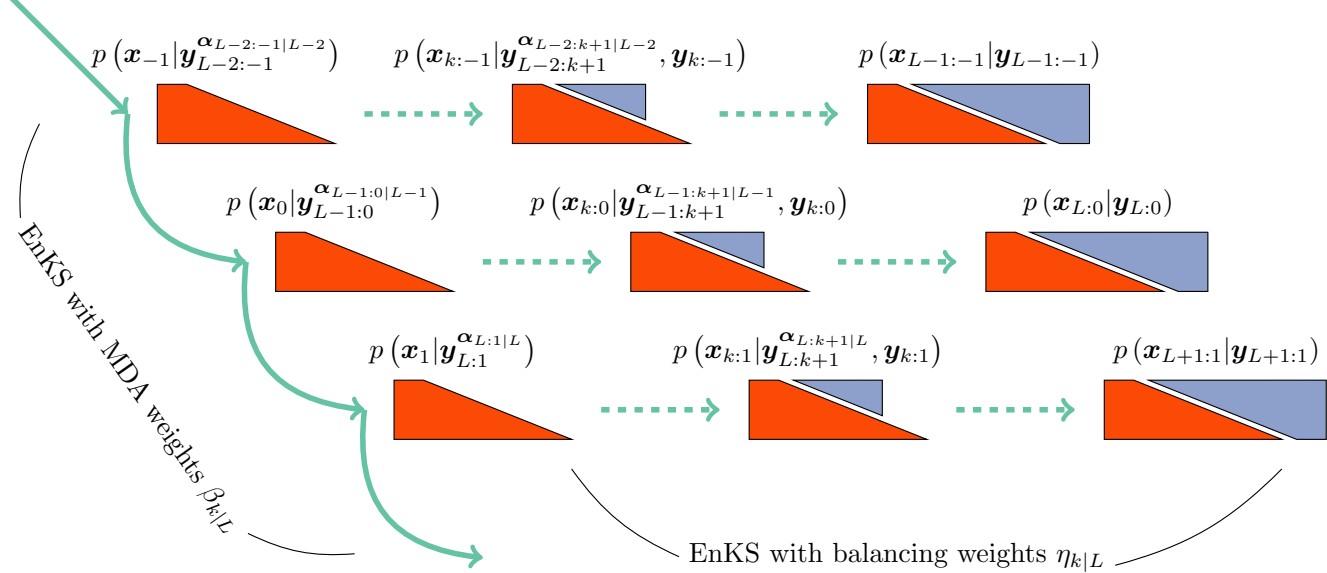

**Figure 4.** A schematic of the two stages of the SIEnKS MDA cycle. The DAW of the SIEnKS moves forward in time from top to bottom, where the EnKS stage using MDA weights pushes the MDA conditional density, left-most, forward in time. The middle layer represents the indexing of the stationary DAW, while the top layer represents a DAW one cycle back in time, and the bottom layer represents a DAW one cycle forward in time. The balancing density is sampled sequentially and recursively with an EnKS stage using the balancing weights, moving from left to right in each cycle. For the current DAW, the middle balancing density has fully assimilated observations $\boldsymbol{y}_{k:0}$ and has partially assimilated observations $\boldsymbol{y}_{L:k+1}^{\boldsymbol{\alpha}_{L-1:k+1|L-1}}$. The EnKS stage with balancing weights completes when sampling the joint posterior, and the EnKS stage with MDA weights begins again.

constraint is revised as

$$\beta_{k|L} = \tilde{\beta}_{i|L} \quad \text{for } i := \left\lceil \frac{k}{S} \right\rceil, \quad \text{with } \sum_{j=1}^{Q} \tilde{\beta}_{j|L} = 1, \tag{96}$$

where the above brackets represent rounding up to the nearest integer. This ensures again that the weights corresponding to the $Q$ total times that $\boldsymbol{y}_k$ is assimilated sum to one. With this weighting scheme, the equivalence between the balancing and MDA stages' first EnKS filter analysis extends to the first $S$-total EnKS filter analyses, and therefore $\mathbf{E}_S^{\text{mda}} \equiv \mathbf{E}_S^{\text{bal}}$ initializes the MDA stage. One can further reduce the memory usage by only performing the retrospective conditioning in the balancing stage on the states $\mathbf{E}_{S:0}^{\text{bal}}$ stored in memory. This samples the density $p(\boldsymbol{x}_{S:0}|\boldsymbol{y}_{L:0})$ in the final cycle before the estimates for these states are discarded from the DAW. MDA variants of the SIEnKS and the (Lin-)IEnKS are presented in Algorithms 12 and 13.

The primary difference between the SIEnKS and IEnKS MDA schemes lies in the sequential balancing analysis versus the global balancing analysis performed in the 4D-MAP optimization. The IEnKS MDA scheme is not always robust in this 4D-MAP balancing estimation because the MDA conditional prior that initializes the scheme may actually lie far away from the





MAP solution for the balanced, joint posterior. As a consequence, this may require many iterations of the balancing stage to perform the nonlinear optimization. On the other hand, the sequential SIEnKS MDA approach uses the partially unassimilated observations in the DAW directly as a boundary condition to the interpolation of the joint posterior over the DAW with the sequential EnKS filter cycle. In particular, for long DAWs, this means that the SIEnKS furthermore controls error growth in the ensemble simulation that accumulates over the long free forecast in the 4D-MAP approach. Every step of the interpolation of the SIEnKS MDA scheme actually arises as a prediction from a kind of filtering density as in Eq. (90).

Note how the cost of assimilation scales differently between the SIEnKS and the IEnKS when performing MDA. Both the IEnKS and the SIEnKS use the same weights $\eta_{k|L}$ and $\beta_{k|L}$ for their balancing and MDA stages. However, each stage of the IEnKS separately performs an iterative optimization of the 4D-MAP cost function. While each iteration therein requires only a single square root inverse calculation of the cost function Hessian, the iterative solution requires at least $2L$ total ensemble simulations in order to optimize and interpolate the estimates over the DAW. An efficient version of the scheme can be performed as such by using the same free ensemble simulation initialized as in Eq. (88) in order to assimilate each of the observation sequences $\boldsymbol{y}_{L:1}^{\boldsymbol{\eta}_{L:1|L}}$ and $\boldsymbol{y}_{L:1}^{\boldsymbol{\beta}_{L:1|L}}$. However, the IEnKS additionally requires $S$ total ensemble simulations in order to shift the DAW thereafter.

This differs from the SIEnKS which, as discussed above, requires a fixed $2L$ ensemble simulations over the DAW. However, the computational barrier to the SIEnKS MDA scheme lies in the fact that it requires $2L - S$ square root inverse calculations, corresponding to each unique filter cost function solution over the two stages; in the case that MDA is combined with, e.g., the ensemble transform in the MLEF this further grows to the sum of the number of iterations $\sum_{j=1}^{2L-S} i_j$ where $i_j$ iterations are used in the $j$-th optimization of a filter cost function. However, when the cost of an ensemble simulation is sufficiently greater than the cost of the square root inverse in the ensemble dimension, the SIEnKS MDA scheme can substantially reduce the leading order the computational cost of ensemble-variational smoothing with MDA, and especially when $S > 1$.

### 4.4 Asynchronous data assimilation

In online settings, fixed-lag smoothers with shifts of $S > 1$ are computationally more efficient in terms of how rapidly an observation / analysis / forecast cycle can be computed in real-time. Specifically, when $S > 1$, the number of smoother cycles necessary to traverse a time series of observations with the sequential DAWs is decreased; versus a shift of one, the number of cycles necessary is reduced by the factor of $S$. A barrier to using the SIEnKS with $S > 1$ lies in the fact that the sequential filter analysis of the EnKS does not in and of itself provide a means to asynchronously assimilate observations. However, the SIEnKS differs from the EnKS in numerically simulating lagged states in the DAW. When one interpolates the posterior estimate with the dynamical model over lagged states, one can easily revise the algorithm to assimilate any newly available data corresponding to a time within the past simulation window. The filter step and retrospective re-analysis of a variety of filter schemes can be used this way, though the weights in MDA need to be adjusted accordingly. There are many ways in which one may actually design methods of excluding observations and re-introducing them in a later DAW with a shift $S > 1$. In the current work, the SIEnKS assimilates all observations synchronously, even with $S > 1$. A systematic investigation of algorithms that would optimize this asynchronous assimilation in single-iteration smoothers goes beyond the scope of the





current work. However, this key difference between the EnKS and the SIEnKS is important to note and will be considered later.

## 5 Numerical benchmarks

### 5.1 Algorithm cost analysis

Let us fix the ensemble size $N_e$ for the following considerations and let us suppose that the cost of the nonlinear ensemble
simulation is fixed for every $\Delta_k$, equal to $C_{\mathcal{M}}$ floating-point operations (flops). In order to compute the ensemble transform in any of the methods, we assume that the inversion of the approximate Hessian $\widetilde{\mathbf{H}}_{\widetilde{\mathcal{J}}}$, and its square root, is performed with an SVD-based approach with cost of order $\mathcal{O}\left(N_e^3\right)$ flops. This assures stability and efficiency in the sense that the computation of all of $\mathbf{T} = \widetilde{\mathbf{H}}_{\widetilde{\mathcal{J}}}^{-\frac{1}{2}}$, $\mathbf{T}^{-1} = \widetilde{\mathbf{H}}_{\widetilde{\mathcal{J}}}^{\frac{1}{2}}$ and $\widetilde{\mathbf{H}}_{\widetilde{\mathcal{J}}}^{-1}$ combined is dominated by the cost of the SVD of the symmetric, $N_e \times N_e$ matrix $\widetilde{\mathbf{H}}_{\widetilde{\mathcal{J}}}$. If a method is iterative, we denote the number of iterations used in the scheme with $i_j$, where the sub-index $j$ distinguishes
distinct iterative optimizations.

A summary of how each of the EnKS, SIEnKS and IEnKS scale in their numerical cost is presented in Tables 1 and 2. This analysis is easily derived based on the pseudo-code in Appendix A and with the discussions in Section 4. Table 1 presents schemes that are used in the SDA configuration, while Table 2 presents schemes that are used in the MDA configurations. Note that, while adaptive inflation in the finite-size formalism can be used heuristically to estimate a power of the joint posterior,
this has not been found to be fully compatible with MDA (Bocquet and Sakov, 2014), and this combination of techniques is not considered here.

**Table 1.** Order of SDA cycle flops for lag=$L$, shift=$S$, tuned inflation (TI) or adaptive inflation (AI) / nonlinear observation operator (NO)

|  | EnKS / MLES | SIEnKS | IEnKS |
|---|---|---|---|
| TI | $SC_{\mathcal{M}} + SN_e^3$ | $(L+S)C_{\mathcal{M}} + SN_e^3$ | $(i_1 L + S)C_{\mathcal{M}} + i_1 N_e^3$ |
| AI / NO | $SC_{\mathcal{M}} + \sum_{j=L-S+1}^{L} i_l N_e^3$ | $(L+S)C_{\mathcal{M}} + \sum_{j=L-S+1}^{L} i_j N_e^3$ | $(i_1 L + S)C_{\mathcal{M}} + i_1 N_e^3$ |

For realistic geophysical models, note that the maximal ensemble size $N_e$ is typically of order $\mathcal{O}\left(10^2\right)$ while the state dimension $N_x$ can be of order $\mathcal{O}\left(10^9\right)$ (Carrassi et al., 2018); therefore, the cost of all algorithms reduce to terms of $C_{\mathcal{M}} \gg N_e^3$ at leading order in target applications. It is easy to see then that the EnKS / MLES has a cost that is of order of the regular ETKF
735 / MLEF filter cycle, representing the least expensive of the estimation schemes. Consider now in row one of Table 1, the $i_1$ in the IEnKS represents the number of iterations utilized to minimize the 4D-MAP cost function. If we set $i_1 = 1$, this represents the cost of the Lin-IEnKS. Particularly, we see that for $S = 1$ and a linear filter cost function, the Lin-IEnKS has the same cost as the SIEnKS. However, even in the case of a linear filter cost function, when $S > 1$ the SIEnKS is more expensive than the Lin-IEnKS. When we let $i_1$ in Table 1 terminate with a maximum possible value, the cost of the IEnKS is bounded at leading
order; however, we demonstrate shortly how, statistically, the number of iterations tends to be bounded without reaching a large maximum in stable filter regimes.





Consider the case when the filter cost function is nonlinear, as when adaptive inflation is used (as defined in Sec. 4.2) or when there is a nonlinear observation operator. Row two of Table 1 shows how the cost of these estimators are differentiated when nonlinearity is introduced; in particular, the cost of the the MLES and the SIEnKS requires one SVD calculation for each iteration used to process each new observation. This renders the SIEnKS notably more expensive than the Lin-IEnKS, which uses a single Hessian SVD calculation to process all observations globally. However, for target applications, such as synoptic meteorology, the additional expense of iteratively optimizing filter cost functions with the SIEnKS versus the single iteration of the Lin-IEnKS in the 4D-MAP cost function is insignificant.

Table 2 describes the cost of the SIEnKS and the IEnKS using MDA when there is a linear observation operator and when there is a nonlinear observation operator. Recall that, at leading order $C_{\mathcal{M}}$, the cost of the SIEnKS is invariant in $S$. This again comes with the caveat that observations are assumed to be assimilated synchronously in this work, while the IEnKS assimilates observations asynchronously by default. Nonetheless, the equivalence between the first $S$ filter cycles in the balancing stage and the MDA stage in the SIEnKS allows the scheme to fix the leading order cost at the expense of two passes over the DAW with the ensemble simulation.

**Table 2.** Order of MDA cycle flops for lag=$L = Q \times S$, shift=$S$, tuned inflation, linear observation operator (LO) or nonlinear observation operator (NO)

|  | SIEnKS | IEnKS |
|---|---|---|
| LO | $2LC_{\mathcal{M}} + (2L - S)N_e^3$ | $[L(i_1 + i_2) + S]C_{\mathcal{M}} + (i_1 + i_2)N_e^3$ |
| NO | $2LC_{\mathcal{M}} + \sum_{j=1}^{2L-S} i_j N_e^3$ | $[L(i_1 + i_2) + S]C_{\mathcal{M}} + (i_1 + i_2)N_e^3$ |

## 5.2 Data assimilation benchmark configurations

To demonstrate the performance, advantages and limitations of the SIEnKS, we produce statistics of its forecast / filter / smoother root mean square error (RMSE) versus the EnKS / Lin-IEnKS / IEnKS in a variety of DA benchmark configurations. Synthetic data is generated in a twin experiment setting, with a simulated "truth twin" generating the observation process. Define the truth twin realization at time $t_k$ as $\boldsymbol{x}_k^{\text{t}}$; we define the ensemble RMSE as

$$\text{RMSE}\left(\mathbf{E}_k^{\text{i}}\right) := \sqrt{\sum_{j=1}^{N_x} \frac{\left(\hat{x}_{j,k}^{\text{i}} - x_{j,k}^{\text{t}}\right)^2}{N_x}}. \tag{97}$$

where i refers to an ensemble label $\text{i} \in \{\text{fore}, \text{filt}, \text{smth}, \text{bal}, \text{mda}\}$, $j$ refers to the state dimension index $j \in \{1, \cdots, N_x\}$ and $k$ refers to time $t_k$ as usual.

A common diagnostic for the accuracy of the linear-Gaussian approximation in the DA cycle is verifying that the ensemble RMSE has approximately the same order as the ensemble spread (Whitaker and Loughe, 1998), which is known as the spread-skill relationship; over-dispersion and under-dispersion of the ensemble both indicate inadequacy of the approximation. Define



the ensemble spread as

$$
\mathrm{spread}\left(\mathbf{E}_k^{\mathrm{i}}\right) := \sqrt{\frac{1}{N_e - 1} \sum_{j=1}^{N_e} \frac{\left(\mathbf{X}_k^{\mathrm{i},j}\right)^\top \left(\mathbf{X}_k^{\mathrm{i},j}\right)}{N_x}}, \tag{98}
$$

where i again refers to an ensemble matrix label, $j$ in this case refers to the ensemble matrix column index and $k$ again refers to time. The spread is then given by the square root of the mean-square-deviation of the ensemble from its mean. Performance of these estimators will be assessed in terms of having low RMSE scores with spread close to the value of the RMSE. Estimators are said to be divergent when either the filter or smoother RMSE is greater than the standard deviation of the observation errors, indicating that initializing a forecast with noisy observations is preferable to the posterior estimate.

The perfect, hidden Markov model in this study is defined by the single layer form of the Lorenz-96 equations (Lorenz, 1996). The state dimension is fixed at $N_x = 40$, with the components of the vector $\boldsymbol{x}$ given by the variables $x_j$ with periodic boundary conditions, $x_0 = x_{40}$, $x_{-1} = x_{39}$ and $x_{41} = x_1$. The time derivatives $\frac{\mathrm{d}\boldsymbol{x}}{\mathrm{d}t} := \boldsymbol{f}(\boldsymbol{x})$, also known as the model tendencies, are given for each state component $j \in \{1, \cdots, 40\}$ by

$$
f_j(\boldsymbol{x}) = -x_{j-2}x_{j-1} + x_{j-1}x_{j+1} - x_j + F. \tag{99}
$$

Each state variable heuristically represents the atmospheric temperature at one of the 40 longitudinal sectors discretizing a lattitudinal circle of the Earth. The Lorenz-96 equations are not a physics-based model, but it mimics fundamental features of geophysical fluid dynamics, including conservative convection, external forcing and linear dissipation of energy (Lorenz and Emanuel, 1998). The term $F$ is the forcing parameter that injects energy into the model, the quadratic terms correspond to energy preserving convection, while the linear term $-x_j$ corresponds to dissipation. With $F \geq 8$, the system exhibits chaotic, dissipative dynamics; we fix $F = 8$ in the following simulations, with the corresponding number of unstable and neutral Lypunov exponents being equal to $N_0 = 14$.

For a fixed $\Delta t$, the dynamical model $\mathcal{M}_k$ is defined by the flow map generated by the dynamical system in Eq. (99). Both the truth twin simulation, generating the observation process, and ensemble simulation, used to sample the appropriate conditional density, are performed with a standard four-stage Runge-Kutta scheme with step size $h = 0.01$. This high-precision simulation is used for generating a ground-truth for these methods, validating the Julia package DataAssimilationBenchmarks.jl (Grudzien et al., 2021) and testing its scalability; however, in general $h = 0.05$ should be sufficient accuracy and is recommended for future use. The nonlinearity of the forecast error evolution is controlled by the length of the forecast window, $\Delta t$. A forecast length $\Delta t = 0.05$ corresponds to a six-hour atmospheric forecast, while for $\Delta t > 0.05$, the level of nonlinearity in the ensemble simulation can be considered to be greater than that is typical for synoptic meteorology.

Localization, hybridization and other standard forms of ensemble-based gain augmentation are not considered in this work for the sake of simplicity. Therefore, in order to control the growth of forecast errors under weakly nonlinear evolution, the rank of the ensemble-based gain must be equal to or greater than the number of unstable and neutral Lyapunov exponents $N_0 = 14$, corresponding to $N_e \geq 15$, see Grudzien et al. (2018) and references therein. In the following experiments, we range the ensemble size as $N_e \in \{15 + 2i\}_{i=0}^{13}$, from the minimal rank needed without gain augmentation to a full-rank ensemble-based





gain. When the number of experimental parameters expands, we restrict to the case where $N_e = 21$ for an ensemble-based gain of actual rank 20, making a reduced-rank approximation of the covariance in analogy to DA in geophysical models.

Observations are full dimensional, such that $N_y = N_x = 40$, and observation errors are distributed according to the Gaussian distribution $N\left(\mathbf{0}, \mathbf{I}_{N_y}\right)$, uncorrelated across state indices with homogeneous variances equal to one. When the observation map is linear, it is defined as $\mathbf{H}_k := \mathbf{I}_{N_x}$; when the observation map is taken to be nonlinear, define

$$\mathcal{H}(\boldsymbol{x}) := \frac{\boldsymbol{x}}{2} \circ \left\{ \mathbf{1} + \left(\frac{\boldsymbol{x}}{10}\right)^{\gamma-1} \right\}, \tag{100}$$

where $\circ$ above refers to the Schur product. This observation operator is drawn from section 6.7.2.2 of Asch et al. (2016), where
the parameter $\gamma$ controls the nonlinearity of the map. In particular, for $\gamma = 1$ this corresponds to the linear observation operator $\mathbf{H}_k$, while $\gamma > 1$ increases the nonlinearity of the map. When we vary the nonlinearity of the observation operator, we take $\gamma \in \{i\}_{i=1}^{11}$ corresponding to ten different nonlinear settings and the linear setting for reference.

When tuned inflation is used to regularize the filters and smoothers, as in Algorithm 4, we take a discretization range of $\lambda \in \{1.0 + 0.01i\}_{i=0}^{10}$, corresponding to the usual Kalman update with $\lambda = 1.0$ and to up to $10\%$ inflation of the empirical
variances with $\lambda = 1.1$. Using tuned inflation, estimator performance is selected for the minimum average forecast RMSE over the experiment for all choices of $\lambda$, unless this is explicitly stated otherwise. When adaptive inflation is used, no additional tuned inflation is utilized. Simulations using the finite-size formalism will be denoted with a -N, following the convention of the EnKF-N. Multiple data assimilation will always be performed with uniform weights as $\beta_{k|L} := \frac{1}{L}$ for all estimators.

For the fully iterative IEnKS, we limit the maximum number of iterations per stage at $i_j = 10$ for $j = 1, 2$. This implies that
the IEnKS can take a maximum of $i_1 + i_2 = 20$ iterations in the MDA configuration to complete a cycle. Iteratively optimizing the filter cost function for the nonlinear observation operator or the adaptive inflation in the MLES(-N) / SIEnKS(-N), the max number of iterations is capped at $i_j = 40$ per analysis. The tolerance for the stopping condition in the filter cost functions is set a $10^{-4}$, while the tolerance for the 4D-MAP estimates is set to $10^{-3}$. However, the scores of the algorithms are to a large extent insensitive to these particular hyper-parameters.

In order to capture the asymptotically stationary statistics of the filter / forecast / smoother processes, we take a long time-average of the RMSE and spread over the time indices $k$. The long experiment average ensures that, for an ergodic dynamical system, we average over the spatial variation on the attractor and that we account for variations in the observation noise realizations that may affect the estimator performance. So that the truth twin simulates observations on the attractor, it is simulated for an initial spin up of $5 \times 10^3$ analysis times before observations are given. Let the time be given as $t_0$ after this initial
spin up. Observations are generated identically for all estimators using the same Gaussian error realizations at a given time to perturb the observation map of the truth twin. At time $t_0$, the ensemble is initialized identically for all estimators (depending on the ensemble size) with the same iid draw from the multivariate Gaussian with mean at the truth twin $\boldsymbol{x}_0^{\mathrm{t}}$ and covariance equal to the identity $\mathbf{I}_{N_x}$. All estimation schemes are subsequently run over observation times indexed as $\{t_k\}_{k=1}^{2.5 \times 10^4}$. As the initial warm-up of the estimators' statistics from this cold start tend to differ from the asymptotically stationary statistics, we discard
the forecast / filter / posterior RMSE and spread corresponding to the observations times $\{t_k\}_{k=1}^{5 \times 10^3}$, taking the time-average of these statistics for the remaining $2 \times 10^4$ analysis time indices. Particularly, this configuration is sufficient to represent





estimator divergence which may have a delayed onset and may not be seen for shorter experimental windows. Empirically, we find that one may miss such a delayed divergence for the EnKS when, e.g., neglecting random, mean-preserving, orthogonal transformations $\mathbf{U}_k$, as in Algorithm 2, and the total experiment length is on $\mathcal{O}\left(10^3\right)$ analysis times.

Forecast statistics are computed for each estimator whenever the ensemble simulates a time index $t_k$ for the first time, before $\boldsymbol{y}_k$ has been assimilated into the estimate. Filter statistics are computed for the first analysis at which the observation $\boldsymbol{y}_k$ has been assimilated into the simulation. For the (Lin-)IEnKS, with $S > 1$, this filter estimate thus includes the information from all observations $\boldsymbol{y}_{L:L-S+2}$ when making a filter estimate for the state at $t_{L-S+1}$. Smoother statistics are computed for the time indices $t_0, t_1, \cdots, t_{S-1}$ in each cycle, as this corresponds to the final analysis for these states before they are passed over by the

shifting DAWs. Whenever a heat plot is shown with missing configurations filled with empty white blocks, this corresponds to "Inf" values in the simulation data. Missing data occurs due to numerical overflow when attempting to invert a close-to-singular cost function Hessian $\widetilde{\mathbf{H}}_{\widetilde{\mathcal{J}}}$, which occurs as a consequence of the collapse of the ensemble spread. When an estimator suffers this catastrophic filter divergence, the experiment output is replaced with Inf values to indicate the failure, and these values are suppressed in the heat plot. Other benchmarks for the EnKS / Lin-IEnKS / IEnKS in the Lorenz-96 model above can be found

in, e.g., Bocquet and Sakov (2014), Asch et al. (2016) and Raanes et al. (2018), which are corroborated here with similar, but slightly different configurations.

### 5.3    Weakly nonlinear forecast error dynamics – linear observations

We fix $\Delta t = 0.05$ in this section, set $S = 1$ and use the linear observation operator in order to demonstrate base-line performance of the estimators in a simple setting. On the other hand, we vary the lag length, the ensemble size and the use of tuned / adaptive inflation or MDA. The lag in this section is varied on a discretization of $L \in \{1 + 3i\}_{i=0}^{30}$. As a first reference simulation, consider the simple case where all schemes use tuned covariance inflation, so that the SIEnKS and the Lin-IEnKS here are formally equivalent. Likewise, with $S = 1$, there is no distinction between asynchronous or synchronous DA. Figure 5 makes a heat plot of the forecast / filter / smoother RMSE and spread as the lag length $L$ is varied along with the ensemble size $N_e$.

It is easy to see the difference in the performance between the EnKS and the iterative SIEnKS / (Lin-)IEnKS schemes.

Particularly, the forecast and fiter RMSE does not change with respect to the lag length in the EnKS, as these statistics are generated independently of the lag with a standard ETKF filter cycle. However, the smoother performance of the EnKS does improve with longer lags, without sacrificing stability over extremely long lag lengths as in the iterative schemes. In particular, all of the iterative schemes use the dynamical model to interpolate the posterior estimate over the DAW. For sufficiently large $L$, this becomes unstable due to the small simulation errors that are amplified by the chaotic dynamics. The scale of the color

map is capped at $0.30$, as a more accurate forecast / filter performance can be attained in this setting with the ETKF alone, as demonstrated by the EnKS.

On the other hand, the iterative estimate of the posterior as in the SIEnKS / (Lin-)IEnKS in this weakly nonlinear setting shows a dramatic improvement in the predictive and posterior accuracy for a tuned lag length. Unlike the standard ETKF observation / analysis / forecast cycle, these iterative smoothers are able to control the error growth in the neutral Lyapunov

susbspace corresponding to the $N_0 = 14$-th Lyapunov exponent. With the ensemble size $N_e = 15$ corresponding to a rank 14



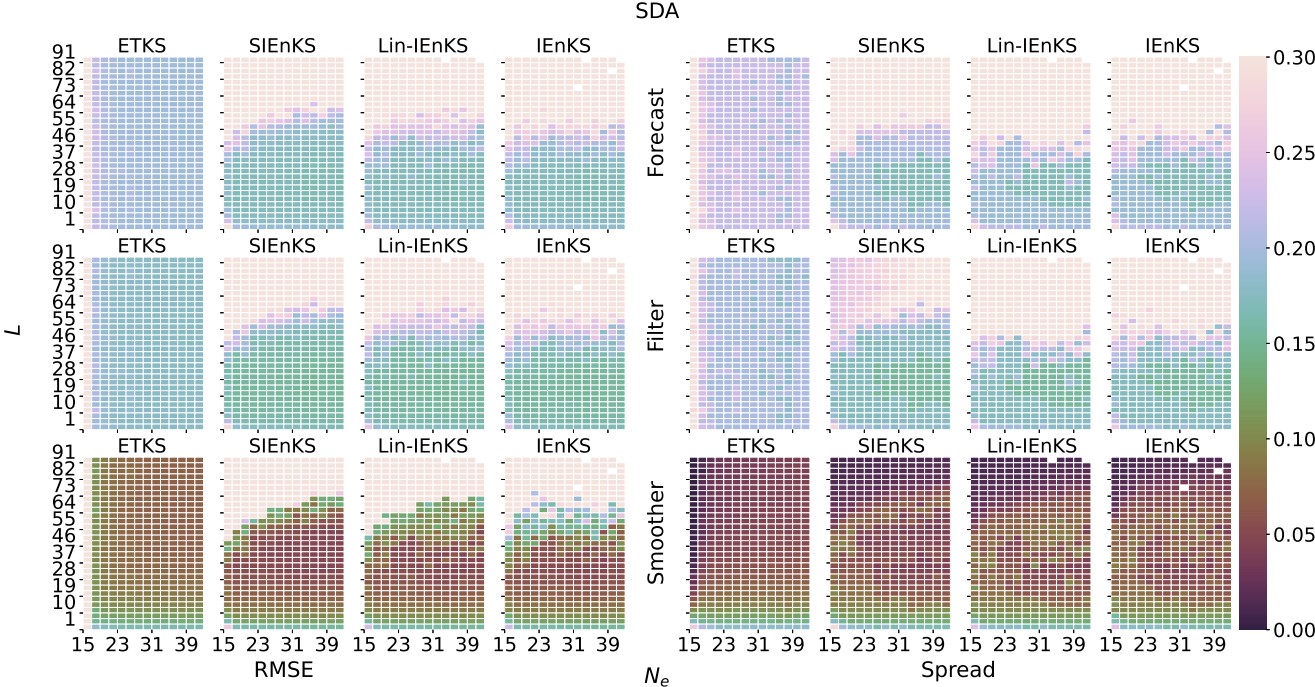

**Figure 5.** Lag length $L$ vertical axis, ensemble size $N_e$ horizontal axis. SDA, tuned inflation, shift $S = 1$, linear observations, $\Delta t = 0.05$.

ensemble-based gain, the iterative smoothers maintain a stable prediction and posterior analysis over a wide range of lags while the EnKS diverges for all lag settings. We notice that the stability regions of the SIEnKS / Lin-IEnKS / IEnKS are otherwise largely the same in this simple benchmark configuration, though the IEnKS has a slightly longer stability over long lags with low sample sizes.

In order to illustrate the difference in accuracy between the iterative schemes and the non-iterative EnKS, Fig. 6 plots a cross section of Fig. 5 for $N_e = 21$. The iterative schemes have almost identical performance until approximately a lag of $L \approx 37$, at which point all schemes become increasingly unstable. The differences shown between the iterative schemes here are insignificant, and may vary between different implementations of these algorithms or pseudo-random seeds. We note that all estimators are also slightly over-dispersive due to selecting the tuned inflation value based on the minimum forecast RMSE,

rather than balancing the RMSE and spread simultaneously. Nonetheless, we clearly demonstrate how all iterative estimators reduce the prediction and posterior error over the non-iterative EnKS approach. Tuning the lag $L$, the forecast error for the iterative schemes is actually lower than the posterior filter error of the EnKS.

Consider now the case where the filter cost function is nonlinear due to the adaptive inflation scheme. Figure 7 makes the same heat plot as in Fig. 5 but where the finite-size formalism is used instead of tuned inflation. All schemes tend to

have slightly weaker performance in this setting, except for the IEnKS-N in the low ensemble size regime. The design of the adaptive inflation scheme is to account for sample error due to the low ensemble size and nonlinearity in the forecast error



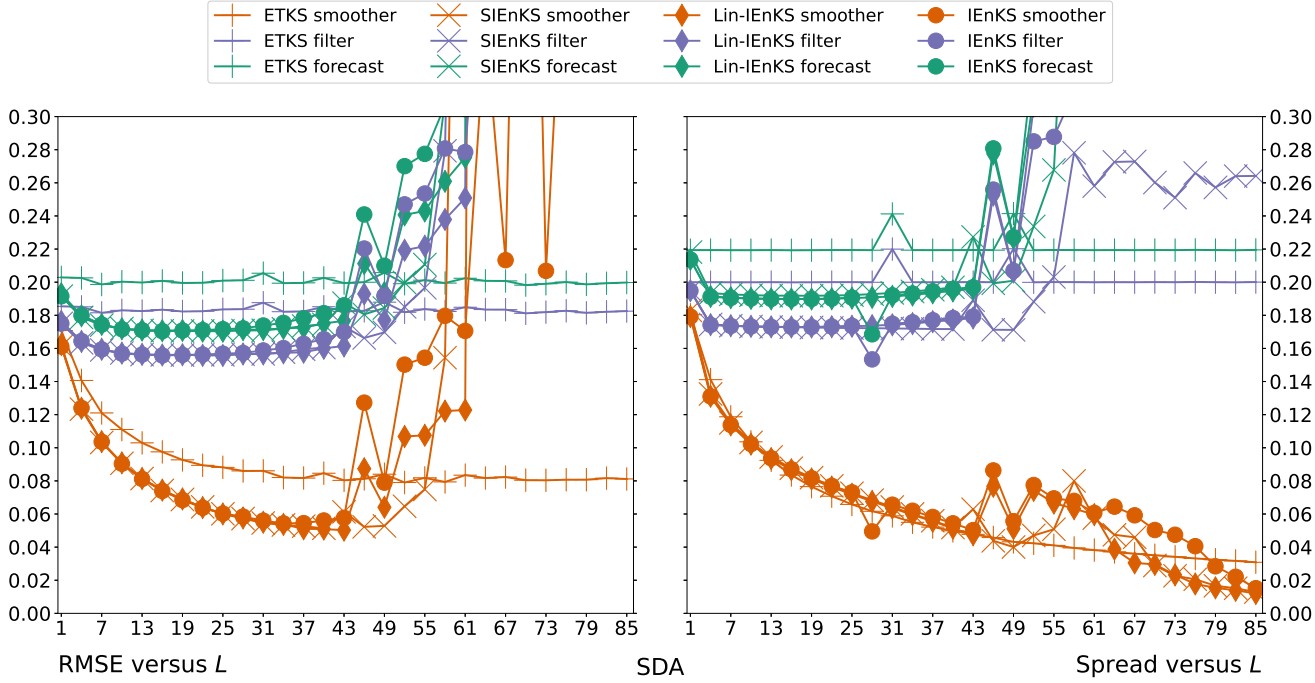

**Figure 6.** Cross section of Fig. 5 at ensemble size $N_e = 21$.

dynamics, typical of mid-range forecasts. The efficacy of the design is illustrated as the scheme is most effective when the low ensemble size and nonlinear forecast error dynamics conditions are present. Note that the Lin-IEnKS-N uses a single iteration of the extended 4D-MAP cost function, optimizing both the weights for the initial condition and the hyper-parameter

simultaneously. On the other hand, while the SIEnKS-N makes a single iteration of the ensemble simulation over the DAW, it is free to iteratively optimize the adaptive inflation hyper-parameter in the filter cost function. This iterative optimization of the filter cost function allows the SIEnKS-N to make substantial improvements over the Lin-IEnKS-N in terms of the stability region for the prediction accuracy, while remaining at the same leading order cost. Unlike the SIEnKS-N and the IEnKS-N, the Lin-IEnKS-N is unable to reach even the prediction accuracy of the EnKS-N for low sample sizes.

Figure 8 plots a cross section of Fig. 7 at $N_e = 21$ in order to further demonstrate the improved accuracy of the forecast / filter / smoother statistics of the SIEnKS-N versus the Lin-IEnKS-N. For a tuned lag $L$, the Lin-IEnKS-N fails to achieve distinctly better forecast and filter accuracy than the EnKS-N. While the smoother RMSE for the Lin-IEnKS-N does make an improvement over the EnKS-N, this improvement is not comparable to the smoother accuracy of the SIEnKS-N, which has the same leading order cost. The performance of the SIEnKS-N is almost indistinguishable from the fully iterative IEnKS-N

up to a lag of $L \approx 25$. At this point, the stability of the SIEnKS-N begins to suffer, while on the other hand the IEnKS-N is able to improve smoother RMSE for slightly longer lags. Nonetheless, both the SIEnKS-N and the IEnKS-N become



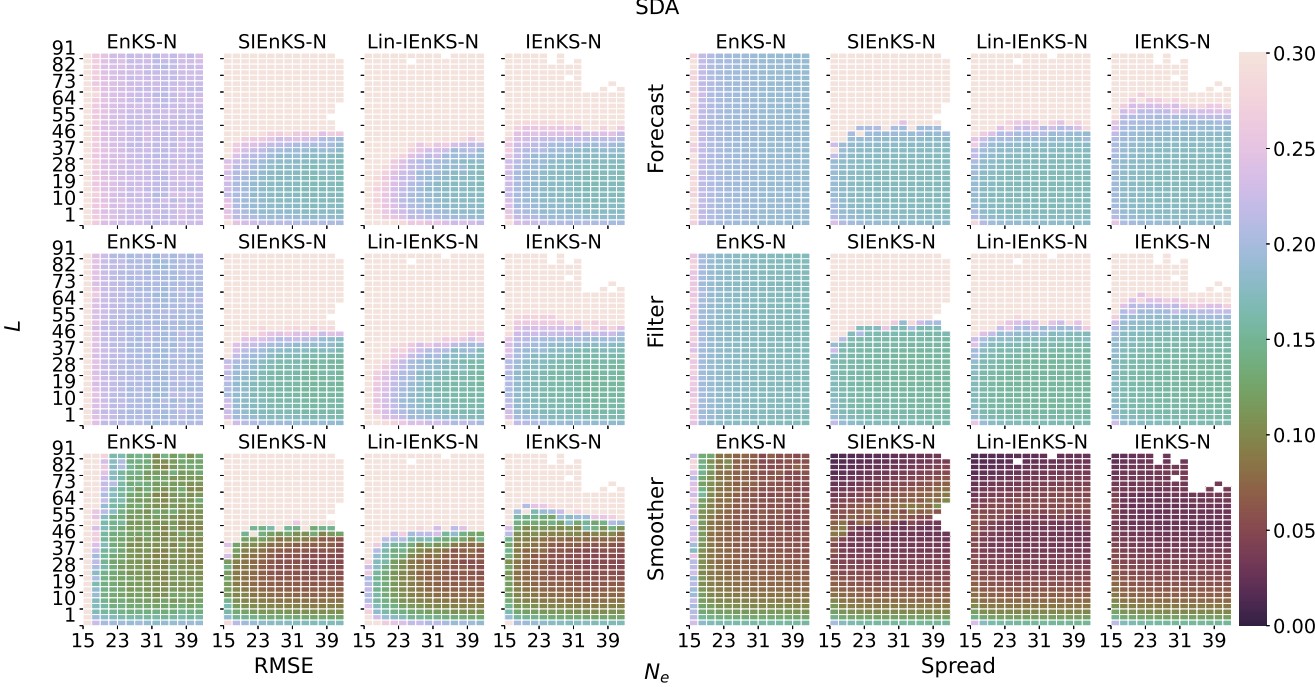

**Figure 7.** Lag length $L$ vertical axis, ensemble size $N_e$ horizontal axis. SDA, adaptive inflation, shift $S = 1$, linear observations, $\Delta t = 0.05$.

increasingly under-dispersive for lags $L \geq 25$, demonstrating systematic underestimation of the estimator's uncertainty that leads to divergence for sufficiently large $L$.

We now use MDA to relax the nonlinearity of the MAP estimation and the interpolation of the posterior over the DAW. Recall that MDA is handled differently in the SIEnKS from the 4D-MAP schemes: the 4D-MAP approach interpolates the DAW with the balancing estimate from a free forecast, while the SIEnKS interpolates the posterior estimate via a sequence of EnKS filter steps using the balancing weights. Likewise, recall that the SIEnKS is the least expensive MDA estimator, requiring only $2L$ ensemble simulations in this configuration, while the (Lin-)IEnKS uses at least $2L + 1$. Figure 9 presents the same experiment configuration as in Figs. 5 and 7, but where MDA is used with tuned inflation. The EnKS does not use MDA, but the results from Fig. 5 are presented here for reference.

It is easy to see that MDA improves all of the iterative smoothing schemes in Fig. 9, with greatly expanded stability regions from Fig. 5. Moreover, a key new pattern emerges that differentiates MDA using a retrospective analysis as in the SIEnKS. In particular, while the stability regions for the SIEnKS / (Lin-)IEnKS are similar for their smoother statistics in this configuration, the forecast / filter statistics are strongly differentiated. Unlike the free forecast solution used to interpolate the posterior over the DAW in the 4D-MAP approach, the filter step within the SIEnKS controls the simulation errors that accumulate when $L$ is large.



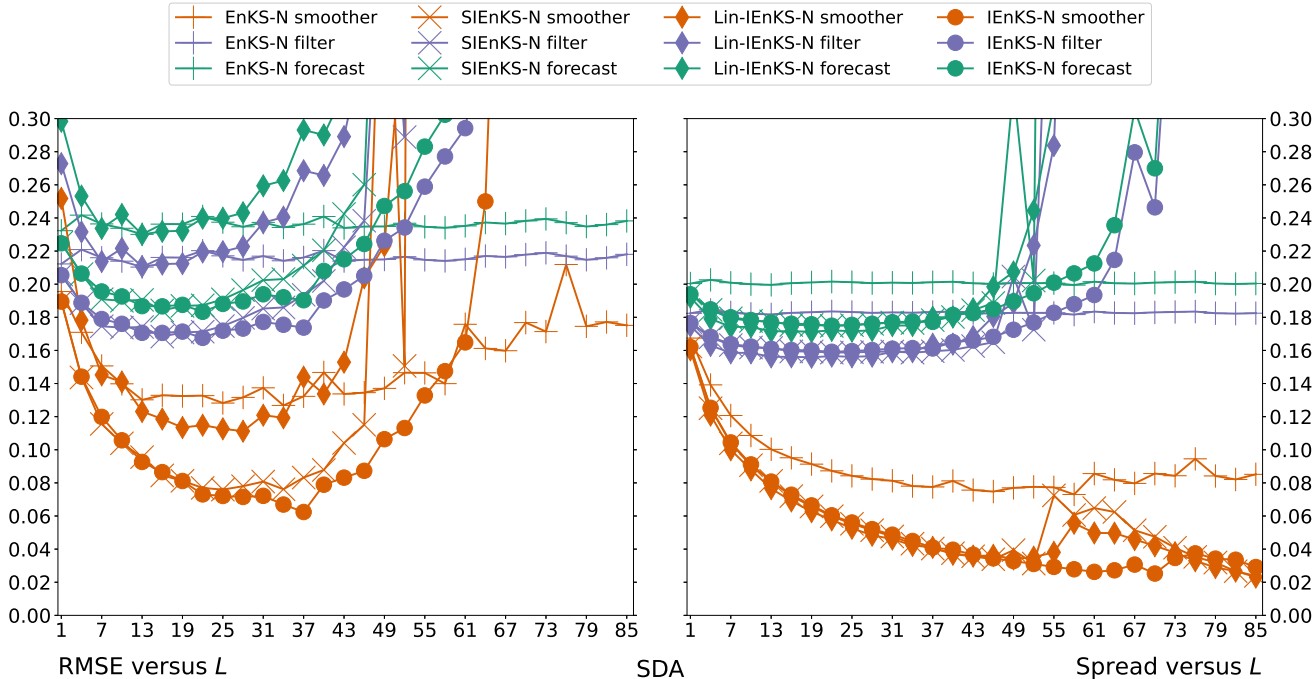

**Figure 8.** Cross section of Fig. 7 at ensemble size $N_e = 21$.

In order to examine the effect more precisely, consider the cross section of Fig. 9 for $N_e = 21$ presented in Fig. 10. Notice that all iterative MDA estimators have almost indistinguishable performance until lag $L \approx 31$. From this point, although the smoother accuracy increases with longer lags for the (Lin-)IEnKS, this comes at a sacrifice in the forecast / filter accuracy. 915 Particularly, for lags $L \geq 31$ the forecast / filter accuracy of the (Lin-)IEnKS begins to degrade; at a lag of $L \approx 61$, the IEnKS performs worse than the EnKS, while the Lin-IEnKS has diverged. This is in stark contrast to the SIEnKS — not only does the forecast / filter accuracy remain stable for lags $L \geq 40$, but each of these improve along with the smoother accuracy until a lag $L \approx 61$. Furthermore, the spread of the SIEnKS indicates that the SIEnKS MDA, perfect, linear-Gaussian approximation is well-satisfied, with the ensemble dispersion very close to the RMSE within the stability region.

The SIEnKS then highlights a performance trade off of the 4D-MAP MDA schemes that the SIEnKS does not itself suffer from. In particular, suppose that the lag $L$ in Fig. 9 is selected in order to optimize each estimator's accuracy in terms of RMSE, for each fixed ensemble size $N_e$. One can optimize the lag $L$ using the forecast RMSE or the smoother RMSE as the criterion. However, Fig. 10 indicates that $L$ may be quite different for the forecast accuracy versus the smoother accuracy in the 4D-MAP schemes. Figures 11 and 12 demonstrate this trade off precisely, where the former plots the RMSE and spread with lag and 925 inflation simultaneously optimized for forecast accuracy and the later is optimized for smoother accuracy.

Tuning for optimum forecast RMSE, as in Fig. 11, the performance of the SIEnKS / (Lin-)IEnKS for any fixed $N_e$ are indistinguishable with respect to this metric. On the other hand, the SIEnKS strongly outperforms the Lin-IEnKS and even



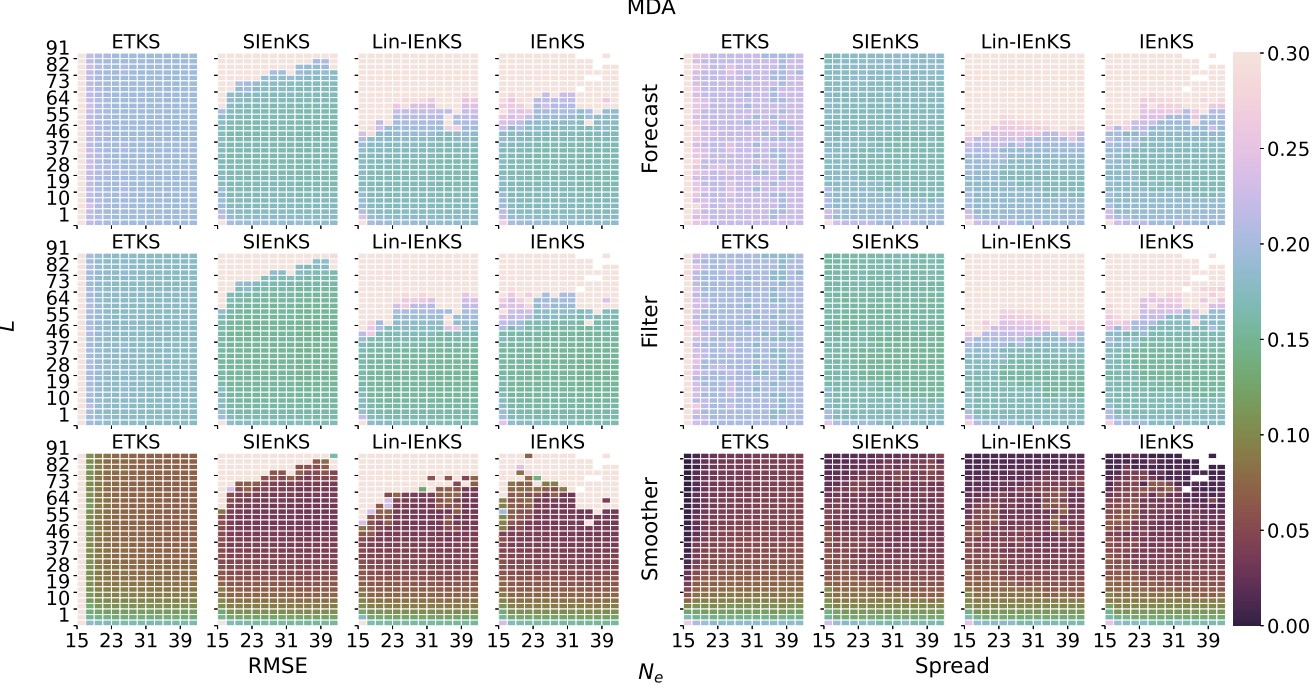

**Figure 9.** Lag length $L$ vertical axis, ensemble size $N_e$ horizontal axis. MDA, tuned inflation, shift $S = 1$, linear observations, $\Delta t = 0.05$. EnKS SDA results presented here for reference.

exhibits a slightly better overall stability and accuracy than the IEnKS across the range of ensemble sizes. The difference in performance is more pronounced when tuning for the minimal smoother RMSE in Fig. 12. Again, the three estimators are
indistinguishable in their smoother estimates, but the SIEnKS forms high-precision smoother estimates without sacrificing its predictive accuracy while interpolating the solution over long lags.

Using MDA or adaptive inflation in DA cycles with weakly nonlinear forecast error dynamics, we demonstrate how the SIEnKS greatly outperforms the Lin-IEnKS with the same, or lower, leading order cost. The SIEnKS MDA scheme also outperforms the IEnKS MDA scheme with less cost, but the fully iterative IEnKS-N is able to extract additional accuracy over
the SIEnKS-N at the cost of $L$ additional ensemble simulations per iteration. Therefore, it is worth considering the statistics on the number of iterations that the IEnKS uses in each of the above studied configurations. Figure 13 shows a heat plot for the mean and the standard deviation of the number of iterations used per cycle for each of the IEnKS with SDA, IEnKS-N and IEnKS with MDA to optimize the 4D-MAP cost function. Notice that in the MDA configuration, the mean and the standard deviation is computed over the two stages of the IEnKS, accounting for both the balancing and MDA 4D-MAP cost functions.
Although the number of possible iterations used is bounded below by one in the case of SDA, and two in the case of MDA, the frequency distribution for the total iterations is not especially skewed within the stability region of the IEnKS. This is evidenced by the small standard deviation, less than or equal to one, that defines the stability region for the scheme. Particularly, the IEnKS typically stabilizes around: (i) three iterations in the SDA, tuned inflation configuration; (ii) three to four iterations in

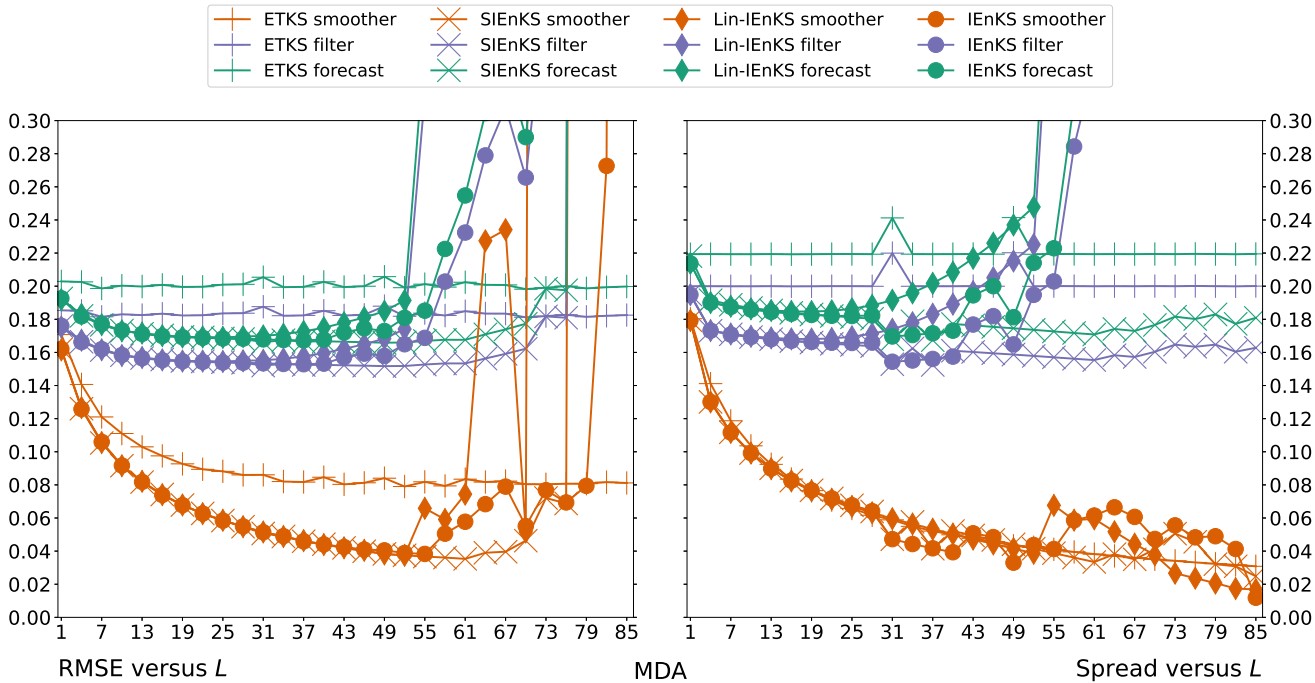

**Figure 10.** Cross section of Fig. 9 at ensemble size $N_e = 21$.

the SDA, adaptive inflation configuration; and (iii) six to eight iterations in the MDA, tuned inflation configuration. Therefore,

the SIEnKS is shown to make a reduction ranging between: (i) $2L$; (ii) $2L$ to $3L$; or $4L$ to $6L$ ensemble simulations of the estimator's cycle, on average, versus the IEnKS. While this is unremarkable for the SDA, tuned inflation configuration where the Lin-IEnKS performs similarly, this demonstrates a strong performance advantage of the SIEnKS in its target application, i.e., in settings with weakly nonlinear forecast error dynamics and other sources of nonlinearity dominating the DA cycle. This an especially profound reduction for the MDA configuration, where the SIEnKS MDA scheme proves to be both the least

expensive and the most stable / accurate estimator by far.

### 5.4 Weakly nonlinear forecast error dynamics – nonlinear observations

A primary motivating application for the SIEnKS is the scenario where the forecast error dynamics are weakly nonlinear but where the observation operator is weakly to strongly nonlinear. There are infinitely many possible ways how nonlinearity in the observation operator can be expressed, and the results are expected to strongly depend on the particular operator. In the

955 following, we consider the operator in Eq. (100) for the ability to tune the strength of this effect with the parameter $\gamma$. In order to avoid conflating the effect of the nonlinearity in the hyper-parameter optimization and the nonlinearity in the observation operator, we suppress adaptive inflation in this section. In this case, SDA and MDA schemes are considered to compare how MDA can be used to temper the effects of local minima in the MAP estimation versus a nonlinear observation operator.



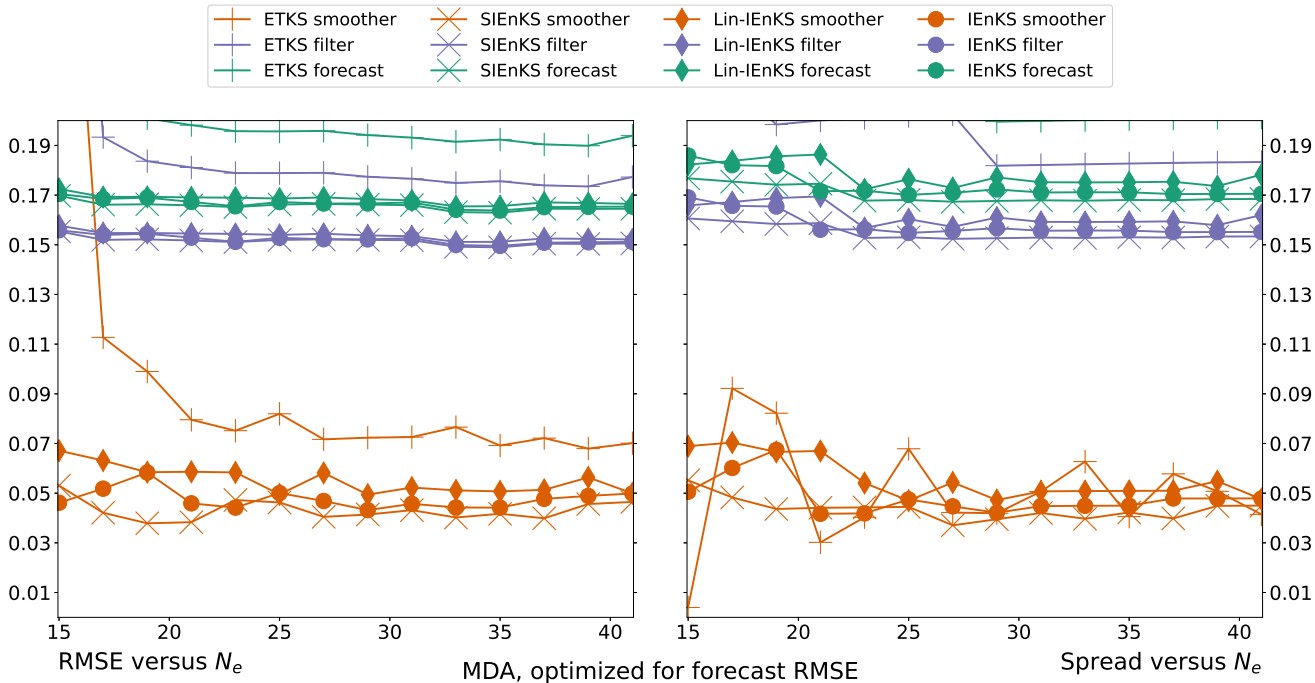

**Figure 11.** MDA, RMSE and spread versus ensemble size $N_e$, lag and inflation optimized for minimum forecast RMSE in Fig. 9.

We again choose $\Delta t = 0.05$ to maintain weakly nonlinear forecast error dynamics. We restrict to $N_e = 21$ as we expand the

960 experimental parameters to include $\gamma$. The lag is varied as $L \in \{1 + 3i\}_{i=0}^{27}$.

Figure 14 demonstrates the effect of varying the nonlinearity in the observation operator, where strong differences once again emerge between the retrospective analysis of the MLES and the iterative schemes. The scale of the color map is raised to a max of $0.5$, as a better performance can be achieved with the MLEF alone, as demonstrated by the MLES. In the MLES, the predictive and posterior error increases almost uniformly in $\gamma$, but a very different picture emerges for the iterative smoothers.

While the stability regions of the iterative schemes tend to shrink for larger $\gamma$, the accuracy of the estimators changes non-monotonically. Moreover, iteratively optimizing the filter cost function in the SIEnKS or the 4D-MAP cost function in the IEnKS does not in and of itself guarantee a better performance than the Lin-IEnKS, due to the increasing presence of local minima. Particularly for the SIEnKS and the IEnKS with highly nonlinear observations, this optimization can also become deleterious to the estimator performance, with evidence of instability and catastrophic divergence in these regimes.

In Fig. 15, we repeat the experimental configuration of Fig. 14 with the exception of using the MDA configuration. As seen in Fig. 9, MDA greatly extends the forecast / filter accuracy of the SIEnKS over the 4D-MAP schemes. Multiple data assimilation in this context additionally weakens the effect of the assimilation step on the model simulation, smoothing the cost function contours and expanding the stability regions of all estimators.

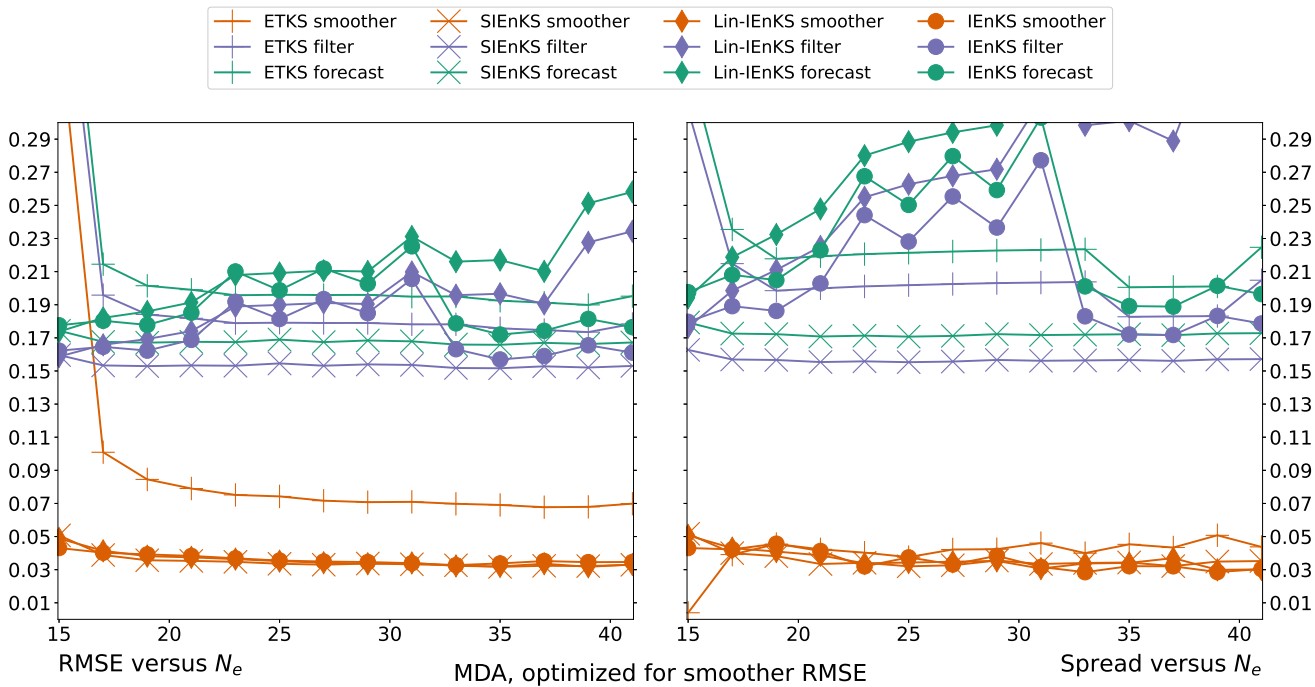

**Figure 12.** MDA, RMSE and spread versus ensemble size $N_e$, lag and inflation optimized for minimum smoother RMSE in Fig. 9.

Figure 16 presents tuned results from Fig. 15 where the lag and inflation are simultaneously optimized for the minimal
forecast RMSE. In this context, we clearly see how the effect of varying $\gamma$ is non-monotonic on the estimator accuracy for the
iterative schemes. However, important differences also emerge in this configuration between the SIEnKS and the (Lin-)IEnKS.
While the forecast and filter accuracy of these schemes remains indistinguishable for $\gamma \leq 7$, the smoother RMSE of the SIEnKS
is almost uniformly lower than these other schemes for all $\gamma$. Interestingly, the degradation of the performance of the IEnKS
for highly nonlinear observations, $\gamma \geq 8$, does not extend to either of the Lin-IEnkS or the SIEnKS in the MDA configuration.
Whereas the iterative optimization of the 4D-MAP cost function becomes susceptible to the effects of the local minima with
large $\gamma$, the Lin-IEnKS remains stable for the full window of the $\gamma$ presented here. Moreover, the SIEnKS demonstrates
significantly improved smoother accuracy over the Lin-IEnKS while remaining at a lower leading order cost. This suggests
that the sequential MDA scheme of the SIEnKS is better equipped to handle highly nonlinear observation operators than the
4D-MAP formalism, which appears to suffer from a greater number of local minima.

**5.5    Weakly nonlinear forecast error dynamics – lag versus shift**

Even for a linear observation operator and tuned inflation, a shift $S > 1$ distinguishes the performance of each of the studied
estimators. In this section, we fix $\Delta t = 0.05$ corresponding to weakly nonlinear forecast error dynamics and we vary $L, S \in$
$\{2, 4, 8, 16, 32, 48, 64, 80, 96\}$ to demonstrate these differences. For the iterative schemes, we only consider combinations of $L$



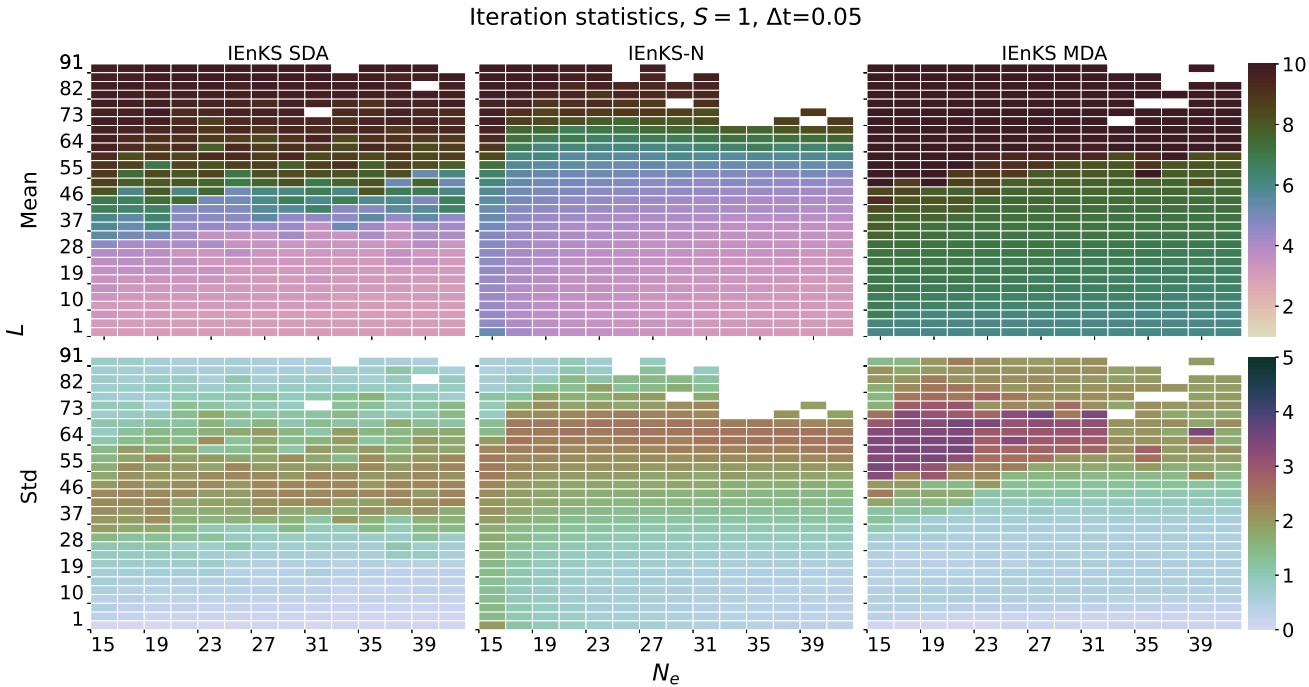

**Figure 13.** Iterations per cycle versus lag $L$ in the vertical axis and ensemble size $N_e$ in the horizontal axis. Mean (top panel) and standard deviation (bottom panel) of iterations used per cycle from simulations generating Figs. 5, 7 and 9 are presented.

divisible by $S$ for compatibility with the MDA schemes. The EnKS is defined for arbitrary $S < L$, and all such configurations are presented for reference.

Recall the qualification that the EnKS and SIEnKS are designed to assimilate observations sequentially and synchronously in this work whereas the (Lin-)IEnKS assimilates observations asynchronously by default. When $S = 1$ there is no distinction between asynchronous and synchronous assimilation, but in this section this distinction is borne in mind. Likewise, it is recalled that for the (Lin-)IEnKS with a shift $S > 1$, filter statistics are computed including the information from all observations $\boldsymbol{y}_{L:L-S+1}$ when making a filter estimate for states at times $t_{S+1}, \cdots, t_L$. This arises from the asynchronous design of the IEnKS, whereas filter statistics are computed sequentially without future information in the SIEnKS.

Figure 17 presents the heat plot of RMSE and spread for each estimator in the SDA configuration. We note that the EnKS for a fixed $L$ has performance that is largely invariant with respect to changes in $S$, except for the special case where $S = L$. In this case, the non-overlapping DAWs impose that posterior estimates are constructed with fewer observations conditioning the final estimate than in overlapping DAWs. Otherwise, the stability regions of the iterative schemes are largely the same, with the SIEnKS only achieving a slight improvement over the Lin-IEnKS, and the IEnKS only slightly improving on the SIEnKS.

The SDA configuration is contrasted with Fig. 18 where we again see the apparent strengths of the SIEnKS MDA scheme. When MDA is introduced, all iterative schemes increase their respective stability regions to include longer lags and larger shifts of the DAW simultaneously. However, the SIEnKS has the largest stability region of all iterative estimators, extending

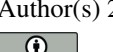



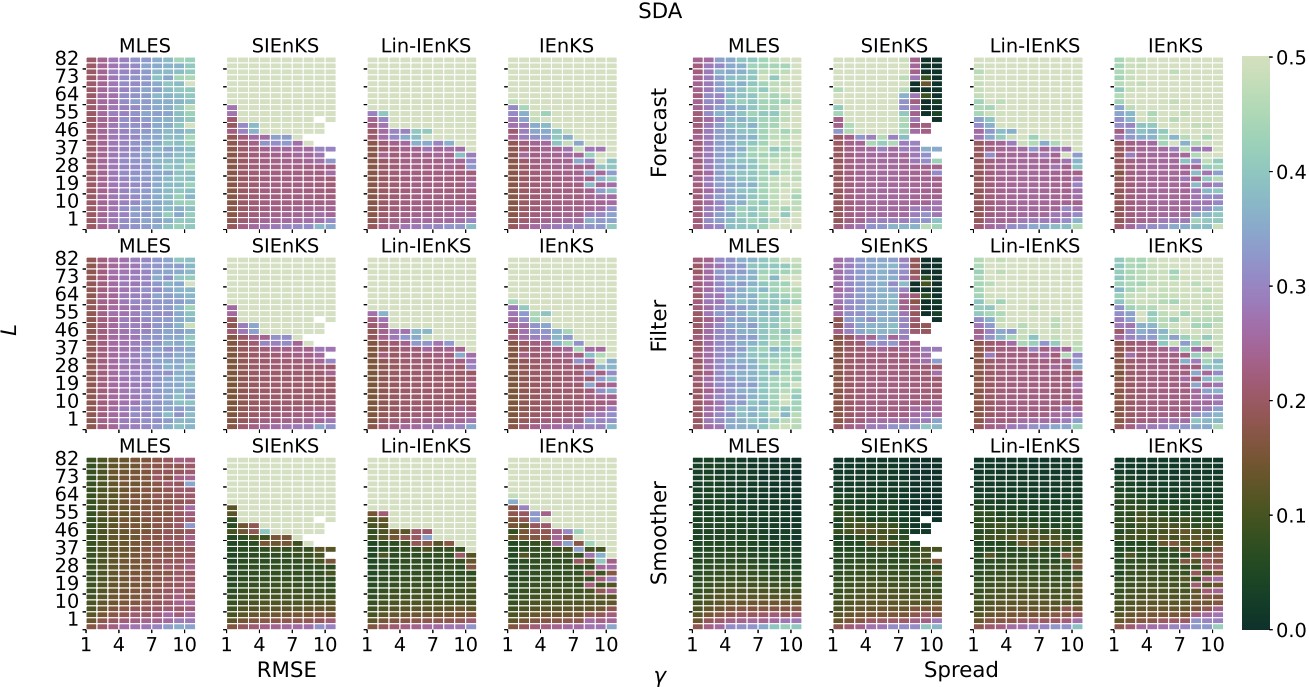

**Figure 14.** Lag length $L$ vertical axis, nonlinearity parameter $\gamma$ horizontal axis. SDA, tuned inflation, shift $S = 1$, $N_e = 21$ and $\Delta t = 0.05$.

to at least as large shifts as the other schemes for every lag setting. Likewise, the earlier distinction between the forecast and filter statistics of the SIEnKS and the 4D-MAP schemes is readily apparent. Not only does the stability region of the SIEnKS improve over the other schemes, it is is also generally more accurate in its predictive statistics at the end of long lag windows.

In order to get a finer picture of the effect of varying the shift $S$, we tune the lag and inflation simultaneously for each estimator for their minimal forecast RMSE given a fixed shift; we plot the results of this tuning in Fig. 19. Given that all iterative estimators uniformly diverge for a shift $S \geq 32$, we only plot results for shifts in the range $\{2^i\}_{i=0}^4$. Several important features stand out in this plot. Firstly, we note that optimizing the lag, the performance of the SIEnKS is almost invariant in the shift, similar to the performance of the EnKS. Particularly, this is because the sequential filter analysis of the SIEnKS constrains the growth of the filter and forecast errors as the DAW shifts. Indeed, prediction of states at times $t_{L-S+1}, \cdots, t_L$ arise from a filter ensemble at the previous time point; in the MDA scheme, the balancing weights for the observations of these newly introduced states in the DAW are, furthermore, all equal to one, equivalent to a standard ETKF filter analysis.

Secondly, we notice that the filter estimates of the (Lin-)IEnKS actually improve with larger shifts – it should be noted, however, that this is partially an artifact of computing the filter statistics over all times $t_{L-S+1}, \cdots, t_L$ using the observations $\boldsymbol{y}_{L:L-S+1}$ simultaneously. This means that the filter estimates for all times except $t_L$ actually contain future information. This is contrasted with the sequential analyses of the EnKS and the SIEnKS which only produce filter statistics with observations from past and current times.



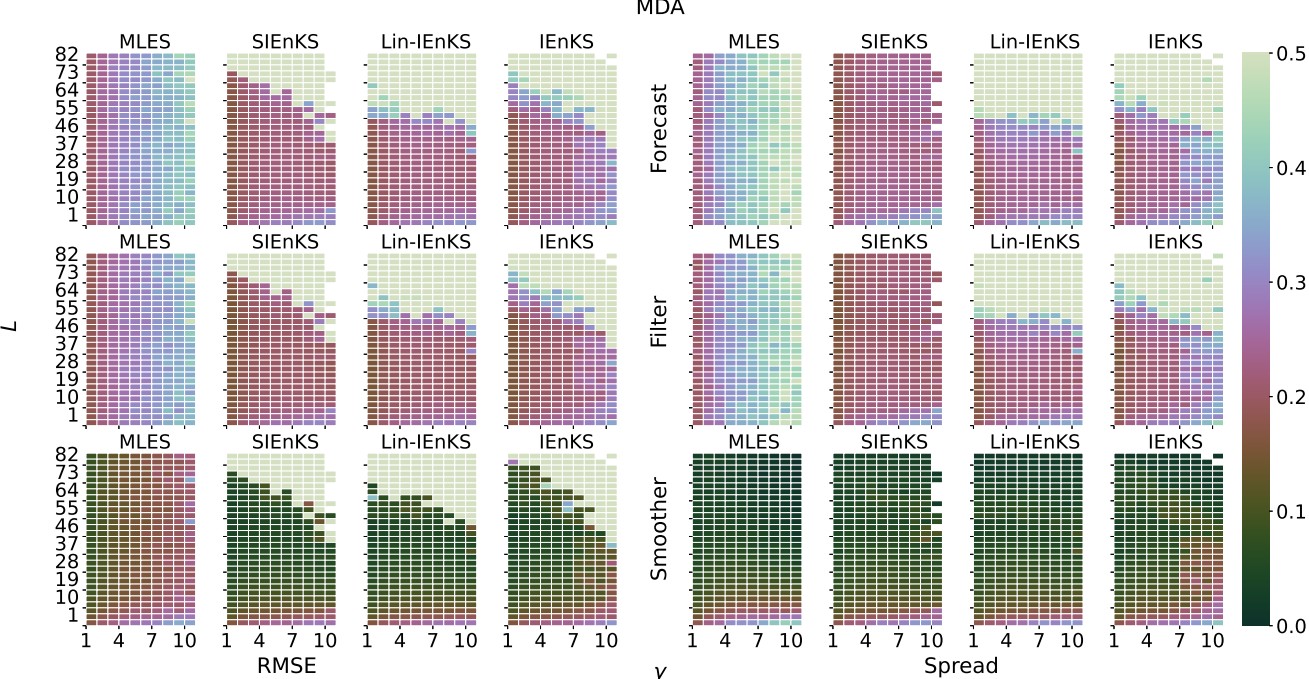

**Figure 15.** Lag length $L$ vertical axis, nonlinearity parameter $\gamma$ horizontal axis. MDA, tuned inflation, shift $S = 1$, $N_e = 21$ and $\Delta t = 0.05$. MLES SDA results presented here for reference.

Thirdly, we notice that the Lin-IEnKS, while maintaining a similar prediction and filtering error to the IEnKS, is less stable and performs almost uniformly less accurate than the IEnKS in its smoothing estimates. The SIEnKS, moreover, tends to exhibit a slight improvement in stability and accuracy over the IEnKS therein.

Finally, it is immediately apparent how $S > 1$ strongly increases the prediction error for the 4D-MAP estimators. The longer free forecasts for $S > 1$, used to shift the DAW, accumulate errors such that, for $S \geq 16$, the Lin-IEnKS actually experiences filter divergence. The difference in the estimators' performances is once again a consequence of how observations are assimilated synchronously as in the EnKS / SIEnKS or asynchronously by default in the (Lin-)IEnKS.

Bearing all the above qualifications in mind, we analyze the performance of the estimators while varying the shift $S$. Firstly, for all experimental settings the leading order cost of the SIEnKS MDA scheme is fixed at $2L$ ensemble simulations, whereas for the other schemes the minimal cost is at $2L + S$ ensemble simulations. For configurations where $S > 1$, the SIEnKS thus makes a dramatic cost reduction versus the other schemes in this aspect alone, requiring fewer ensemble simulations per cycle. We consider that the leading order cost for the Lin-IEnKS is similar to the SIEnKS for $S = 1$, requiring only one more ensemble simulation per cycle. However, the SIEnKS with a shift $S = 16$ maintains a prediction and posterior error that is comparable to the Lin-/IEnKS for a shift of $S = 1$. This implies that the SIEnKS can maintain performance similar to the $S = 1$ IEnKS MDA scheme, while using one sixteenth of the total cycles needed by the IEnKS to pass over the same observations in real-time. If we assume that the observations can be assimilated synchronously, the above SIEnKS MDA scheme is thus able to run in



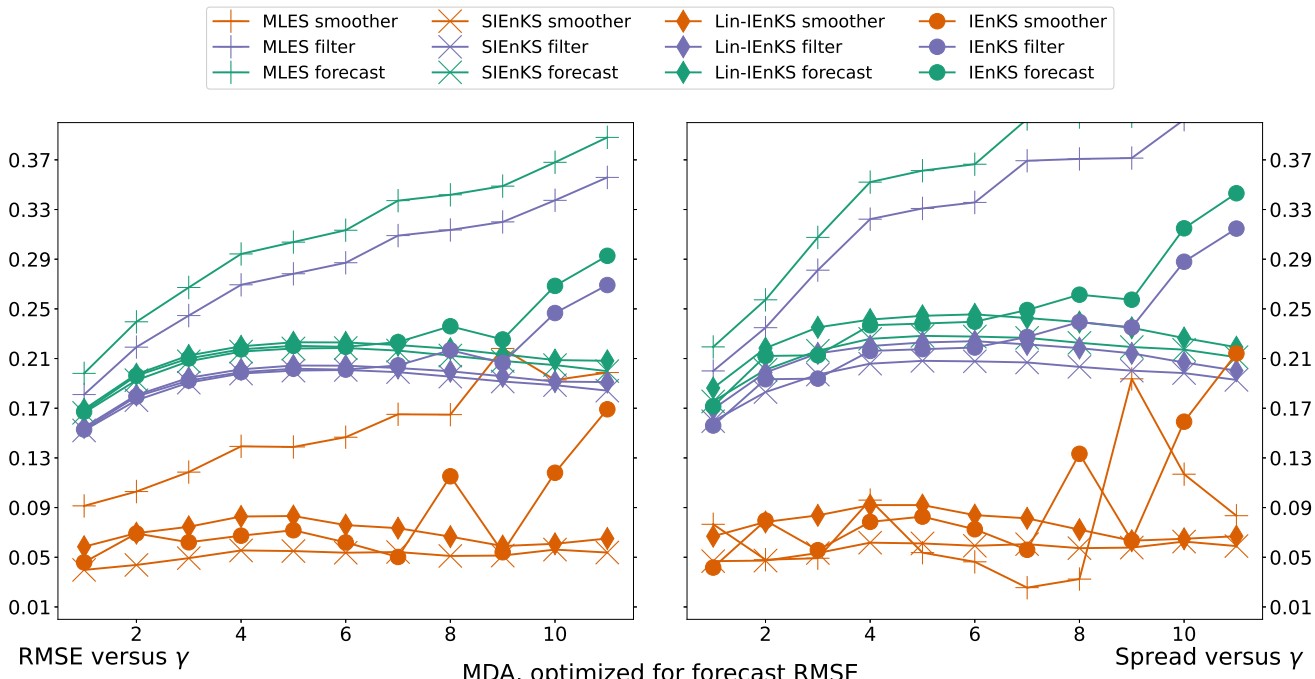

**Figure 16.** MDA, RMSE and spread versus $\gamma$, lag and inflation optimized for minimum forecast RMSE.

its EnKS cycle over a long time series of observations while needing infrequent re-initalization with its smoothed estimates. For an online forecast cycle, where the computational cost / prediction accuracy trade off is the most important consideration, this once again demonstrates how the SIEnKS can balance this trade off, performing as well and often better than other fully iterative estimators with a substantially lower leading order cost. Not only is each cycle less expensive in the SIEnKS than in the (Lin-)IEnKS, but the SIEnKS reduces the number of required cycles by an order of magnitude.

### 5.6 Strongly nonlinear forecast error dynamics – lag versus $\Delta t$

In all other numerical benchmarks, we focus on the scenario that the SIEnKS is designed for – namely, DA cycles in which the forecast error evolution is weakly nonlinear. In this section, we instead demonstrate the limits of our single-iteration formalism when the forecast error dynamics dominate the nonlinearity of the DA cycle. Specifically, we vary $\Delta t \in \{0.05 \times i\}_{i=1}^{10}$ while the ensemble size $N_e = 21$ and the shift $S = 1$ are fixed. The lag is varied as $L \in \{1 + 3i\}_{i=0}^{17}$. We neglect nonlinear observation operators in this section, though we include the finite-size adaptive inflation formalism, which is itself designed to ameliorate the increasing nonlinearity in the forecast error dynamics. Single data assimilation and MDA configurations are considered for the iterative schemes as usual.

Figure 20 demonstrates the effect of the increasing nonlinearity of the forecast error evolution with tuned inflation. Due to the extreme nonlinearity for large $\Delta t$, we raise the heat map scale for the RMSE and spread to $1.0$, as an absolute cut-off between

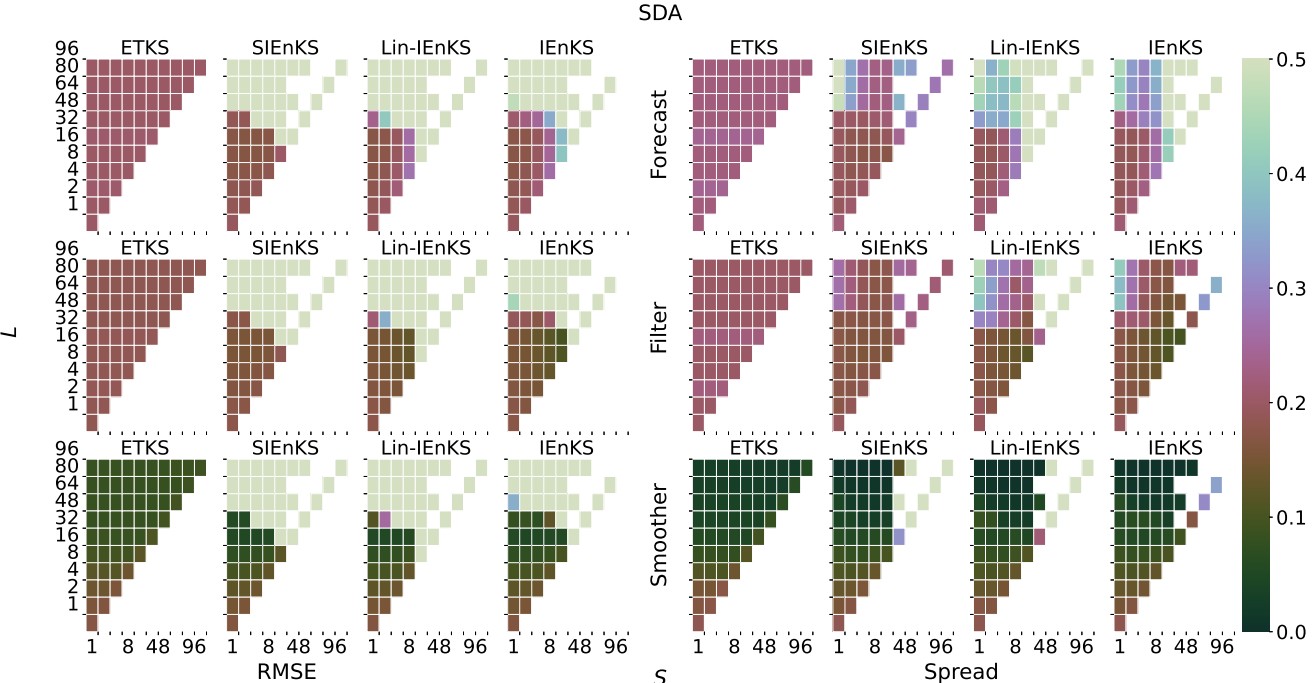

**Figure 17.** Lag length $L$ vertical axis, shift $S$ horizontal axis. SDA, tuned inflation, linear observations, ensemble size $N_e = 21$, $\Delta t = 0.05$.

acceptable filter performance and filter divergence. Several features become apparent with the increasing forecast nonlinearity. Firstly, the EnKS, which has performance dependent on the standard ETKF cycle, is fully divergent for $\Delta t \geq 0.2$. This is contrasted with all iterative schemes which maintain adequate performance for $\Delta t \leq 0.25$. We note that the performance of the SIEnKS and the Lin-IEnKS, in this first scenario, are nearly identical; this corresponds to the fact that they are formally equivalent in this setting. However, appropriately, it is the fully iterative IEnKS that maintains the most stable and accurate performance over the range of forecast lengths. Indeed, this demonstrates the benefit precisely of the iterative solution to 4D-MAP cost function for moderately nonlinear, non-Gaussian DA.

In Fig. 21, we repeat the same experiments as in Fig. 20 but using the finite-size adaptive inflation, rather than tuned inflation, for each estimator. Once again, the efficacy of the finite-size formalism in ameliorating the nonlinearity of the forecast error dynamics is demonstrated. In particular, all schemes except the SIEnKS see an overall improvement in their stability region and often in their overall accuracy. The EnKS-N actually strongly outperforms the tuned inflation EnKS, extending an adequate filter performance as far as $\Delta t \leq 0.35$. Likewise, the IEnKS-N has a strongly enhanced stability region, though increasingly suffers from catastrophic filter divergence outside of this zone. Notably, whereas the SIEnKS-N outperformed the Lin-IEnKS-N for $\Delta t = 0.05$, the Lin-IEnKS-N generally yields better performance for moderately to strongly nonlinear forecast error dynamics. Indeed, the finite-size formalism appears to become incompatible with the design of the SIEnKS for strongly nonlinear forecast error dynamics, as suggested by the widespread ensemble collapse and catastrophic divergence.

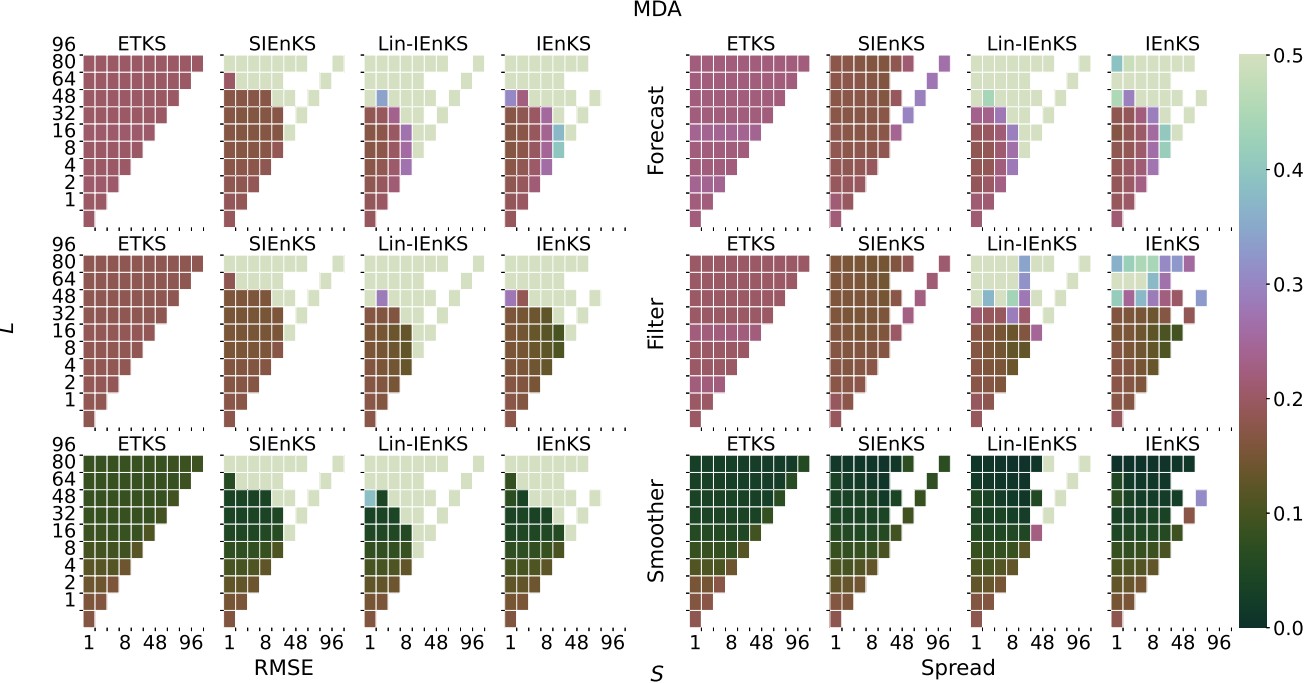

**Figure 18.** Lag length $L$ vertical axis, shift $S$ horizontal axis. MDA, tuned inflation, linear observations, ensemble size $N_e = 21$, $\Delta t = 0.05$. EnKS SDA results presented here for reference.

As a final experimental configuration, we consider how MDA affects the increasing nonlinearity of the forecast error dynamics. Figure 22 demonstrates the performance of these estimators in the MDA configuration with tuned inflation, where the SDA results of the EnKS are pictured for reference. In particular, we see the usual increase in the estimators' stability regions over the SDA configuration. However, the improvement of the SIEnKS over the Lin-IEnKS is marginal to non-existent for moderately to strongly nonlinear forecast error dynamics. The fully iterative IEnKS, furthermore, is again the estimator with the largest stability region and greatest accuracy over a wide range of $\Delta t$.

The results in this section indicate that, while the SIEnKS is very successful in weakly nonlinear forecast error dynamics, the approximations used in this estimator strongly depend on the source of nonlinearity in the DA cycle. Particularly, when the nonlinearity of the forecast error dynamics dominates the DA cycle, the approximations of the SIEnKS break down. It is favorable thus to consider the Lin-IEnKS, or setting a low threshold for the iterations in the IEnKS, instead of applying the SIEnKS in this regime. Notably, as the finite-size inflation formalism is designed for a scenario different than the SIEnKS, one may consider instead designing adaptive covariance inflation in such a way that it exploits the design principles of the SIEnKS. Such a study goes beyond the scope of this work and will be considered later.





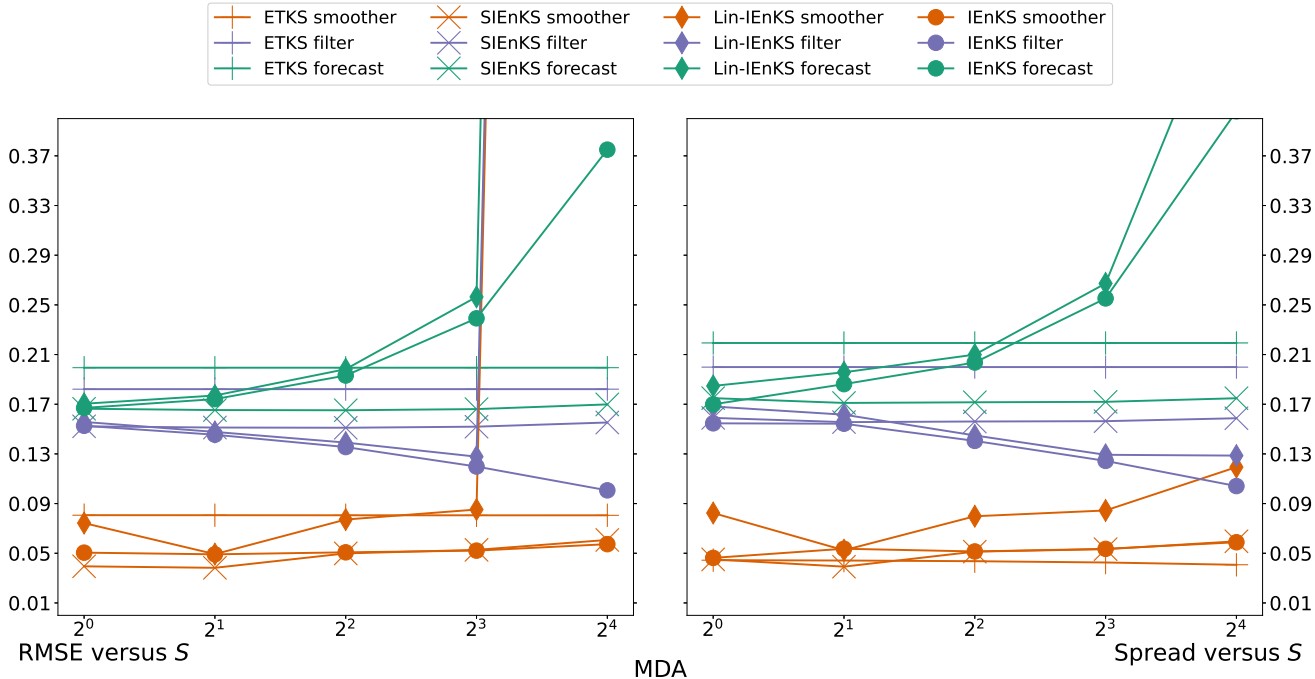

**Figure 19.** MDA, RMSE and spread versus shift $S$, lag $L$ optimized for minimum forecast RMSE in Fig. 18.

## 6 Conclusion

In this long work, we achieve our three primary objectives. Firstly, we provide a detailed review of the state-of-the-art for sequential, ensemble-variational Kalman filters and smoothers in perfect models within the Bayesian MAP formalism of the IEnKS. Secondly, using this framework, we rigorously derive our single-iteration formalism as a novel approximation of the Bayesian MAP estimation, explaining how this relates to other well-known smoothing schemes and how its design is differentiated in a variety of contexts. Thirdly, using the high-performance numerical framework of DataAssimilationBenchmarks.jl (Grudzien et al., 2021), we extensively demonstrate how the SIEnKS has a unique advantage in balancing the computational cost / prediction accuracy trade off in short-range forecast applications. Pursuant to this, we provide a cost analysis and pseudo-code for all of the schemes studied in this work, in addition to the open-source implementations available in the supporting Julia package. Together, this work provides a practical reference for a variety of topics at the state-of-the-art in ensemble-variational smoothing, which now includes our fully validated SIEnKS scheme.

The rationale of the SIEnKS is, once again, to optimize an iterative Bayesian MAP estimation in a cost-effective design for online, short-range forecast applications, where the forecast error dynamics are weakly nonlinear. The central result in this study is the novel SIEnKS MDA scheme, which not only improves the forecast accuracy and posterior stability in this regime, but also simultaneously reduces the leading order cost versus the 4D-MAP MDA schemes under consideration. This novel MDA scheme is demonstrated to produce significant performance advantages in the simple setting where there is a linear





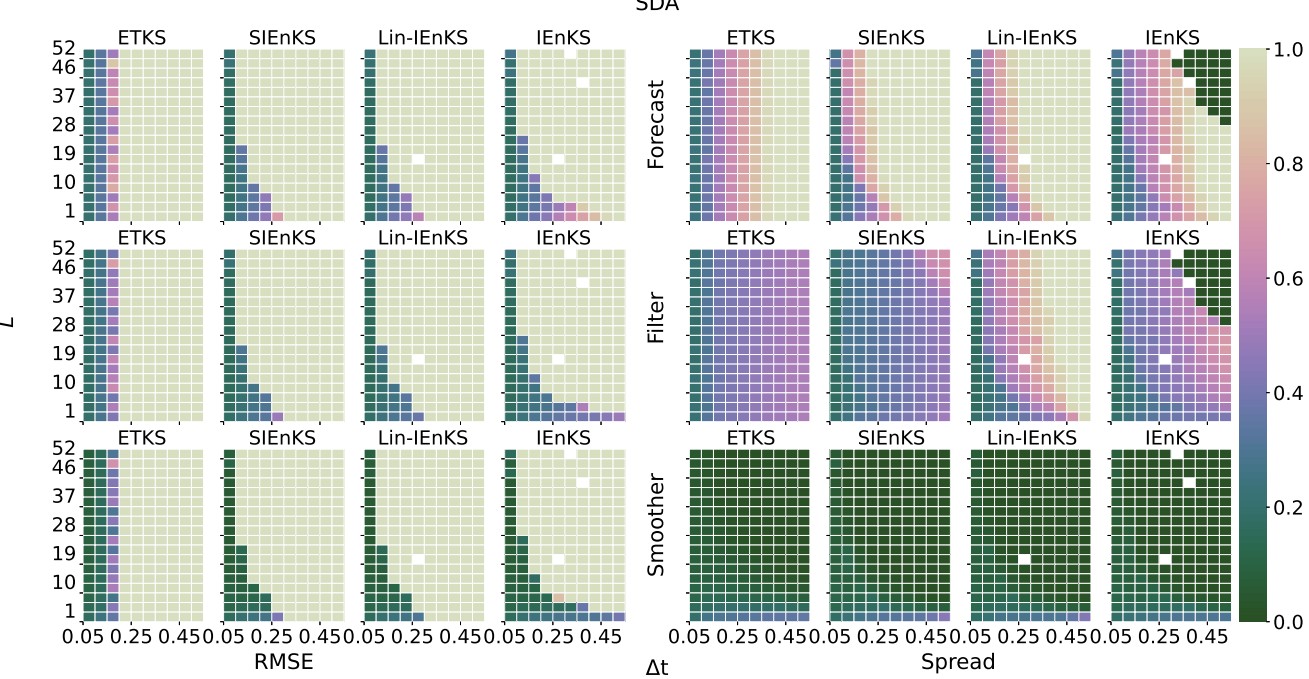

**Figure 20.** Lag length $L$ vertical axis, $\Delta t$ horizontal axis. SDA, tuned inflation, ensemble size $N_e = 21$.

observation operator, and especially when the shift $S$ can be taken greater than one. Not only is each cycle of the SIEnKS MDA scheme significantly less expensive than the other estimators for $S > 1$, the performance while varying $S$ tends to be invariant; this crucial aspect means that one can, in principle, reduce the number of cycles actually needed by the estimator to produce

forecasts in real-time. Our scheme also appears better equipped than the 4D-MAP MDA estimation to handle highly nonlinear observation operators, where it maintains greater accuracy and is more robust to the effects of local minima. Separately we find that, in our target regime, the single-iteration formalism is cost-effective for optimizing hyper-parameters of the estimation scheme, as with the SIEnKS-N.

The above successes of the SIEnKS come with three important qualifications, notably that: (i) we have focused on syn-

chronous DA in this study, assuming that we can sequentially assimilate observations before producing a prediction step; (ii) we have not studied localization or hybridization, which are widely used in ensemble-based estimators to overcome the curse of dimensionality for realistic geophysical models; and (iii) we have relied upon the perfect model assumption, whereas realistic forecast settings include significant modelling errors. These restrictions come by necessity, to limit the scope of an already lengthy study. However, we note that the SIEnKS is capable of asynchronous DA, as already discussed in Sec. 4.4. Likewise, it

is possible that some of the issues faced by the IEnKS in integrating localization / hybridization (Bocquet, 2016) may actually be ameliorated by the design principles of the SIEnKS. Similarly, it is possible that an extension of the single-iteration formalism could provide a novel alternative to other iterative ensemble smoothers designed for model error, such as the IEnKS-Q





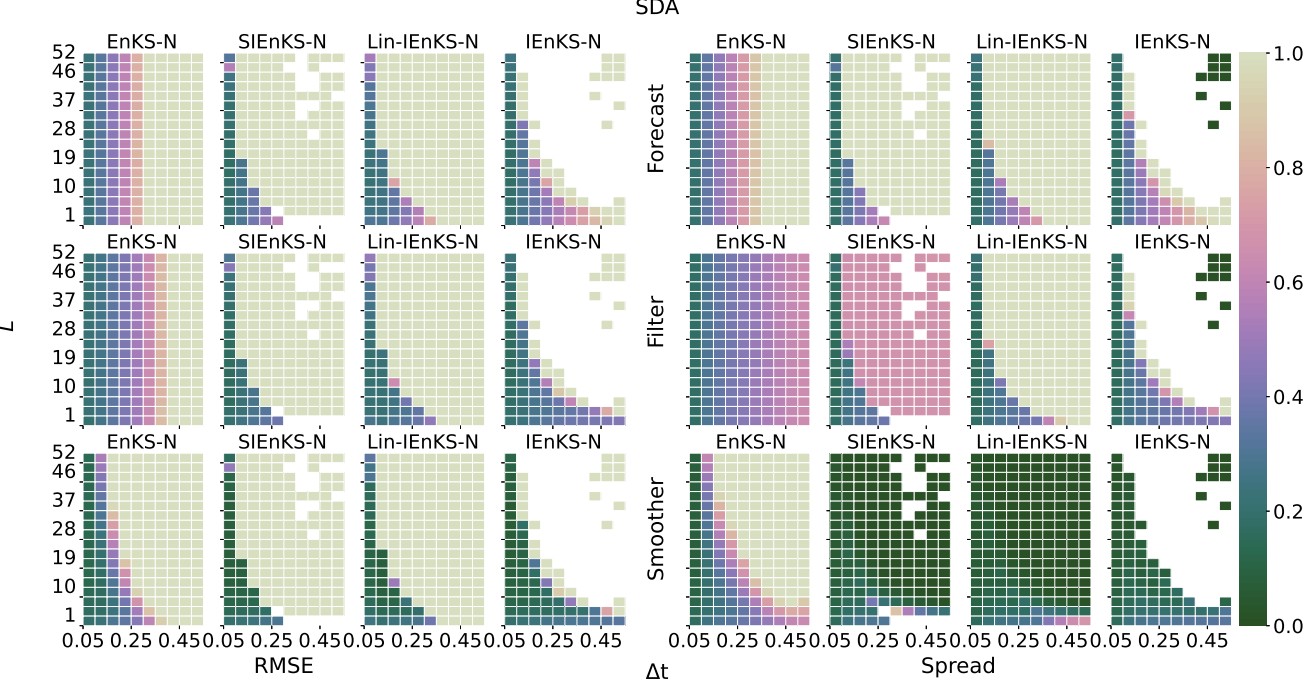

**Figure 21.** Lag length $L$ vertical axis, $\Delta t$ horizontal axis. SDA, adaptive inflation, ensemble size $N_e = 21$.

(Sakov et al., 2018; Fillion et al., 2020), EnKS expectation maximization schemes (Pulido et al., 2018) or the family of OSA smoothers (Gharamti et al., 2015; Ait-El-Fquih et al., 2016; Raboudi et al., 2018).

For the reasons mentioned above, this initial study provides a number of novel directions in which our single-iteration formalism can be extended. Localization and hybridization are both prime targets to translate the benefits of the SIEnKS to an operational short-range forecasting setting. Likewise, asynchronous DA design is an important operational topic for this estimator, with a variety of possible ways that one might design such a system. In addition, noting that the finite-size adaptive inflation formalism is designed to perform in a different regime than the SIEnKS and is not fully compatible with MDA

schemes, developing an adaptive inflation and / or model error estimation based on the design principles of the SIEnKS is an important direction of future study. Having currently demonstrated the initial success of this single-iteration formalism, each of these above directions can be considered in a devoted work. We intend that the framework provided in this manuscript will guide these future studies, and will provide a robust basis of comparison for the SIEnKS with other schemes at the state-of-the-art.



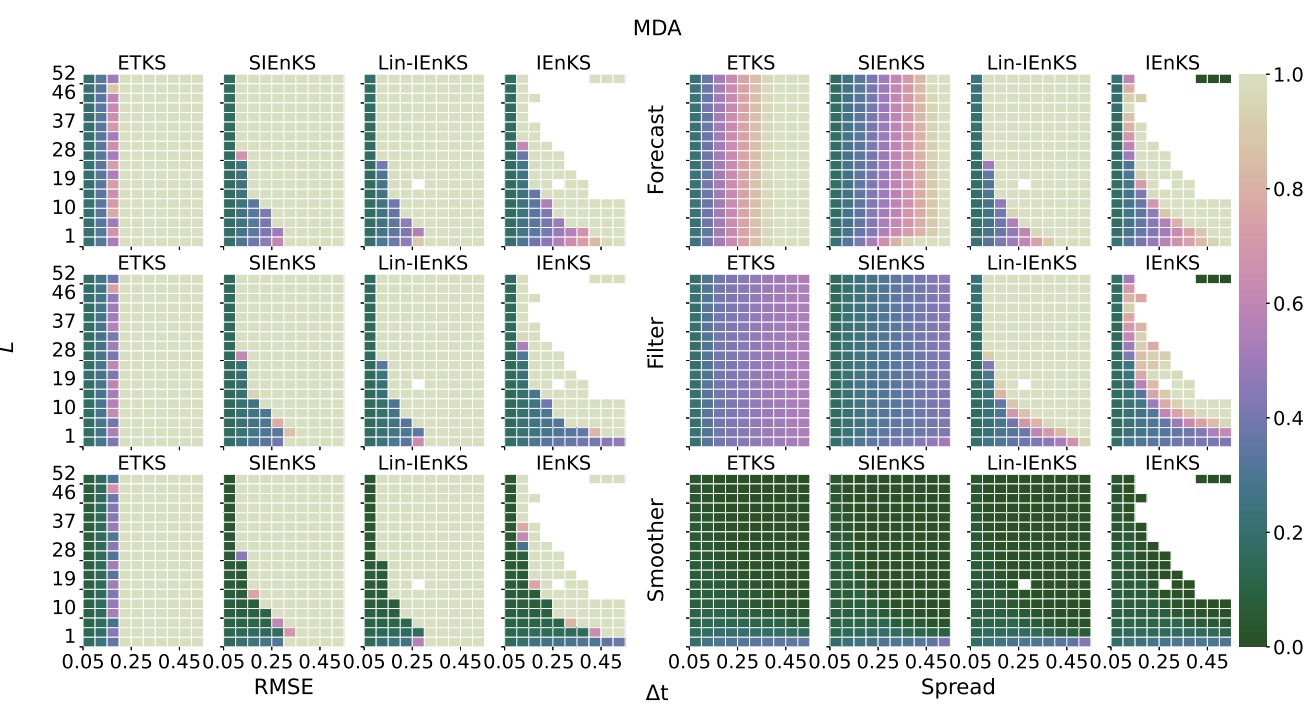

**Figure 22.** Lag length $L$ vertical axis, $\Delta t$ horizontal axis. MDA, tuned inflation, ensemble size $N_e = 21$.





**Appendix A: Algorithm pseudo-code**

---

**Algorithm 1** Ensemble transform (ET)

---

**Require:** Ensemble matrix $\mathbf{E} \in \mathbb{R}^{N_x \times N_e}$, observation map $\mathcal{H}$, observation error covariance $\mathbf{R} \in \mathbb{R}^{N_y \times N_y}$ and observation vector $\boldsymbol{y}$

1: $\mathbf{Y} = \mathcal{H}(\mathbf{E})$

2: $\hat{\boldsymbol{y}} = \mathbf{Y}\mathbf{1}/N_e$

3: $\mathbf{S} = \mathbf{R}^{-\frac{1}{2}}\left(\mathbf{Y} - \hat{\boldsymbol{y}}\mathbf{1}^\top\right)$

4: $\hat{\boldsymbol{\delta}} = \mathbf{R}^{-\frac{1}{2}}\left(\boldsymbol{y} - \hat{\boldsymbol{y}}\right)$

5: $\nabla\widetilde{\mathcal{J}} = -\mathbf{S}^\top\hat{\boldsymbol{\delta}}$

6: $\widetilde{\mathbf{H}}_{\widetilde{\mathcal{J}}} = (N_e - 1)\mathbf{I}_{N_e} + \mathbf{S}^\top\mathbf{S}$

7: $\boldsymbol{w} = -\widetilde{\mathbf{H}}_{\widetilde{\mathcal{J}}}^{-1}\nabla\widetilde{\mathcal{J}}$

8: $\mathbf{T} = \widetilde{\mathbf{H}}_{\widetilde{\mathcal{J}}}^{-\frac{1}{2}}$

9: **return** $\mathbf{T}, \boldsymbol{w}$

---

**Algorithm 2** Random mean-preserving orthogonal matrix (RO)

---

**Require:** Ensemble size $N_e$, let QR represents the QR algorithm.

1: Let $\mathbf{Q} \in \mathbb{R}^{(N_e-1)\times(N_e-1)}$ with entries drawn iid from $\mathcal{N}(0,1)$

2: $\mathbf{Q}, \mathbf{R} = \mathrm{QR}(\mathbf{Q})$

3: $\mathbf{U} = \begin{pmatrix} 1 & 0 \\ 0 & \mathbf{Q} \end{pmatrix}$

4: Let $\{\boldsymbol{a}_i\}_{i=1}^{N_e}$ be an arbitrary orthogonal basis of $\mathbb{R}^{N_e}$ up to the requirement that $\boldsymbol{a}_1 = \mathbf{1}/\sqrt{N_e}$; let $\mathbf{A} = [\boldsymbol{a}_i]_{i=1}^{N_e}$

5: **return** $\mathbf{U} = \mathbf{A}\mathbf{U}\mathbf{A}^\top$

---

**Algorithm 3** Ensemble update (EU)

---

**Require:** Ensemble matrix $\mathbf{E} \in \mathbb{R}^{N_x \times N_e}$, transform $\mathbf{T}$, weights $\boldsymbol{w}$ and mean-preserving orthogonal matrix $\mathbf{U}$.

1: $\hat{\boldsymbol{x}} = \mathbf{E}\mathbf{1}/N_e$

2: $\mathbf{X} = \mathbf{E} - \hat{\boldsymbol{x}}\mathbf{1}^\top$

3: **return** $\mathbf{E} = \hat{\boldsymbol{x}}\mathbf{1}^\top + \mathbf{X}\left(\boldsymbol{w}\mathbf{1}^\top + \sqrt{N_e - 1}\mathbf{T}\mathbf{U}\right)$

---





---

**Algorithm 4** Covariance inflation (CI)

---

**Require:** Ensemble matrix $\mathbf{E} \in \mathbb{R}^{N_x \times N_e}$, inflation $\lambda$.

1: $\hat{\boldsymbol{x}} = \mathbf{E}\mathbf{1}/N_e$

2: $\mathbf{X} = \mathbf{E} - \hat{\boldsymbol{x}}\mathbf{1}^{\top}$

3: **return** $\mathbf{E} = \hat{\boldsymbol{x}}\mathbf{1}^{\top} + \lambda\mathbf{X}$

---

**Algorithm 5** ETKF

---

**Require:** Observation $\boldsymbol{y}_1$, filter ensemble $\mathbf{E}_0^{\text{filt}} \in \mathbb{R}^{N_x \times N_e}$, inflation $\lambda$.

**Require:** Let ET, RO, EU and CI represent Algorithms 1, 2, 3 and 4, respectively.

1: $\mathbf{E}_1^{\text{fore}} = \mathcal{M}_1\left(\mathbf{E}_0^{\text{filt}}\right)$

2: $\mathbf{T}, \boldsymbol{w} = \text{ET}\left(\mathbf{E}_1^{\text{fore}}, \mathcal{H}_1, \mathbf{R}_1, \boldsymbol{y}_1\right)$

3: $\mathbf{U} = \text{RO}(N_e)$

4: $\mathbf{E}_1^{\text{filt}} = \text{EU}\left(\mathbf{E}_1^{\text{fore}}, \mathbf{T}, \boldsymbol{w}, \mathbf{U}\right)$

5: $\mathbf{E}_1^{\text{filt}} = \text{CI}\left(\mathbf{E}_1^{\text{filt}}, \lambda\right)$

**Require:** Store $\mathbf{E}_0^{\text{filt}} := \mathbf{E}_1^{\text{filt}}$ for the next cycle

---

**Algorithm 6** Lag $L$, shift $S$, EnKS

---

**Require:** Lag= $L$, shift= $S$, observations $\boldsymbol{y}_{L:L-S+1}$, smoother ensemble states $\mathbf{E}_{L-S:0}^{\text{smth}}$, ensemble size $N_e$, inflation $\lambda$.

**Require:** Let ET, RO, EU and CI represent Algorithms 1, 2, 3 and 4 respectively.

1: $\mathbf{E}_{L-S}^{\text{filt}} := \mathbf{E}_{L-S}^{\text{smth}}$

2: **for** $k \in \{L-S+1, \cdots, L\}$ **do**

3:     $\mathbf{E}_k^{\text{fore}} = \mathcal{M}_k(\mathbf{E}_{k-1}^{\text{filt}})$

4:     $\mathbf{T}, \boldsymbol{w} = \text{ET}\left(\mathbf{E}_k^{\text{fore}}, \mathcal{H}_k, \mathbf{R}_k, \boldsymbol{y}_k\right)$

5:     $\mathbf{U} = \text{RO}(N_e)$

6:     $\mathbf{E}_k^{\text{filt}} = \text{EU}\left(\mathbf{E}_k^{\text{fore}}, \mathbf{T}, \boldsymbol{w}, \mathbf{U}\right)$

7:     **for** $j \in \{0, \cdots, k-1\}$ **do**

8:         $\mathbf{E}_j^{\text{smth}} = \text{EU}\left(\mathbf{E}_j^{\text{smth}}, \mathbf{T}, \boldsymbol{w}, \mathbf{U}\right)$

9:     **end for**

10:     $\mathbf{E}_k^{\text{filt}} = \text{CI}\left(\mathbf{E}_k^{\text{filt}}, \lambda\right)$

11:     $\mathbf{E}_k^{\text{smth}} := \mathbf{E}_k^{\text{filt}}$

12: **end for**

**Require:** Store $\mathbf{E}_{L-S:0}^{\text{smth}} := \mathbf{E}_{L:S}^{\text{smth}}$ for the next cycle

---





---

**Algorithm 7** Gauss-Newton, lag $L$ shift $S$ IEnKS, SDA transform version

---

**Require:** Lag= $L$, shift=$S$, observations $\boldsymbol{y}_{L:L-S+1}$.

**Require:** $\mathbf{E}_0^{\mathrm{smth}} \in \mathbb{R}^{N_e \times N_e}$

**Require:** Let RO, EU and CI represent algorithms 2, 3 and 4 respectively.

**Require:** Parameters tol, $j_{\max}$, inflation $\lambda$.

1: $\mathbf{T} := \mathbf{I}_{N_e}$

2: $\mathbf{E}_0 := \mathbf{E}_0^{\mathrm{smth}}$

3: $j = 0, \boldsymbol{w} = \mathbf{0}$

4: **loop**

5:     **for** $k \in \{1, \cdots, L\}$ **do**

6:         $\mathbf{E}_k = \mathcal{M}_k(\mathbf{E}_{k-1})$

7:         **if** $k \in \{L-S+1, \cdots, L\}$ **then**

8:             $\mathbf{Y}_k = \mathcal{H}_k(\mathbf{E}_k)$

9:             $\hat{\boldsymbol{y}}_k = \mathbf{Y}_k \mathbf{1}/N_e$

10:            $\mathbf{S}_k = \mathbf{R}_k^{-\frac{1}{2}} \left(\mathbf{Y}_k - \hat{\boldsymbol{y}}_k \mathbf{1}^\top\right) \mathbf{T}^{-1}$

11:            $\hat{\boldsymbol{\delta}}_k = \mathbf{R}_k^{-\frac{1}{2}} \left(\boldsymbol{y}_k - \hat{\boldsymbol{y}}_k\right)$

12:         **end if**

13:     **end for**

14:     $\nabla \widetilde{\mathcal{J}} = (N_e - 1)\boldsymbol{w} - \sum_{k=L-S+1}^{L} \mathbf{S}_k^\top \hat{\boldsymbol{\delta}}_k$

15:     $\widetilde{\mathbf{H}}_{\widetilde{\mathcal{J}}} = (N_e - 1)\mathbf{I}_{N_e} + \sum_{k=L-S+1}^{L} \mathbf{S}_k^\top \mathbf{S}_k$

16:     $\Delta \boldsymbol{w} = \widetilde{\mathbf{H}}_{\widetilde{\mathcal{J}}}^{-1} \nabla \widetilde{\mathcal{J}}$

17:     $\boldsymbol{w} := \boldsymbol{w} - \Delta \boldsymbol{w}$

18:     $j := j + 1$

19:     **if** $\| \Delta \boldsymbol{w} \| < $ tol **or** $j = j_{\max}$ **then**

20:         **break loop**

21:     **else**

22:         $\mathbf{T} = \widetilde{\mathbf{H}}_{\widetilde{\mathcal{J}}}^{-\frac{1}{2}}$

23:         $\mathbf{E}_0 = \mathrm{EU}\left(\mathbf{E}_0^{\mathrm{smth}}, \mathbf{T}, \boldsymbol{w}, \mathbf{I}_{N_e}\right)$

24:     **end if**

25: **end loop**

26: $\mathbf{T} = \widetilde{\mathbf{H}}_{\widetilde{\mathcal{J}}}^{-\frac{1}{2}}$

27: $\mathbf{U} = \mathrm{RO}(N)$

28: $\mathbf{E}_0 := \mathrm{EU}\left(\mathbf{E}_0^{\mathrm{smth}}, \mathbf{T}, \boldsymbol{w}, \mathbf{U}\right)$

29: **for** $k = 1, \cdots, L+S$ **do**

30:     $\mathbf{E}_k = \mathcal{M}_k(\mathbf{E}_{k-1})$

31: **end for**

32: $\mathbf{E}_{L-S:0}^{\mathrm{smth}} := \mathbf{E}_{L-S:0}$

33: $\mathbf{E}_{L:L-S+1}^{\mathrm{filt}} := \mathbf{E}_{L:L-S+1}$

34: $\mathbf{E}_{L+S:L+1}^{\mathrm{fore}} := \mathbf{E}_{L+S:L+1}$

35: $\mathbf{E}_S^{\mathrm{smth}} = \mathrm{CI}\left(\mathbf{E}_S^{\mathrm{smth}}, \lambda\right)$

**Require:** $\mathbf{E}_0^{\mathrm{smth}} := \mathbf{E}_S^{\mathrm{smth}}$ for the next cycle.





---

**Algorithm 8** Lag $L$ shift $S$ SIEnKS, SDA version

---

**Require:** Lag= $L$, shift= $S$, observations $\boldsymbol{y}_{L:L-S+1}$, ensemble posterior states $\mathbf{E}_0^{\mathrm{smth}}$ and $\mathbf{E}_{L-S}^{\mathrm{smth}}$, inflation $\lambda$.

**Require:** Let ET, RO, EU and CI represent Algorithms 1, 2, 3 and 4, respectively.

1: $\mathbf{E}_{L-S}^{\mathrm{filt}} := \mathbf{E}_{L-S}^{\mathrm{smth}}$

2: **for** $k \in \{L-S+1, \cdots, L\}$ **do**

3:      $\mathbf{E}_k^{\mathrm{fore}} = \mathcal{M}_k(\mathbf{E}_{k-1}^{\mathrm{filt}})$

4:      $\mathbf{T}, \boldsymbol{w} = \mathrm{ET}\left(\mathbf{E}_k^{\mathrm{fore}}, \mathcal{H}_k, \mathbf{R}_k, \boldsymbol{y}_k\right)$

5:      $\mathbf{U}_k = \mathrm{RO}(N)$

6:      $\mathbf{E}_k^{\mathrm{filt}} = \mathrm{EU}\left(\mathbf{E}_k^{\mathrm{fore}}, \mathbf{T}, \boldsymbol{w}, \mathbf{U}_k\right)$

7:      $\mathbf{E}_0^{\mathrm{smth}} = \mathrm{EU}\left(\mathbf{E}_0^{\mathrm{smth}}, \mathbf{T}, \boldsymbol{w}, \mathbf{U}_k\right)$

8: **end for**

9: $\mathbf{E}_0^{\mathrm{smth}} := \mathrm{CI}\left(\mathbf{E}_0^{\mathrm{smth}}, \lambda\right)$

10: **for** $k = 1, \cdots, L$ **do**

11:      $\mathbf{E}_k^{\mathrm{smth}} = \mathcal{M}_k^{\mathrm{smth}}(\mathbf{E}_{k-1})$

12: **end for**

**Require:** $\mathbf{E}_0^{\mathrm{smth}} := \mathbf{E}_S^{\mathrm{smth}}$, $\mathbf{E}_{L-S}^{\mathrm{smth}} := \mathbf{E}_L^{\mathrm{smth}}$ for the next cycle.

---





---

**Algorithm 9** Maximum Likelihood Ensemble Transform (MLET)

---

**Require:** Ensemble matrix $\mathbf{E} \in \mathbb{R}^{N_x \times N_e}$, observation map $\mathcal{H}$, observation error covariance $\mathbf{R} \in \mathbb{R}^{N_y \times N_y}$ and observation vector $\boldsymbol{y}$.

**Require:** Parameters $\mathrm{tol}, j_{\max}$

1: **Require:**

2: $\mathbf{T} = \mathbf{I}_{N_e}$

3: $j = 0, \boldsymbol{w} = \mathbf{0}$

4: $\mathbf{E}_0 = \mathbf{E}$

5: **loop**

6: $\quad \mathbf{Y} = \mathcal{H}(\mathbf{E})$

7: $\quad \hat{\boldsymbol{y}} = \mathbf{Y}\mathbf{1}/N_e$

8: $\quad \mathbf{S} = \mathbf{R}^{-\frac{1}{2}} \left( \mathbf{Y} - \hat{\boldsymbol{y}}\mathbf{1}^\top \right) \mathbf{T}^{-1}$

9: $\quad \hat{\boldsymbol{\delta}} = \mathbf{R}^{-\frac{1}{2}} \left( \boldsymbol{y} - \hat{\boldsymbol{y}} \right)$

10: $\quad \nabla \widetilde{\mathcal{J}} = (N_e - 1)\boldsymbol{w} - \mathbf{S}^\top \hat{\boldsymbol{\delta}}$

11: $\quad \widetilde{\mathbf{H}}_{\widetilde{\mathcal{J}}} = (N_e - 1)\mathbf{I}_{N_e} + \mathbf{S}^\top \mathbf{S}$

12: $\quad \Delta \boldsymbol{w} = \widetilde{\mathbf{H}}_{\widetilde{\mathcal{J}}}^{-1} \nabla \widetilde{\mathcal{J}}$

13: $\quad \boldsymbol{w} := \boldsymbol{w} - \Delta \boldsymbol{w}$

14: $\quad$ **if** $\| \Delta \boldsymbol{w} \| < \mathrm{tol}$ **or** $j = j_{\max}$ **then**

15: $\quad\quad$ **break loop**

16: $\quad$ **else**

17: $\quad\quad \mathbf{T} = \widetilde{\mathbf{H}}_{\widetilde{\mathcal{J}}}^{-\frac{1}{2}}$

18: $\quad\quad \mathbf{E} = \mathrm{EU}\left( \mathbf{E}_0, \mathbf{T}, \boldsymbol{w}, \mathbf{I}_{N_e} \right)$

19: $\quad$ **end if**

20: **end loop**

21: $\mathbf{T} = \widetilde{\mathbf{H}}_{\widetilde{\mathcal{J}}}^{-\frac{1}{2}}$

22: **return** $\mathbf{T}, \boldsymbol{w}$

---





---

**Algorithm 10** Finite-size ensemble transform, Gauss-Newton approximation (FSET)

---

**Require:** Ensemble matrix $\mathbf{E} \in \mathbb{R}^{N_x \times N_e}$, observation map $\mathcal{H}$, observation error covariance $\mathbf{R} \in \mathbb{R}^{N_y \times N_y}$ and observation vector $\boldsymbol{y}$.

**Require:** Parameters tol, $j_{\max}$

1: $\mathbf{T} = \mathbf{I}_{N_e}$

2: $j = 0, \boldsymbol{w} = \mathbf{0}$

3: $\mathbf{E}_0 = \mathbf{E}$

4: $\epsilon_{N_e} = 1 + 1/N_e$, $N_{\text{eff}} = N_e + 1$

5: **loop**

6:     $\mathbf{Y} = \mathcal{H}(\mathbf{E})$

7:     $\hat{\boldsymbol{y}} = \mathbf{Y}\mathbf{1}/N_e$

8:     $\mathbf{S} = \mathbf{R}^{-\frac{1}{2}}\left(\mathbf{Y} - \hat{\boldsymbol{y}}\mathbf{1}^{\top}\right)\mathbf{T}^{-1}$

9:     $\hat{\boldsymbol{\delta}} = \mathbf{R}^{-\frac{1}{2}}\left(\boldsymbol{y} - \hat{\boldsymbol{y}}\right)$

10:     $\zeta = 1/\left(\epsilon_{N_e} + \boldsymbol{w}^{\top}\boldsymbol{w}\right)$

11:     $\nabla\widetilde{\mathcal{J}} = \zeta\left(N_{\text{eff}}\right)\boldsymbol{w} - \mathbf{S}^{\top}\hat{\boldsymbol{\delta}}$

12:     $\widetilde{\mathbf{H}}_{\widetilde{\mathcal{J}}} = (N_e - 1)\mathbf{I}_{N_e} + \mathbf{S}^{\top}\mathbf{S}$

13:     $\Delta\boldsymbol{w} = \widetilde{\mathbf{H}}_{\widetilde{\mathcal{J}}}^{-1}\nabla\widetilde{\mathcal{J}}$

14:     $\boldsymbol{w} := \boldsymbol{w} - \Delta\boldsymbol{w}$

15:     $j := j + 1$

16:     **if** $\|\Delta\boldsymbol{w}\| < \text{tol}$ **or** $j = j_{\max}$ **then**

17:        **break loop**

18:     **else**

19:        $\mathbf{T} = \widetilde{\mathbf{H}}_{\widetilde{\mathcal{J}}}^{-\frac{1}{2}}$

20:        $\mathbf{E} = \text{EU}\left(\mathbf{E}_0, \mathbf{T}, \boldsymbol{w}, \mathbf{I}_{N_e}\right)$

21:     **end if**

22: **end loop**

23: $\zeta = 1/\left(\epsilon_N + \boldsymbol{w}^{\top}\boldsymbol{w}\right)$

24: $\widetilde{\mathbf{H}}_{\widetilde{\mathcal{J}}} = N_{\text{eff}}\left(\zeta\mathbf{I}_N - 2\zeta^2\boldsymbol{w}\boldsymbol{w}^{\top}\right) + \mathbf{S}^{\top}\mathbf{S}$

25: $\mathbf{T} = \widetilde{\mathbf{H}}_{\widetilde{\mathcal{J}}}^{-\frac{1}{2}}$

26: **return** $\mathbf{T}, \boldsymbol{w}$

---





---

**Algorithm 11** Gauss-Newton, lag $L$ shift $S$ IEnKS-N, SDA transform version

---

**Require:** All lines are identical to Algorithm 7 with the exception of the following lines:

0: $\epsilon_{N_e} = 1 + \frac{1}{N_e}$, $N_{\text{eff}} = N_e + 1$

14: $\zeta = 1/\left(\epsilon_{N_e} + \boldsymbol{w}^\top \boldsymbol{w}\right)$,

$\nabla \widetilde{\mathcal{J}} = \zeta \left(N_{\text{eff}}\right) \boldsymbol{w} - \sum_{k=L-S+1}^{L} \mathbf{S}_k^\top \hat{\boldsymbol{\delta}}_k$

15: $\widetilde{\mathbf{H}}_{\widetilde{\mathcal{J}}} = (N_{\text{eff}} - 1)\mathbf{I}_N + \sum_{k=L-S+1}^{L} \mathbf{S}_k^\top \mathbf{S}_k$

26: $\zeta = 1/\left(\epsilon_{N_e} + \boldsymbol{w}^\top \boldsymbol{w}\right)$,

$\widetilde{\mathbf{H}}_{\widetilde{\mathcal{J}}} = N_{\text{eff}} \left(\zeta \mathbf{I}_{N_e} - 2\zeta^2 \boldsymbol{w}\boldsymbol{w}^\top\right) + \sum_{k=L-S+1}^{L} \mathbf{S}_k^\top \mathbf{S}_k$,

$\mathbf{T} = \widetilde{\mathbf{H}}_{\widetilde{\mathcal{J}}}^{-\frac{1}{2}}$

35:

---

---

**Algorithm 12** Lag $L$, shift $S$ SIEnKS, MDA version

---

**Require:** Lag= $L$, shift= $S$, observations $\boldsymbol{y}_{L:1}$, MDA conditional ensemble $\mathbf{E}_0^{\text{mda}}$, ensemble size $N_e$, inflation $\lambda$.

**Require:** Let ET, RO, EU and CI represent Algorithms 1, 2, 3 and 4, respectively.

**Require:** Let $\{\beta_k\}_{k=1}^L$ and $\{\eta_k\}_{k=1}^L$ be the multiple data assimilation and balancing weights, respectively.

1: $\mathbf{E}_0^{\text{bal}} := \mathbf{E}_0^{\text{mda}}$

2: **for** $k = 1, \cdots, L$ **do**

3:      $\mathbf{U} = \text{RO}(N_e)$

4:      $\mathbf{E}_k^{\text{bal}} = \mathcal{M}_k(\mathbf{E}_{k-1}^{\text{bal}})$

5:      **if** $k \in \{L-S+1, \cdots, L\}$ **then**

6:          $\mathbf{E}_k^{\text{fore}} := \mathbf{E}_k^{\text{bal}}$

7:      **end if**

8:      $\mathbf{T}, \boldsymbol{w} = \text{ET}\left(\mathbf{E}_k^{\text{bal}}, \mathcal{H}_k, \mathbf{R}_k/\eta_k, \boldsymbol{y}_k\right)$

9:      $\mathbf{E}_k^{\text{bal}} = \text{EU}\left(\mathbf{E}_k^{\text{bal}}, \mathbf{T}, \boldsymbol{w}, \mathbf{U}\right)$

10:      **if** $k \in \{L-S+1, \cdots L\}$ **then**

11:          $\mathbf{E}_k^{\text{filt}} := \mathbf{E}_k^{\text{bal}}$

12:      **end if**

13:      **for** $k = 0, \cdots, k-1$ **do**

14:          $\mathbf{E}_k^{\text{bal}} = \text{EU}\left(\mathbf{E}_k^{\text{bal}}, \mathbf{T}, \boldsymbol{w}, \mathbf{U}\right)$

15:      **end for**

16:      **if** k=S **then**

17:          $\mathbf{E}_0^{\text{mda}} = \mathbf{E}_0^{\text{bal}}$

18:          $\mathbf{E}_S^{\text{mda}} = \mathbf{E}_k^{\text{bal}}$

19:      **end if**

20: **end for**

21: $\mathbf{E}_{0:L-S}^{\text{smth}} := \mathbf{E}_{0:L-S}^{\text{bal}}$

22: **for** $k = S+1, \cdots, L$ **do**

23:      $\mathbf{U} = \text{RO}(N_e)$

24:      $\mathbf{E}_k^{\text{mda}} = \mathcal{M}_k(\mathbf{E}_{k-1}^{\text{mda}})$

25:      $\mathbf{T}, \boldsymbol{w} = \text{ET}\left(\mathbf{E}_k^{\text{mda}}, \mathcal{H}_k, \mathbf{R}_k/\beta_k, \boldsymbol{y}_k\right)$

26:      $\mathbf{E}_k^{\text{mda}} = \text{EU}\left(\mathbf{E}_k^{\text{mda}}, \mathbf{T}, \boldsymbol{w}, \mathbf{U}\right)$

27:      $\mathbf{E}_0^{\text{mda}} = \text{EU}\left(\mathbf{E}_0^{\text{mda}}, \mathbf{T}, \boldsymbol{w}, \mathbf{U}\right)$

28: **end for**

29: $\mathbf{E}_0^{\text{mda}} = \text{CI}\left(\mathbf{E}_0^{\text{mda}}, \lambda\right)$

30: **for** $k = 1, \cdots, S$ **do**

31:      $\mathbf{E}_k^{\text{mda}} = \mathcal{M}_k(\mathbf{E}_{k-1}^{\text{mda}})$

32: **end for**

**Require:** Store $\mathbf{E}_0^{\text{mda}} = \mathbf{E}_S^{\text{mda}}$ for the next cycle

---





---

**Algorithm 13** Gauss-Newton, lag $L$ shift $S$ IEnKS, MDA transform version

---

**Require:** Lag$= L$, shift$=S$, observations $\boldsymbol{y}_{L:1}$, conditional MDA ensemble $\mathbf{E}_0^{\mathrm{mda}}$, ensemble size $N_e$.

**Require:** Let RO, EU and CI represent algorithms 2, 3 and 4, respectively.

**Require:** Let $\{\beta_k\}_{k=1}^L$ and $\{\eta_k\}_{k=1}^L$ be the multiple data assimilation and balancing weights, respectively.

**Require:** Parameters tol, $j_{\max}$, inflation $\lambda$.

1: $\mathbf{T} = \mathbf{I}_{N_e}$

2: $j = 0, \boldsymbol{w} = \mathbf{0}$

3: **for** stage $= 1, 2$ **do**

4:     $\mathbf{E}_0 = \mathbf{E}_0^{\mathrm{mda}}$

5:     **if** stage $= 1$ **then**

6:         $\theta_k = \eta_k$

7:     **else**

8:         $\theta_k = \beta_k$

9:     **end if**

10:    **loop**

11:        **for** $k \in \{1, \cdots, L\}$ **do**

12:            $\mathbf{E}_k = \mathcal{M}_k(\mathbf{E}_{k-1})$

13:            $\hat{\boldsymbol{y}}_k = \mathcal{H}_k(\mathbf{E}_k)\mathbf{1}/N_e$

14:            $\mathbf{Y}_k = \mathcal{H}_k(\mathbf{E}_k)$

15:            $\mathbf{S}_k = \sqrt{\theta_k}\mathbf{R}_k^{-\frac{1}{2}}\left(\mathbf{Y}_k - \hat{\boldsymbol{y}}_k\mathbf{1}^\top\right)\mathbf{T}^{-1}$

16:            $\hat{\boldsymbol{\delta}}_k = \sqrt{\theta_k}\mathbf{R}_k^{-\frac{1}{2}}\left(\boldsymbol{y}_k - \hat{\boldsymbol{y}}_k\right)$

17:        **end for**

18:        $\nabla\widetilde{\mathcal{J}} = (N_e - 1)\boldsymbol{w} - \sum_{k=L-S+1}^{L}\mathbf{S}_k^\top\hat{\boldsymbol{\delta}}_k$

19:        $\widetilde{\mathbf{H}}_{\widetilde{\mathcal{J}}} = (N_e - 1)\mathbf{I}_{N_e} + \sum_{k=L-S+1}^{L}\mathbf{S}_k^\top\mathbf{S}_k$

20:        $\Delta\boldsymbol{w} = \widetilde{\mathbf{H}}_{\widetilde{\mathcal{J}}}^{-1}\nabla\widetilde{\mathcal{J}}$

21:        $\boldsymbol{w} := \boldsymbol{w} - \Delta\boldsymbol{w}$

22:        $j := j + 1$

23:        **if** $\| \Delta\boldsymbol{w} \| < $ tol **or** $j = j_{\max}$ **then**

24:            **break loop**

25:        **else**

26:            $\mathbf{T} = \widetilde{\mathbf{H}}_{\widetilde{\mathcal{J}}}^{-\frac{1}{2}}$

27:            $\mathbf{E}_0 = \mathrm{EU}\left(\mathbf{E}_0^{\mathrm{mda}}, \mathbf{T}, \boldsymbol{w}, \mathbf{I}_{N_e}\right)$

28:        **end if**

29:    **end loop**

30:    $\mathbf{T} = \widetilde{\mathbf{H}}_{\widetilde{\mathcal{J}}}^{-\frac{1}{2}}$

31:    $\mathbf{U} = \mathrm{RO}(N_e)$

32:    $\mathbf{E}_0 := \mathrm{EU}\left(\mathbf{E}_0^{\mathrm{mda}}, \mathbf{T}, \boldsymbol{w}, \mathbf{U}\right)$

33:    **if** stage $= 1$ **then**

34:        **for** $k = 1, \cdots, L + S$ **do**

35:            $\mathbf{E}_k = \mathcal{M}_k(\mathbf{E}_{k-1})$

36:        **end for**

37:        $\mathbf{E}_{L-S:0}^{\mathrm{smth}} := \mathbf{E}_{L-S:0}$

38:        $\mathbf{E}_{L:L-S+1}^{\mathrm{filt}} := \mathbf{E}_{L:L-S+1}$

39:        $\mathbf{E}_{L+1:L+S}^{\mathrm{fore}} := \mathbf{E}_{L+S:L+1}$

40:    **end if**

41: **end for**

42: **for** $k = 1, \cdots, S$ **do**

43:    $\mathbf{E}_k = \mathcal{M}_k(\mathbf{E}_{k-1})$

44: **end for**

45: $\mathbf{E}_S^{\mathrm{smth}} = \mathrm{CI}\left(\mathbf{E}_S^{\mathrm{smth}}, \lambda\right)$

**Require:** $\mathbf{E}_0^{\mathrm{smth}} := \mathbf{E}_S^{\mathrm{smth}}$ for the next cycle.

---



*Code availability.* The current version of DataAssimilationBenchmarks.jl is available on the project Github:

https://github.com/cgrudz/DataAssimilationBenchmarks.jl

and in the Julia General Registries under the Apache-2.0 License. The exact version of the package used to produce the results used in this paper is archived on Zenodo (Grudzien et al., 2021), as are scripts to process data and produce the plots for all the simulations presented in

this paper.

*Author contributions.* CG mathematically derived the original SDA and MDA SIEnKS schemes. CG and MB together refined and improved upon these mathematical results for their final form. All numerical simulation code and plotting code was developed by CG, MB shared original Python code for the IEnKS and the finite-size formalism schemes which contributed to the development of the Julia code supporting this work. CG and MB worked together on all conceptual diagrams. All numerical experiments and benchmark configurations for the SIEnKS

were devised together between CG and MB. The manuscript was written by CG with contributions from MB in refining the narrative and presentation of results in their final form.

*Competing interests.* The authors declare that they have no conflict of interest.

*Acknowledgements.* Special thanks go to Eric Olson, Grant Schissler and Mihye Ahn for high-performance computing support and logistics at the University of Nevada, Reno. Thanks go to Patrick Raanes for the open-source DAPPER Python package which was referenced at times

for the development of DA schemes in Julia. CEREA is a member of Institut Pierre-Simon Laplace.





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
