# Peer review of "A fast, single-iteration ensemble Kalman smoother for sequential data assimilation"

_Geoscientific Model Development, 2021_

## Referee Comment (RC2)

**Review of the manuscript gmd-2021-306 "A fast, single-iteration ensemble Kalman smoother for sequential data assimilation" by Colin Grudzien and Marc Bocquet**

Pavel Sakov

November 7, 2021

**1 General comments**

The paper gives an overview of the iterative ensemble Kalman smoother methods; introduces a new scheme called single-iteration ensemble Kalman smoother (SIEnKS); and runs a number of tests on it with Lorenz-96 model. It provides algorithms for the most significant in the context of the paper methods, and uses the open-source Julia package DataAssimilationBenchmarks.jl for the benchmarking.

It is not an easy paper to review, mainly due to its sheer length, but also because of some vagueness in formulating the purpose and results, and some language used. Just to be more specific – it is impossible to get an idea about the new method from a rather lengthy abstract apart from that it is new.

It would be a too large effort for me to give a proper review of the paper of this length; instead I will list a few points that may or may not be accepted by the authors.

**2 Issues**

1. My preference (perhaps contrary to the established practice) is to avoid characterising EnKF methods as "ensemble variational". The Kalman filter *is* a variational method, even if formulated in a sequential way. To me, it can make sense to talk about "ensemble variational" if the method explicitly uses the model adjoint.

2. Further, unlike to 3/4D-Var, I can not get any sense of the "outer loop" terminology in the paper. How is it different to the iterative minimisation?

(Also, I am not a big fan of the "4D-MAP" abbreviation.)

3. **L. 158-159: "ensemble is drawn", "columns sampled".**

In deterministic EnKF methods the ensemble *is not* a stochastic ensemble, but rather a (possibly, lossly compressed) factorisation of the state error covariance. Indeed, there can be some stochastic elements even in mainly deterministic systems, e.g. due to random perturbations of forcing etc., but using that statistical terminology is largely misleading, I belive.

4. **L. 363: "Raanes (2016) demonstrates the equivalence of the EnKS and the Rauch-Tung-Striebel (RTS)."**

I am surprised that this needs to be demonstrated.

5. **L. 410: "In the perfect, linear-Gaussian model, this formulation of the IEnKS is actually equivalent to the 4D-EnKF ..."**

This may be true in the specific context, but does not make sense on its own: how can a smoother be equivalent to a filter?

6. **. L. 449: "A revised and simplified form of the Gauss-Newton IEnKS, transform variant is presented for the first time in Algorithm 7."**

Hmm... I trust the authors that this algorithm must be a substantial achievement. Just wanted to note that it has 35 lines (with some functions), while a similar one takes only 19 lines in Table 2 of Bocquet and Sakov (2014).

7. **The paragraph l. 449-454.**

I would add "similarly to MLEF" somewhere in this paragraph.

8. **The SIEnKS.**

I struggle to understand what is actually new in the SIEnKS compared to the Lin-IEnKS (sorry). It would help if the authors explained it explicitely and/or put the two algorithms side by side to see the difference.

9. **The paragraph l. 491-504.**

   Because this paragraph writes more than just a few words on RIP, it may be useful to note for a non-specialist reader that RIP does not minimise the same cost function as the IEnKF. It makes the best fit to observations in the model subspace by assimilating them multiple times until convergence. This is equivalent to minimising the cost function with the forecast covariance multiplied by a large number.

   Further, in regard to l. 502-504 – there is no such thing as a single iteration RIP, I guess.

**3   Conclusion**

Overall, the paper is well written and in a way represents a nice overview of the ensemble smoother methods. Perhaps, it could read a bit easier if a more straightforward terminology was adopted. I have no comments on the experimental part of the paper, partly because it failed for me to generate the excitement of benchmarking a new extension, due my failure to understand the essence of the novelty of the SIEnKS. Having said that, the experiments look well done.

In my view, the paper aligns with the goals of the GMD and will be interesting to the growing EnKF data assimilating community. I leave it to the authors' discretion whether and how revise it to address the issues raised in this review. This probably translates as recommendation to **accept with minor revision**.

**References**

Bocquet, M. and P. Sakov, 2014: An iterative ensemble Kalman smoother. *Q. J. R. Meteorol. Soc.*, **140**, 1521–1535.

---

## Author Comment (AC1)

**1 Introduction**

We wish firstly to thank the anonymous reviewer and the named reviewer, Pavel Sakov, for their helpful comments and critique. We acknowledge that this is an unusually long work and a difficult manuscript to review for this reason. Also, we wish to thank the handling editor Adrian Sandu for his patience and for granting an extension to our response, as the lead author recently changed institutions. The extra time gave us the opportunity to carefully review the comments from the reviewers and revise our manuscript accordingly. We hope that our responses in the below are satisfactory. Changes to the manuscript are highlighted in a LaTeX diff attached at the end of this document, but specific items are discussed point-wise in the below.
* * *
**2 Responses to Referee 1: Anonymous**

**Point** 1 ⎯⎯⎯⎯⎯⎯⎯⎯⎯⎯⎯⎯⎯⎯⎯⎯⎯⎯⎯⎯⎯⎯⎯⎯⎯⎯⎯⎯⎯⎯⎯⎯⎯⎯⎯⎯⎯⎯⎯⎯⎯⎯⎯⎯⎯⎯⎯⎯⎯

**Referee:**

"It is easy to feel lost in the jargons. It would be helpful to the reader if the authors were to explain certain words more explicitly, which I have outlined in the specific comments section."

**Response:**

This is noted and we replaced jargon with plain English wherever it was simple to do so. Particularly, the "outer-loop" terminology was removed and replaced with descriptive English and the "inner-loop" terminology was replaced with "smoother loop" in the limited context of the EnKS. The "online" terminology was replaced with "real-time" as mentioned in the comments below. The data assimilation window (DAW) terminology is unfortunately key to most of our arguments, but we hope that the explicit definition and explanatory figure in our notations Section 2 is satisfactory.

**Point** 2 ⎯⎯⎯⎯⎯⎯⎯⎯⎯⎯⎯⎯⎯⎯⎯⎯⎯⎯⎯⎯⎯⎯⎯⎯⎯⎯⎯⎯⎯⎯⎯⎯⎯⎯⎯⎯⎯⎯⎯⎯⎯⎯⎯⎯⎯⎯⎯⎯⎯

**Referee:**

"The experiments were done on Lorenz-96 with the 40-variable setting while observing all states. The experiments would be more compelling in an operational sense with a larger test problem and sparse observations. Are you not running these experiments purely because of the difficulty in formulating localization?"

**Response:**

This is correct. Testing the SIEnKS in a realistic geophysical model would require the use of localization or hybridization to regularize the estimator, and we feel that studying localization / hybridization in the SIEnKS framework deserves a devoted and systematic investigation. We now clarify in our conclusion that domain localization, as in the LETKF, is likely to have an extension to the SIEnKS. However, there are also rich opportunities to iteratively optimize a localization hyper-parameter within this framework, and a mathematical / computational treatment of possible implementations to overcome the curse of dimensionality in this framework goes beyond the current scope.

**Point** 3 —————————————————————————————————————————————

**Referee:**

"Could you specify what is meant by outer-loop? Does this refer to the filtering step which is done first and then the inner loop of smoothing the lagged states?"

**Response:**

The "outer-loop" terminology was used as a reference to the cycle loop that shifts the algorithm between distinct data assimilation windows, and that contains the subroutines like the transform step and the ensemble update step. However, to clarify our work, we replaced this terminology with more plain English.

**Point** 4 —————————————————————————————————————————————

**Referee:**

"What is meant by 'online' forecast systems? Does this mean real-time?"

**Response:**

This is correct. We wish to distinguish between DA problems that can be run over a static data assimilation window (with a fixed set of observations) versus a shifting data assimilation window where new observations are added in real-time. We replaced "online" with "real-time" in all instances.

**Point** 5 —————————————————————————————————————————————

**Referee:**

"In line 10 of the abstract you write '...prediction/posterior accuracy...'. I feel that the word 'posterior' must be replaced by 'analysis', since prior/posterior is used with respect to the distributions while forecast/analysis is used with respect to the sample/realization."

**Response:**

We accept this suggestion and we adopted "analysis" or "posterior estimate" as terminology where applicable throughout the work to clarify the text.

**Point** 6 —————————————————————————————————————————————

**Referee:**

"In line 18, you write 'four-dimensional ensemble var'. If you just mean 4D-EnVar, you should go with writing 4D-EnVar."

**Response:**

This is correct and we accept this suggestion.

**Point 7** ―――――――――――――――――――――――――――――――――――――――――――――――――

**Referee:**

"In line 40, the last word 'by' should be replaced by 'be' to read '... may instead BE dominated by...' "

**Response:**

Thank you for your careful reading, this was a mistake and we accept this suggestion.

**Point 8** ―――――――――――――――――――――――――――――――――――――――――――――――――

**Referee:**

"In equations 28b) and 28c), you should be having $(\mathbf{I}_{N_x} + \boldsymbol{\Gamma}_1^\top \boldsymbol{\Gamma}_1)$ instead of the incorrect $(\mathbf{I}_{N_x} + \boldsymbol{\Gamma}_1 \boldsymbol{\Gamma}_1^\top)$ which you have. It should also be replaced in equation 36c."

**Response:**

Again, thank you for your careful reading, this was a mistake and we accept this suggestion.

―――――――――――――――――――――――――――――――――――――――――――――――――

**3   Responses to Referee 2: Pavel Sakov**

**Point 1** ―――――――――――――――――――――――――――――――――――――――――――――――――

**Referee:**

"It is not an easy paper to review, mainly due to its sheer length, but also because of some vagueness in formulating the purpose and results, and some language used. Just to be more specific – it is impossible to get an idea about the new method from a rather lengthy abstract apart from that it is new."

**Response:**

Thank you for the critique; the abstract and introduction are revised to make the implementation of the estimator clear early on. We trimmed some of the flourish in the language to make this more concise. We made a finer point of the objectives and results in the introduction. We hope that these changes make our work more clear.

**Point 2** ―――――――――――――――――――――――――――――――――――――――――――――――――

**Referee:**

"My preference (perhaps contrary to the established practice) is to avoid characterising EnKF methods as 'ensemble variational'. The Kalman filter is a variational method, even if formulated in a sequential way. To me, it can make sense to talk about 'ensemble variational' if the method explicitly uses the model adjoint."

**Response:**

We appreciate the characterization of the Kalman filter as a variational method, though, in our view, this is one of the many possible formulations of Kalman filtering methods. In order to emphasize the variational development of the ensemble Kalman filter and smoother in this work, we have adopted the ensemble-variational terminology, though we have not called this an "EnVAR" method which we agree refers to methods using explicit adjoints as you describe.

**Point** 3 ─────────────────────────────────────────────

**Referee:**

"Further, unlike to 3/4D-Var, I can not get any sense of the 'outer loop' terminology in the paper. How is it different to the iterative minimisation?
(Also, I am not a big fan of the '4D-MAP' abbreviation.)"

**Response:**

Thank you for the suggestions, the 4D-MAP terminology has been removed for greater clarity. The "outer-loop" terminology was used as a reference to the cycle loop that shifts the algorithm between distinct data assimilation windows, and that contains the sub-routines like the transform step and the ensemble update step. However, to clarify our work, we have replaced this terminology with more plain English.

**Point** 4 ─────────────────────────────────────────────

**Referee:**

"L. 158-159: 'ensemble is drawn', 'columns sampled'.
In deterministic EnKF methods the ensemble is not a stochastic ensemble, but rather a (possibly, lossly compressed) factorisation of the state error covariance. Indeed, there can be some stochastic elements even in mainly deterministic systems, e.g. due to random perturbations of forcing etc., but using that statistical terminology is largely misleading, I belive."

**Response:**

We believe that the uncertainty in the choice of optimal data to initialize a forecast simulation is reasonably represented with a probability distribution. The first prior in our work is thus meant to represent the uncertainty in the ensemble initialization. While the independent and identically distributed assumption is overly idealistic in practice, the statistical model for the uncertainty in initialization allows us to use the framework of hidden Markov models and Bayesian inference to derive our estimator. We agree that this mathematical framework is only a heuristic model for many of the realistic challenges in operational DA and we now put greater emphasis in the text that this is only an approximation. Please see the comments in, e.g., lines 116 - 133 and in lines 157 - 168 in the new version of the manuscript.

**Point** 5 ─────────────────────────────────────────────

**Referee:**

"L. 363: Raanes (2016) demonstrates the equivalence of the EnKS and the Rauch-Tung-Striebel (RTS).'
I am surprised that this needs to be demonstrated."

**Response:**

We feel that Raanes [2016] is a nontrivial manuscript and, while one may suspect this equivalence to hold without any demonstration, we believe it provides a good reference for readers unfamiliar with the RTS scheme. Furthermore, introducing this scheme in relationship to the EnKS is useful to the development of the SIEnKS because of the well-known property of the equivalence of the retrospective analysis and the reverse-time model forecast for this estimator in perfect dynamics.

**Point 6** ______________________________________________________________

**Referee:**

*"L. 410: 'In the perfect, linear-Gaussian model, this formulation of the IEnKS is actually equivalent to the 4D-EnKF...'*
This may be true in the specific context, but does not make sense on its own: how can a smoother be equivalent to a filter?"

**Response:**

The 4D extended state formalism is used in a variety of contexts to prove mathematical results about smoothers using classical filtering results in the linear-Gaussian model hypothesis, see, e.g., Chapter 7 of Anderson and Moore [1979]. Because the 4D-EnKF was a seminal work that influenced the development of a variety of modern ensemble Kalman smoothers, we felt that this correspondence was worthy of mention.

**Point 7** ______________________________________________________________

**Referee:**

*"L. 449: 'A revised and simplified form of the Gauss-Newton IEnKS, transform variant is presented for the first time in Algorithm 7.'*
Hmm... I trust the authors that this algorithm must be a substantial achievement. Just wanted to note that it has 35 lines (with some functions), while a similar one takes only 19 lines in Table 2 of Bocquet and Sakov (2014)."

**Response:**

This statement was not correct on our part, and we have removed language about the simplification and / or refinement of the IEnKS from the manuscript. Our intent was to update the algorithm, and to represent it in terms of common sub-routines that exist throughout the variety of ETKF style estimators that we study in this work. Decomposing these estimators as such, we wished to highlight the commonality and differences of the schemes for our algorithmic cost analysis. We agree that this was not a simplification, and instead was a change of perspective that suited our analysis.

**Point 8** ______________________________________________________________

**Referee:**

*"The paragraph l. 449-454.*
I would add 'similarly to MLEF' somewhere in this paragraph."

**Response:**

It is not totally clear to us in what way the above comparison with the MLEF is intended. If this refers to how the MLEF scales in computational cost in Hessian inversions, this is specifically addressed at the end of section 4.1, lines 519 - 523. If this is regarding how the ensemble transform of the MLEF is similar to algorithm 1, this is addressed in section 4.1, lines 512 - 518, and algorithm 9. Please clarify this point if we have not addressed it in the current version.

**Point 9** _______________________________________________________________

**Referee:**

"*The SIEnKS.*
I struggle to understand what is actually new in the SIEnKS compared to the Lin-IEnKS (sorry). It would help if the authors explained it explicitely and/or put the two algorithms side by side to see the difference."

**Response:**

We try to address this critical point of your review carefully by including the revised abstract, as well as emphasizing the difference between the 4D global analysis approach used in the Lin-IEnKS, and the 3D filtering / retrospective analysis approach of the SIEnKS in the language used throughout the work. Please note that the comparison between the SIEnKS and the Lin-IEnKS is specifically addressed in Section 3.4.1, lines 467 - 477, and in Figure 3 of the new version. Likewise, the discussion of nonlinear observation operators in Section 4.1 and the finite-size formalism in section 4.2 highlight how the cost of the Lin-IEnKS and the SIEnKS scale differently due to the optimization of the 3D versus 4D cost functions.

**Point 10** _______________________________________________________________

**Referee:**

"*The paragraph l. 491-504.*
Because this paragraph writes more than just a few words on RIP, it may be useful to note for a non-specialist reader that RIP does not minimise the same cost function as the IEnKF. It makes the best fit to observations in the model subspace by assimilating them multiple times until convergence. This is equivalent to minimising the cost function with the forecast covariance multiplied by a large number. Further, in regard to l. 502-504 – there is no such thing as a single iteration RIP, I guess."

**Response:**

Thank you for pointing this out, actually this was a subtly that was missed by the lead author in the original version of the manuscript. In order to clarify this important point, additional explanation of the correspondences and differences between the methods is added in lines of the new version of the manuscript in lines.

**References**

B. D. O. Anderson and J. B. Moore. *Optimal Filtering*. Prentice-Hall, Inc, Englewood Cliffs, New Jersey, 1979.

[revised manuscript text omitted]

---

## Author Response (AR2)

**1 Introduction**

We wish firstly to thank the anonymous reviewer and the named reviewer, Pavel Sakov, for their helpful comments and critique once again. At request of Pavel Sakov, we have introduced an additional Figure 4 in the current draft to demonstrate the differences in the analyses of the SIEnKS and the Lin-IEnKS. We believe that this new figure is indeed a helpful demonstration of our concept, and we have introduced some additional minor changes to the language in the text to emphasize the differences in the analyses demonstrated in the new figure. Note, we have not currently formatted the two diagrams of the SIEnKS and the Lin-IEnKS side-by-side as Pavel Sakov requested, though we expect all figures to have some minor revisions to their formatting for a two-column layout and we will address this formatting for the final version. Changes to the manuscript are highlighted in a LaTeX diff file in the supplementary documents.